# Connectome-constrained networks predict neural activity across the fly visual system

Janne K. Lappalainen[1,2,3], Fabian D. Tschopp[3], Sridhama Prakhya[3], Mason McGill[3,4], Aljoscha Nern[3], Kazunori Shinomiya[3], Shin-ya Takemura[3], Eyal Gruntman[3,5], Jakob H. Macke[1,2,6] & Srinivas C. Turaga[3✉]

We can now measure the connectivity of every neuron in a neural circuit[1–9], but we cannot measure other biological details, including the dynamical characteristics of each neuron. The degree to which measurements of connectivity alone can inform the understanding of neural computation is an open question[10]. Here we show that with experimental measurements of only the connectivity of a biological neural network, we can predict the neural activity underlying a specified neural computation. We constructed a model neural network with the experimentally determined connectivity for 64 cell types in the motion pathways of the fruit fly optic lobe[1–5] but with unknown parameters for the single-neuron and single-synapse properties. We then optimized the values of these unknown parameters using techniques from deep learning[11], to allow the model network to detect visual motion[12]. Our mechanistic model makes detailed, experimentally testable predictions for each neuron in the connectome. We found that model predictions agreed with experimental measurements of neural activity across 26 studies. Our work demonstrates a strategy for generating detailed hypotheses about the mechanisms of neural circuit function from connectivity measurements. We show that this strategy is more likely to be successful when neurons are sparsely connected—a universally observed feature of biological neural networks across species and brain regions.

Electrical signals propagating through networks of neurons form the basis of computations such as visual motion detection. The propagation of neural activity is shaped by both the functional properties of individual neurons and their synaptic connectivity. Additional factors[10,13], including electrical synapses, neuromodulation and glia, are known to further influence neural activity on multiple timescales. Volume electron microscopy can now be used to comprehensively measure the connectivity of each neuron in a neural circuit, and even entire nervous systems[1–9]. However, we do not yet have the means to also comprehensively measure all other biological details, including the dynamical properties of every neuron and synapse in the same circuit[13]. For these reasons, there has been considerable debate about the utility of connectome measurements for understanding brain function[14]. It is unclear whether it is possible to use only measurements of connectivity to generate accurate predictions about how the neural circuit functions, especially in the absence of direct measurements of neural activity from a living brain. There is considerable evidence from computer science and neuroscience that there is not necessarily a strong link between the connectivity of a neural network and its computational function. Universal function approximation theorems for artificial neural networks[15] imply that the same computational task can be performed by many different networks with very different neural connectivity. Empirically, there exist many classes of general-purpose artificial neural network architectures that can be trained to perform the same computational task[11]. Such differences in connectivity can correspond to qualitatively different computational mechanisms[16]. Similarly, in neuroscience there have been competing proposals for the same computation (for instance, the computation of visual motion)[17,18]. Furthermore, even circuits with the same connectivity can function differently[19]. Thus, neither the connectivity of a circuit alone, nor its computational task alone, can uniquely determine the mechanism of circuit function[20].

Here we show that the connectivity of a neural circuit, together with knowledge of its computational task, enables accurate predictions of the role played by individual neurons in the circuit in the computational task. We constructed a differentiable[21] model neural network with a close correspondence to the brain, whose connectivity was given by connectome measurements and with unknown single-neuron and single-synapse parameters. We optimized the unknown parameters of the model using techniques from deep learning[11], to enable the model to accomplish the computational task[22]. We call such models connectome-constrained and task-optimized deep mechanistic networks (DMNs) (Fig. 1a).

We applied this approach to model the motion pathways in the optic lobe of the *Drosophila* visual system. We constructed a DMN with experimentally measured connectivity[1–5], and unknown parameters for the

[1]Machine Learning in Science, Tübingen University, Tübingen, Germany. [2]Tübingen AI Center, Tübingen, Germany. [3]Janelia Research Campus, Howard Hughes Medical Institute, Ashburn, VA, USA. [4]Computation and Neural Systems, California Institute of Technology, Pasadena, CA, USA. [5]Dept of Biological Sciences, University of Toronto Scarborough, Toronto, Ontario, Canada. [6]Max Planck Institute for Intelligent Systems, Tübingen, Germany. ✉e-mail: turagas@janelia.hhmi.org

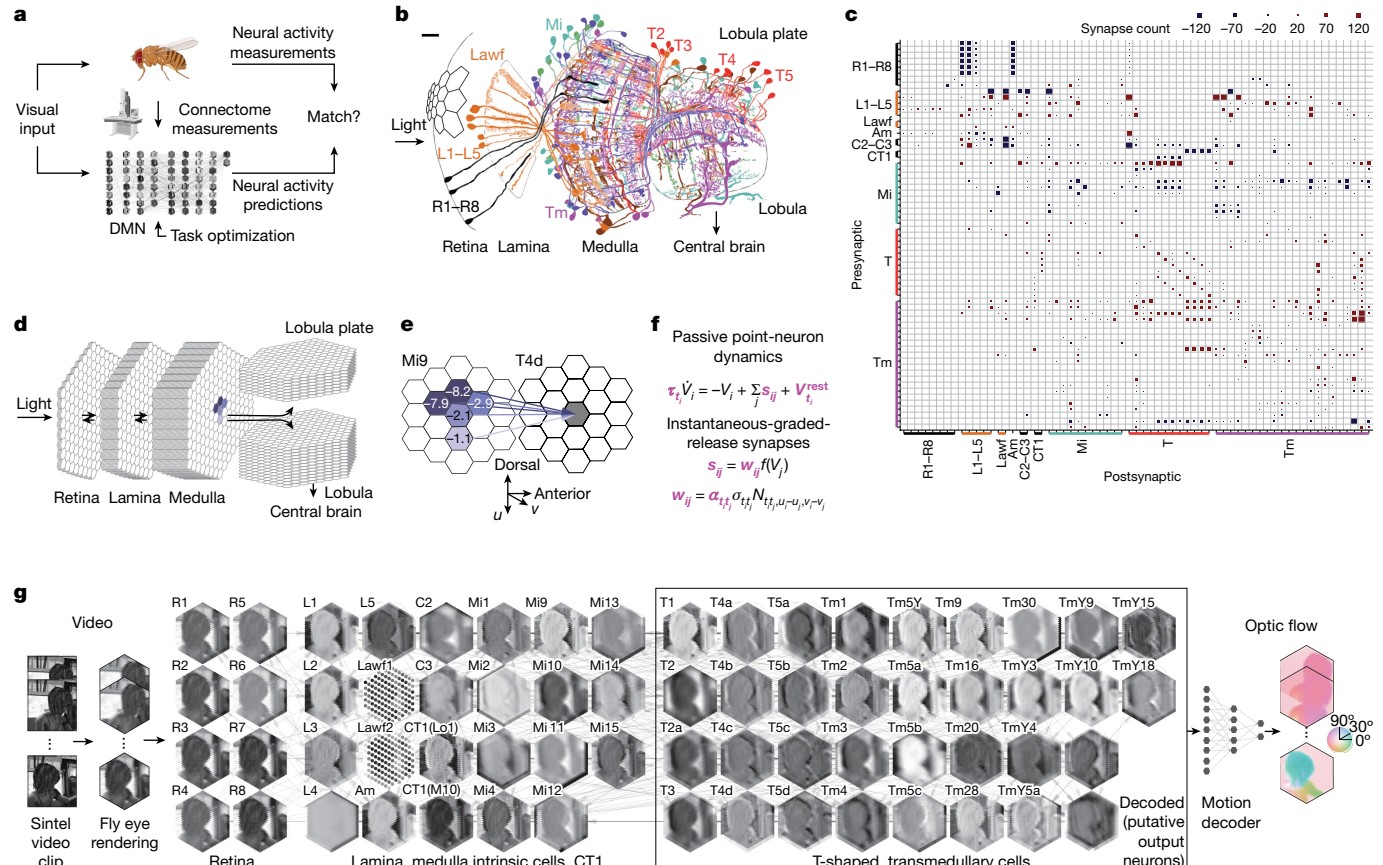

**Fig. 1 | Connectome-constrained and task-optimized models of the fly visual system. a**, DMNs aim to satisfy three constraints: the architecture is based on connectome measurements (**b**–**e**); cellular and synaptic dynamics are given by simple mechanistic models (**f**); and free parameters are task-optimized by training the model to perform optic flow estimation (**g**). Graphics of fruit fly and microscope were created with BioRender.com. **b**, Schematic of optic lobe of *D. melanogaster* with several processing stages (neuropils) and cell types, including photoreceptor (R1–R8), lamina monopolar (L), lamina wide-field (Lawf), medulla intrinsic (Mi), transmedullary (Tm) and T-shaped (T) neurons (adapted from ref. 25, Springer Nature). Scale bar, 10 μm. **c**, Identified connectivity between 64 modelled cell types, represented by total number of synapses from all neurons of a given presynaptic cell type to a postsynaptic cell of a given type. Amacrine (Am), centrifugal (C2–C3) and complex tangential (CT) neurons are also included. Blue (red) colour indicates putative hyperpolarizing (depolarizing) inputs; size of squares indicates number of input synapses.

**d**, Retinotopic hexagonal lattice columnar organization of visual system model. Each lattice represents a cell type; each hexagon represents an individual cell. Positions of photoreceptor ommatidia are aligned with downstream columns. The model comprises synapses from all neuropils (Supplementary Fig. 1). **e**, Example of convolutional filter, representing Mi9 inputs onto T4d cells. Values represent the average number of synapses projecting from presynaptic Mi9 cells in columns with indicated offset onto the postsynaptic dendrite of T4d cells. **f**, Single-neuron and synaptic dynamics are given by simple mechanistic models. Free parameters (magenta) are optimized by training the recurrent network model to perform optic flow estimation. **g**, Illustration of DMN performing optic flow estimation. Each hexagonal lattice shows a snapshot of simulated voltage levels of all cells of each type in response to input to the photoreceptors (R1–R8). Edges illustrate connectivity between cell types. A decoder receives the simulated neural activity of all output neurons to compute optic flow. Parameters of DMN and decoder are optimized using deep learning.

single-neuron dynamics and the strength of a unitary synapse. We optimized the model parameters on the computer vision task of detecting motion in dynamic visual stimuli[12]. Visual motion computation in the fly and its mechanistic underpinnings have been extensively studied[23]. Thus, we were able to compare the detailed predictions of our model with experimental measurements of neural activity in response to visual stimuli, on a neuron-by-neuron basis. We found that our connectome-constrained and task-optimized DMN accurately predicts the separation of the visual system into light-increment (ON) and light-decrement (OFF) channels, as well as the generation of direction selectivity in the well-known T4 and T5 motion detector neurons[24]. We release our model as a resource for the community (https://github.com/TuragaLab/flyvis).

## DMN of the fly visual system

The optic lobes of the fruit fly are equivalent to the mammalian retina. They comprise several layered neuropils whose columnar arrangement has a one-to-one correspondence with the ommatidia, both possessing a remarkably crystalline organization in a hexagonal lattice. Visual input from the photoreceptors is received by the lamina and medulla, which send projections to the lobula and lobula plate[25] (Fig. 1b). Many components of the optic lobe are highly regular, with columnar cell types appearing once per column, and multicolumnar neurons appearing with only small deviations from a well-defined periodicity in columnar space[25,26]. Several studies have reported on the local connectivity in the optic lobe and its motion pathways[1–5]. We assembled these separate local reconstructions into a coherent local connectome spanning the retina, lamina, medulla, lobula and lobula plate (Fig. 1c, Supplementary Note 1, Supplementary Fig. 1 and Supplementary Data files 1–3).

We approximated the circuitry across the entire visual field as perfectly periodic[2,26], and tiled this local connectivity architecture in a hexagonal lattice across retinotopic space to construct a consensus connectome for 64 cell types across the central visual field of the right eye (Fig. 1d, Methods, Extended Data Fig. 1, Supplementary Fig. 2 and Supplementary Data file 4). By this assumption of translation invariance

due to periodic tiling, the synapse count between each pair of neurons was the same across all pairs of neurons with the same presynaptic and postsynaptic cell type and relative location in retinotopic space. For simplicity, we refer here to this partial connectome of the motion pathways as the connectome.

We built a recurrent neural network modelling these first stages of visual processing in the optic lobe based on the connectome for the right eye. Each neuron in this DMN corresponds to a real neuron in the fly visual system, belonging to an identified cell type, and is connected to other neurons only if they are connected by synapses in the connectome (Fig. 1e). We constructed a model with detailed connectivity, but simplified models of single neurons and chemical synapses (Fig. 1f). We used passive leaky linear non-spiking voltage dynamics to model the time-varying activity of single neurons, as many neurons in the early visual system are non-spiking. We modelled neurons with a single electrical compartment, as this has previously been shown to be a good approximation given the small size of many neurons in the optic lobe[27]. The CT1 (complex tangential) neuron, which is among the largest in the brain, spanning the entire optic lobe, was modelled with one compartment per column in the medulla and lobula, as it is highly electrotonically compartmentalized[28] (Supplementary Note 2). We modelled the graded-release chemical synapses between non-spiking neurons with a threshold-linear function to approximate the nonlinear voltage-gated release of neurotransmitters. The resulting network model follows well-known threshold-linear dynamics and is piece-wise differentiable. Such dynamics are typically used to approximate the firing rates of a network of spiking neurons with the nonlinearity arising from spike generation, whereas in our network, the nonlinearity represents the voltage-gated neurotransmitter release. We used the cell-type structure of the connectome to reduce the number of free parameters in the model (Fig. 1f). We assumed that neurons of the same cell type shared the same neuron time constant and resting membrane potential. We modelled synaptic weights as proportional to the discrete number of synapses as reported in the connectome between a connected neuron pair[29], with a scale factor representing the strength of a unitary synapse. The unitary synapse scale factor and the sign of each synapse was the same for all pairs of neurons with the same pre- and postsynaptic cell type. In other words, a connection of five synapses from an Mi1 (medulla intrinsic) neuron to a T4 (T-shaped) neuron is assumed to be exactly half as strong as ten synapses between another pair of neurons of the same presynaptic and postsynaptic cell types, but could be stronger or weaker than five synapses between neurons of a different cell-type pair, for instance, from a Tm3 (transmedullar) neuron to a T4 neuron. The sign of each cell-type connection was determined by neurotransmitter and receptor expression profiling[30] (Methods and Supplementary Data file 2). In total, the connectome-constrained model comprises 45,669 neurons and 1,513,231 connections, across 64 cell types arranged in a hexagonal lattice consisting of 721 columns, modelling the central visual field of the roughly 700–900 ommatidia typically found in the fruit fly retina[31]. Connectome constraints, and our assumption of spatial homogeneity (that is, the hexagonally convolutional structure of the network), result in a marked reduction to just 734 free parameters for this large network model. The only free parameters in our model are the single-neuron time constants and resting membrane potentials (two parameters per cell type), and the unitary synapse strengths (one parameter per type-to-type connection). In the absence of connectome measurements, we would have needed to estimate well in excess of a million parameters corresponding to the weights of all possible connections (Methods).

We used task optimization[22] to further constrain the parameters of the model (that is, by training the model to perform a computational task that is thought to approximate the computations carried out by the circuit). We therefore implemented our recurrent DMN using the PyTorch library[21] (Methods) and used automatic differentiation to optimize the model using gradient-based deep learning training methods[11].

As the computational task constraining the input–output function of the circuit, we chose the computation of visual motion from naturalistic stimuli[12]. Motion computation in the fly visual system and its mechanistic underpinnings have been extensively studied[23]. This computation requires the neural circuit to compare visual stimuli across space and time, and thereby critically relies on temporal integration of visual information by the dynamics of the network. We reasoned that training our model to perform the computer vision task of optic flow computation[12] could help us identify circuit elements involved in motion computation. As our model contains many of the circuit elements that have been experimentally characterized and implicated in the computation of visual motion, we could then validate our model predictions.

To decode optic flow from the DMN, we used a decoding network to map the representation of motion used in the fly nervous system to the representation of optic flow specified by the computer vision task. This two-layer convolutional decoding network is given only the instantaneous neural activity of the medulla and downstream areas as input. Importantly, the decoding network cannot by itself detect motion, which requires the comparison of current and past visual stimuli, but must instead rely on the temporal dynamics of the DMN to compute motion-selective visual features. The resulting combination of our recurrent connectome-constrained DMN model and the feedforward decoding network was then trained end-to-end: we rendered video sequences from the Sintel database[12] as direct input to the photoreceptors of the connectome-constrained model, and used gradient descent (backpropagation through time[11]) to minimize the task error in predicting optic flow (Fig. 1g and Methods).

## DMN ensemble predicts known activity

We used only connectome and task constraints to construct our DMN, without any measurements of neural activity. We can therefore validate the model by comparing predictions of neural activity for each of the 64 identified cell types to experimental measurements. As it is possible that these constraints might not uniquely constrain model parameters[32], we generated an ensemble of 50 models, all constrained with the same connectome, and optimized to perform the same task. Each model in the ensemble corresponds to a local optimum of task performance. As the models achieved similar (but not identical) task performance, the ensemble reflects the diversity of possible models consistent with these constraints. The ensemble of models found a variety of parameter configurations (Supplementary Fig. 3), and exhibited superior task performance to both the decoder network alone and models with random parameter configurations (Extended Data Fig. 2a). We focused on the ten models that achieved the best task performance (Fig. 2a). We simulated neural responses to multiple experimentally characterized visual stimuli, and comprehensively compared model responses for each cell type to experimentally reported responses from 26 previously reported studies (Supplementary Note 3 and Supplementary Data files 5 and 6).

First, neural responses in the fly visual system are known to segregate into ON- and OFF-channels[33] defined by whether a neuron depolarizes more strongly to an increase or decrease in stimulus intensity, respectively, a hallmark of visual computation across species[34,35]. We probed the contrast preference of each cell type using flash stimuli[36] and found that the ensemble predicts the segregation into ON- and OFF-pathways with high accuracy: the median flash response index (FRI) across the ensemble predicts the correct ON- and OFF-preferred contrast selectivity for all 32 cell types for which contrast selectivity has been experimentally established (terminals of CT1 in the medulla and lobula are listed as CT1(M10) and CT1(Lo1)). This is also the case for the model with the best task performance (task-optimal model), which correctly predicts the preferred contrast of 30 of 32 cells (Fig. 2b). Furthermore, the ensemble provides predictions for the remaining

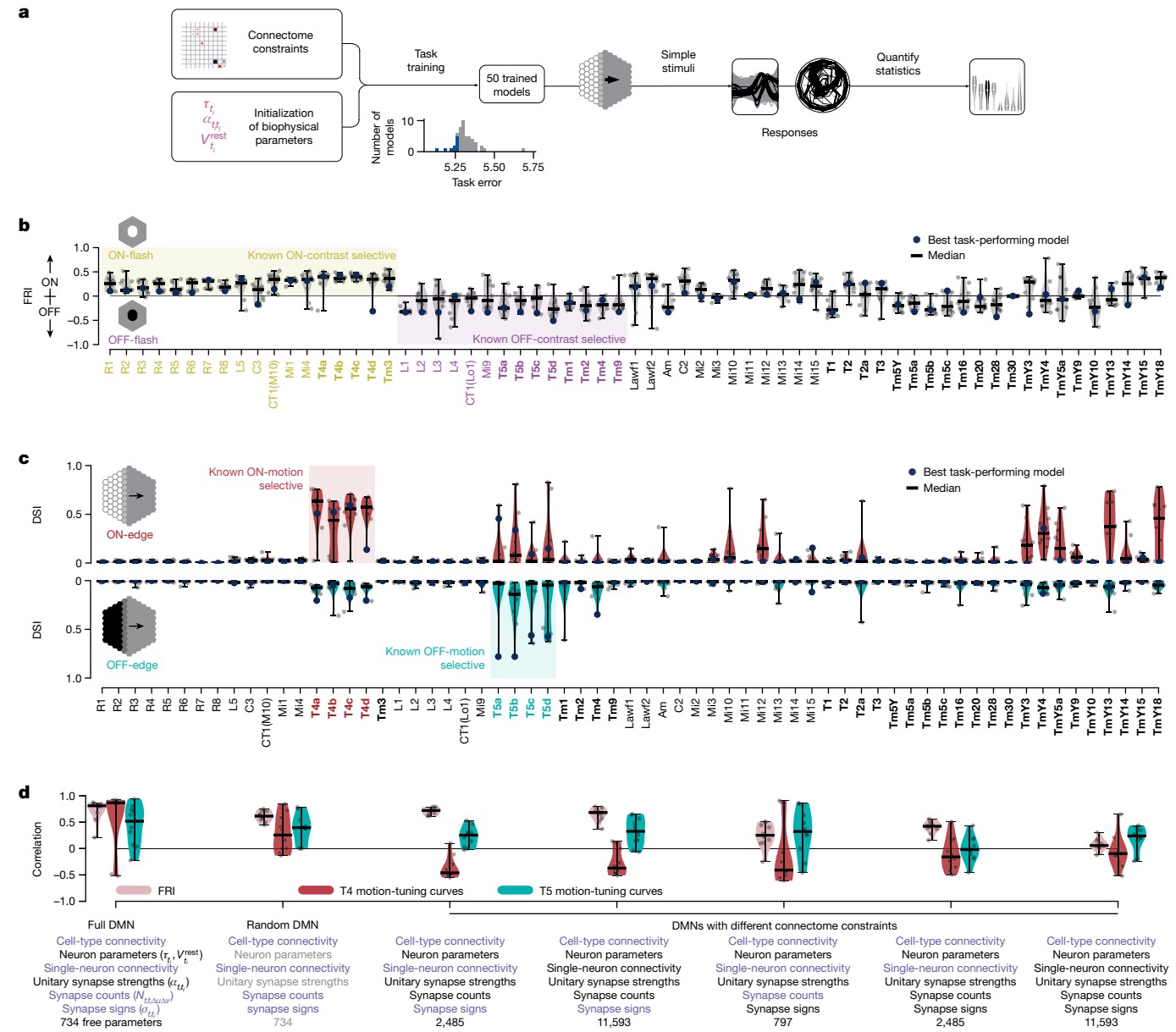

**Fig. 2 | Ensembles of DMNs predict tuning properties. a**, We task-optimized 50 connectome-constrained DMNs, yielding different solutions for the biophysical parameters (magenta), and compared the tuning properties of their cell types to experimental measurements. Inset: distribution of task errors. Blue: ten best models; also shown in **b**–**d**. **b**, ON- and OFF-contrast selectivity indices (FRI) for each cell type from ten models with best task performance (ten worst models in Extended Data Fig. 6). Yellow: cell types known to be ON-selective. Violet: known OFF-selective types. Black: selectivity not yet established experimentally. Bold: inputs to optic flow decoder.

**c**, Direction selectivity index (DSI) from neural responses to moving edges; same ten models as above. **d**, Correlations between measurements and predictions of neural activity from seven types of DMN with different connectome and parameter constraints: FRI (pink); motion-tuning curves for T4 (red) and T5 (turquoise); dashes indicate median correlation across models. The first DMN type on the left corresponds to the main DMNs analysed in **b,c** and all subsequent figures; the remaining six DMNs incorporate fewer constraints. Ten best task-performing models from each DMN type.

33 cell types, and consistency across the ensemble provides a measure of confidence in the predictions (Fig. 2b).

Second, a major result in fly visual neuroscience has been the identification of the T4 (ON) and T5 (OFF) neurons as the first direction-selective neurons with four subtypes (T4a, T4b, T4c and T4d, and T5a, T5b, T5c and T5d), each responding to motion in the four cardinal directions[37]. We characterized the motion selectivity of all 64 cell types by their responses to ON- and OFF-edges moving in 12 different directions. We found that the ensemble of models correctly predicts that T4 neurons are ON-motion selective, and T5 neurons are OFF-motion selective (Fig. 2c). The ensemble also correctly predicts

the lack of motion tuning in the input neurons to T4 and T5 motion detector neurons (Mi1, Tm3, Mi4, Mi9, Tm1, Tm2, Tm4, Tm9 and CT1; Methods and Supplementary Data file 7).

Our models also suggest the possibility that the transmedullary cell types, TmY3, TmY4, TmY5a, TmY13 and TmY18, might be tuned to ON-motion. Of these cell types, TmY3 neurons do not receive inputs from other known motion-selective neurons suggesting the possibility that these neurons constitute a parallel motion computation pathway from T4 and T5 neurons (Extended Data Figs. 3 and 4 and Supplementary Note 4). We questioned whether our model predicted motion selectivity for all cell types with asymmetric, multicolumnar

inputs, as this is a necessary connectivity motif for direction selectivity. On the basis of their local spatial connectivity profiles, we estimated that 19 cell types receive asymmetric, multicolumnar inputs (Methods and Extended Data Fig. 5b), but found that only 12 are predicted to be motion selective by the ensemble (Methods). Spatial offset of excitatory and/or inhibitory inputs did not correlate strongly with direction selectivity (Extended Data Fig. 5c,d). This suggests that our model integrates connectivity across the entire network with the task constraint to determine which neurons are most likely to be motion selective, rather than simply focusing on local connectivity.

## Connectome and task are both necessary

We investigated the importance of connectome constraints and task optimization to enable accurate predictions of neural activity. We found that both task optimization and detailed connectome constraints at the single-neuron resolution were critical to the prediction of the preferred contrast of the 32 characterized cell types, and the preferred direction of motion for the T4 and T5 subtypes (Fig. 2d and Extended Data Fig. 2).

We conducted 'ablation' studies comparing the DMN ensemble studied in this paper (full DMN) with a range of models with other modelling assumptions. First we verified the importance of task optimization by constructing an ensemble (random DMN) with random single-cell and synapse parameters, and full connectome constraints. This ensemble yielded accurate predictions of preferred contrast, but poor predictions of direction selectivity and preferred direction.

We then studied which aspects of the connectome must be measured accurately to lead to accurate predictions. We found that task-optimized models with access to only cell-type connectivity predicted neural activity poorly. We then considered several scenarios adding more connectome measurements beyond cell-type connectivity. We considered several scenarios: access to synapse signs and single-neuron connectivity but not strength, requiring task optimization of synapse counts; access to signs, but requiring optimization of single-neuron connectivity and synapse counts; full connectome measurements but optimization of synapse signs; access to connectivity but optimization of both synapse counts and signs. Across these modelling assumptions, we found that accurate predictions of contrast preference (FRI) were possible as long as measurements of the connection-signs were available, and that accurate predictions of the direction selectivity—but not preferred direction—could be achieved with measurements of cell connectivity, without the need for synapse count measurements (Extended Data Fig. 2c). This demonstrates the importance of both detailed connectome measurements and task optimization to achieve accurate predictions of neural activity.

Across the ensemble constrained by all connectome measurements and task training, we found that models that exhibited lower task error (Methods) also had more realistic tuning: models with higher task performance predict the direction selectivity index of T4 and T5 cells and their inputs better ($r = 0.60$, $P = 2.6 \times 10^{-6}$; Extended Data Figs. 2e and 6b). This suggests the possibility of using task error to rank models in terms of their likelihood to accurately predict neural activity.

Our model relies on an accurate classification of neurons into cell types to share single-neuron and synapse parameters across all neurons of the same cell type. We investigated the degree to which we could coarse-grain the cell-type categorization, leading to fewer cell types and fewer parameters (Extended Data Fig. 2a–d). We found that grouping the four T4 subtypes into a single T4 cell type, and grouping the four T5 subtypes into a single T5 cell type, had no negative impact on the quality of ensemble predictions. However, grouping all 37 excitatory cell types into a single E-type, 22 inhibitory cell types into a single I-type, and 4 mixed cell types into a single mixed type, led to poor performance on par with the random DMN.

## Predictions cluster across DMN ensemble

We sought to determine how similar or dissimilar the predictions of different models in an ensemble with the same connectome constraints and task optimization are. To address this for each cell type, we simulated neural activity in response to naturalistic video sequences from the Sintel dataset. We then used uniform manifold approximation and projection[38] to perform nonlinear dimensionality reduction on high-dimensional activity vectors of a representative neuron of each cell type across the model ensemble, and clustered the models in the resulting two-dimensional projections (Fig. 3a and Methods). For many cell types, we found that models predict strongly clustered neural responses (Supplementary Data file 7). For T4c neurons, for example, we found three clusters corresponding to qualitatively distinct responses of this cell type for naturalistic inputs: two clusters contain models with direction-selective T4c cells (Fig. 3a,b) with up- and down-selective cardinal tuning, respectively, whereas neurons in the third cluster are not direction-tuned. The direction-selective cluster with the (correct) upward preference has the lowest average task error (circular marker, average task error 5.297), followed by the cluster with the opposite preference (triangular marker, average task error 5.316). The non-selective cluster has the worst performance (square marker, average task error 5.357), suggesting that models with accurate tuning correlate with lower task error (see also Extended Data Fig. 2e).

We sought to determine what differences in circuit mechanisms underlie the different predictions for direction selectivity in the three clusters (Fig. 3c). Our results showed that direction selectivity in the two tuned clusters is associated with opposite preferred contrast tuning of Mi4 and Mi9 neurons, which provide direct flanking inhibitory input to T4 neurons (Fig. 3d). Models with the correct direction selectivity for T4 neurons also predict the correct contrast selectivity for Mi4 and M9 neurons, and vice versa (Fig. 3e).

Thus, the ensemble can be used to provide hypotheses about different circuit mechanisms that might underlie the response properties of individual cells. Furthermore, it shows that experimentally measuring the tuning of one neuron automatically translates to constraints on other neurons in the circuit. Here, filtering models in the ensemble with the experimentally measured direction selectivity for the T4c neurons (by only selecting models from the correct cluster) is sufficient to correctly constrain the tuning of both Mi4 and Mi9 neurons.

## Predicted mechanism of T4 and T5 tuning

Our DMN modelling approach enables a large number of model-based analyses, which can illuminate the mechanistic basis of computation in a circuit and suggest new visual stimuli for experimental characterization. We illustrate these analyses using averages from the model cluster with the best task performance (task-optimal cluster), focusing on the well-studied T4 and T5 neurons (Fig. 4). See Supplementary Data file 7 for a comprehensive set of analyses for all cell types and models. In the task-optimal cluster, the four subtypes of the T4 neurons respond strongly to bright (ON) edges, and the four subtypes of the T5 neurons to dark (OFF) edges, moving in the four cardinal directions, in agreement with experimental findings[27,37,39,40] (Fig. 4a). We probed the mechanism of direction selectivity in T4 and T5 neurons (Fig. 4b and Extended Data Figs. 7 and 8). Examining the input currents to a single T4 neuron (Extended Data Fig. 7a), we found that fast excitatory input and offset delayed inhibitory input currents enable T4 in the model to detect motion, in agreement with experimental findings[27]. The differential response of T4 neurons to motion in the preferred versus null direction is primarily produced by the differential timing of inhibition from Mi4. Additionally, excitatory T4-to-T4 currents between neurons with the same preferred direction lead to an increased response to coherent motion across the visual field. Although research into T4 motion selectivity has largely focused on the role of feedforward inputs,

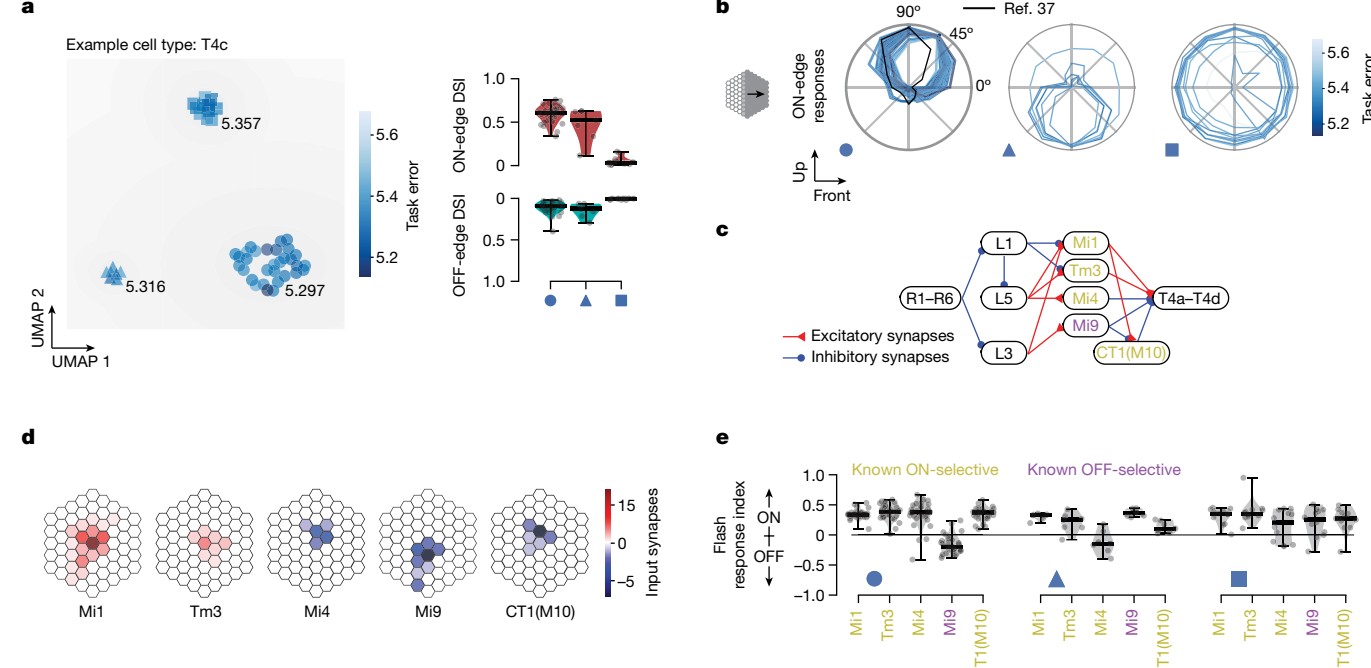

**Fig. 3 | Cluster analysis of DMN ensembles enables hypothesis generation and suggests experimental tests.** We clustered 50 DMNs after carrying out nonlinear dimensionality reduction of their responses to naturalistic scenes for each cell type, and aimed to identify whether clusters correspond to qualitatively different tuning mechanisms. **a**, Responses of T4c cells exhibit three clusters, two with ON-motion direction selectivity (circular and triangular marker) and one (square marker) without. **b**, T4c tuning in the three clusters. Circular marker: upwards tuning (cluster with lowest average task error 5.297; black: known tuning of T4c). Triangular marker: downwards (5.316 error). Square marker: no motion tuning (5.357 error). **c**, Schematic of corresponding ON-motion detection pathway. **d**, Connectivity of major input elements to T4c.

Blue and red colour: putative hyper- and depolarizing inputs. Saturation: average number of input synapses for each offset location. **e**, Tuning properties within each cluster reveal dependencies between T4 tuning and that of Mi4 and Mi9 cells in the ensemble: switching Mi4 (known ON-contrast selective) and Mi9 (known OFF-contrast selective) contrast preferences results in directionally opposite motion-tuning solutions in T4. DMNs in first cluster (T4c in DMN upwards tuned, circle) exhibit ON-selectivity for Mi1, Tm3, Mi4 and CT1(M10), and OFF-selectivity for Mi9. For ON-motion stimuli, in these DMNs T4c receives central depolarizing input from Mi1 and Tm3 and dorsal hyperpolarizing input from Mi4 and CT1(M10).

our modelling predicts an important role for the lateral connectivity between T4 neurons. The mechanisms in our model for T5 motion computation are similar (Extended Data Fig. 8a), with differential timing of inhibition from CT1, as well as excitation from Tm9, contributing to motion-selective responses.

To relate the mechanism of direction selectivity to the well-studied mechanisms of preferred direction enhancement and null direction suppression, we compared the responses of T4 and T5 neurons to moving bars and static bars as in ref. 27. Consistent with voltage measurements[27,40], voltage response predictions by our model show null direction suppression but no preferred direction enhancement (Extended Data Figs. 7b and 8b).

We computed and compared the spatial and temporal receptive fields of the major columnar input neurons to T4 and T5 neurons. These input neurons have been the focus of multiple experimental studies of the motion detection pathways[28,41–45] (Fig. 4d). In agreement with experimental findings[28,43], the prediction of the DMNs is that Tm3 and Tm4 have broad spatial receptive fields (two-column radius, 11.6°), whereas Mi1, Mi4, Mi9, Tm1, Tm2, Tm9 and CT1 compartments in both medulla and lobula have narrow spatial receptive fields (single-column radius, 5.8°).

We characterized the temporal response properties of cells in the motion pathways, including the lamina monopolar cells (L1–L5) and direct inputs to the T4 and T5 neurons. We simulated neural responses to single-ommatidium flashes of varying contrast and duration and compared them to empirically characterized temporal responses (Fig. 4e). The model accurately predicts the preferred contrast of each cell type[33] (that is, whether they depolarize more strongly to ON or OFF

single-ommatidium flashes; 5 ms to 300 ms duration; Methods). These cells either depolarize (which we call ON-selective) or hyperpolarize (which we call OFF-selective) in response to light-increment flashes. The temporal response properties are correctly predicted for all except the Tm4 cell in this model: for major T4 inputs, Mi1, Tm3 and Mi4 respond with transient depolarization to ON-flashes. By contrast, CT1(M10) responds with a longer sustained depolarization. Mi9 hyperpolarizes. For major T5 inputs, Tm1, Tm2, Tm9 and CT1(Lo1) respond with transient hyperpolarization. Tm4 is incorrectly predicted to depolarize. For lamina cell types, the DMNs predict biphasic hyperpolarization in L1 and L2 and monophasic hyperpolarization in L3 and L4, as well as depolarization in L5.

For motion-selective neurons such as T4 and T5, the spatio-temporal receptive fields are not separable in space and time. We characterized the full spatio-temporal receptive field for T4c and T5c neurons (Fig. 4f) using single-ommatidium ON- and OFF-flashes (20 ms; Methods). ON-flashes on the leading side of the receptive field of the ON-contrast, upwards-direction-selective T4c cell lead to fast depolarization, whereas ON-flashes on the trailing side lead to delayed hyperpolarization, again matching experimental findings[27]. As T5c is OFF-selective, its OFF-impulse responses are inverted, resembling the T4c spatio-temporal receptive field (Extended Data Fig. 9a). This reflects that T5c implements a similar motion-tuning mechanism to OFF-edges as T4c to ON-edges, in agreement with experimental findings[40].

Finally, we show that the model can be used to design optimized stimuli. We used the task-optimal model to screen for video sequences from the Sintel dataset that elicited the largest responses in the

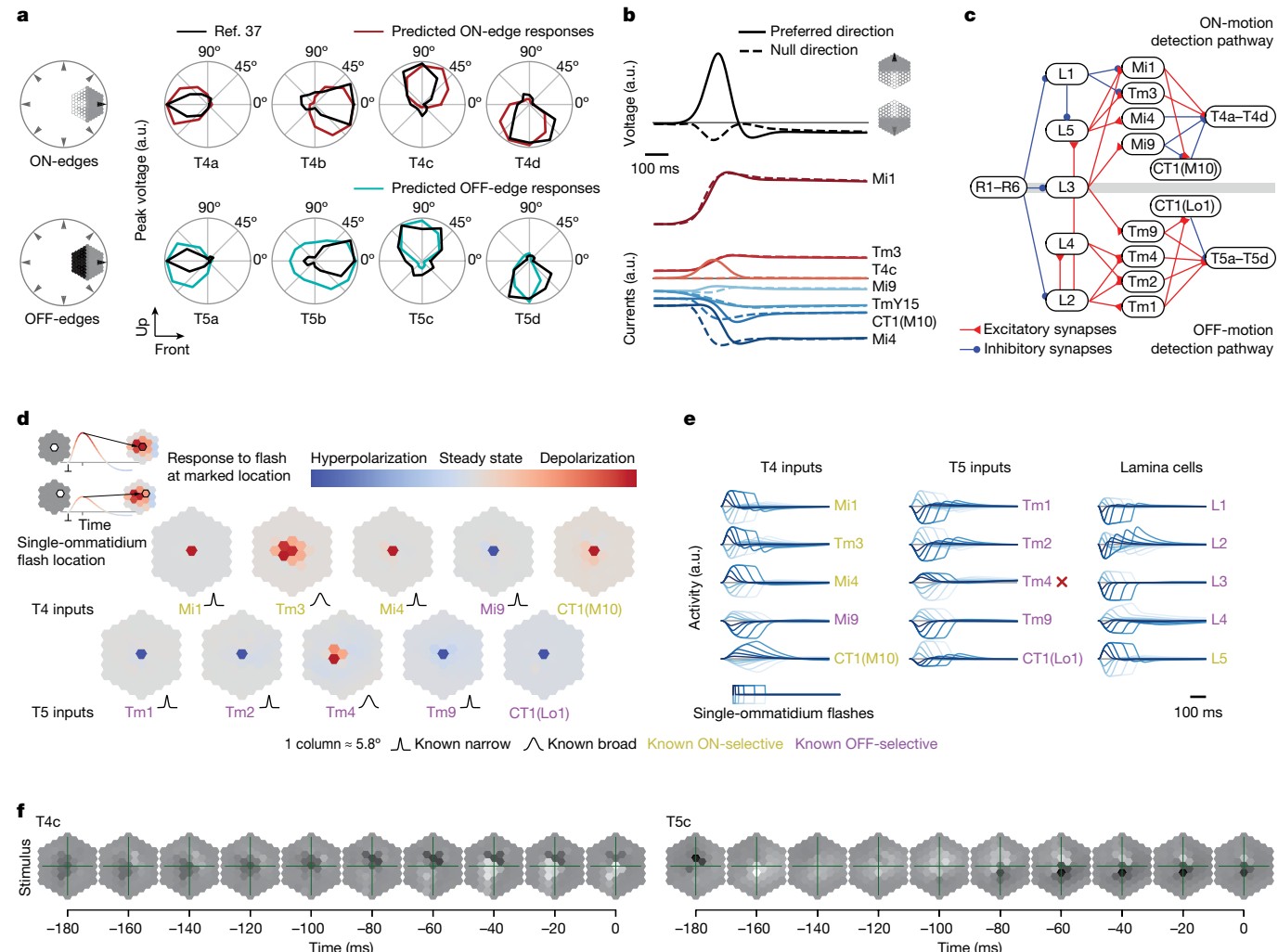

**Fig. 4 | Task-optimal DMNs largely recapitulate known mechanisms of motion computation. a**, Responses to moving edges for T4 and T5 subtypes from task-optimal model clusters, and comparison with experimental measurements[37,39] (null-contrasts in Supplementary Fig. 4). a.u., arbitrary units. **b**, Voltage of T4c neuron (top) and contributions from major input cells (bottom) during movement of an ON-edge across the visual field in preferred (solid) and null (dashed) direction. **c**, Major cell types and connectivity in the ON-motion (T4) and OFF-motion (T5) detection pathways (simplified). **d**, Spatial

receptive fields of major motion detector input neurons revealed by single-ommatidium flashes and comparison with experimental measurements[28,43]. **e**, Single-ommatidium flash responses agree with experimental measurements[41,43], with the exception of Tm4 (red cross). **f**, Stimulus sequence predicted to elicit the strongest responses in T4c and T5c cells. A central OFF-disc followed by an ON-edge moving upwards elicits the strongest response in a T4c cell (ON-disc followed by OFF-edge for T5c).

motion-selective neurons (Fig. 4f, Methods and Supplementary Data file 7 for all cell types). One might expect that pure ON- or OFF-stimuli would elicit the largest responses in T4 and T5, respectively. However, we found both ON- and OFF-elements in optimized stimuli, suggesting an interplay between ON- and OFF-pathways. The stimulus that elicited the strongest response in the T4c cell was a central OFF-disc followed by an ON-edge moving upwards, matching the preferred direction of the cell. Similarly, for the T5c cell, the stimulus that elicits the strongest response is a central ON-disc followed by an OFF-edge moving upwards in the preferred direction of the cell (Extended Data Fig. 9c for corresponding full-field naturalistic stimuli, numerically optimized stimuli and preferred moving-edge stimuli). Taken together, these findings show that the model predicts a large number of tuning properties for the T4 and T5 cells and their inputs.

## Sparsity leads to accurate predictions

Here we consider when connectome-constrained and task-optimized DMN models might accurately predict neural responses at single-neuron

resolution. Sparse connectivity is a hallmark of biological neural circuits. We questioned whether sparse connectivity enables DMNs to make accurate predictions of neural activity. For sparsely connected circuits—assuming the connectome is known—there are fewer synapse parameters left to estimate using task optimization. We reasoned that such networks might support fewer possible mechanisms by which to perform a given task, and so that a task-optimized DMN model is more likely to find the true mechanism and accurately predict single-neuron activity.

We tested this hypothesis in a simulation (Fig. 5), by constructing feedforward artificial neural networks solving the classic MNIST (Modified National Institute of Standards and Technology) handwritten digit classification task. These networks had varying degrees of sparse connectivity, and random assignment of neurons as excitatory and inhibitory respecting Dale's law (25 ground-truth networks for each sparsity level, Methods). We then simulated the process of making connectome measurements from these ground-truth networks, and building connectome-constrained task-optimized DMN simulations of each ground-truth network (Fig. 5a).

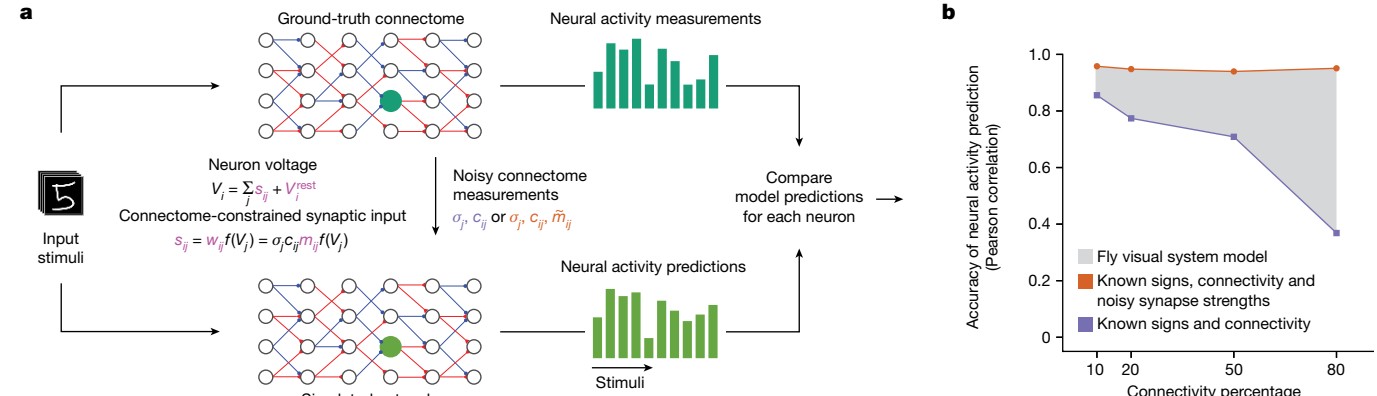

**Fig. 5 | Connectome measurements constrain neural networks in circuits with sparse connectivity. a**, We constructed synthetic 'ground-truth connectome' networks with varying degrees of sparse connectivity for classifying handwritten digits. For each ground-truth connectome network, we simulated connectome measurements and constructed a connectome-constrained and task-optimized the unknown parameters (magenta) in the 'simulated network' (Methods). We measured the correlation of the neural response vector, across all stimuli, between a ground-truth (dark green) and a simulated (light green) network. **b**, Median neural response correlation coefficients from 100 randomly sampled neuron pairs from each layer and across 25 network pairs. Two conditions were considered in which connectome measurements revealed either only binary connectivity (blue) or also connection strength (orange). The fly visual system model presented here probably falls in the region between the two curves, as measured synapse counts inform relative connection strengths between pairs of neurons for the same pair of cell types, but not absolute connection strength.

As the degree to which connection strength can be inferred from noisy connectome measurements is still unknown, we simulated two settings. In the first, we assumed that connectome measurements reveal connectivity but not connection strength. In this setting, DMNs were task-optimized to infer both the resting membrane potential of each neuron, and the connection strength of each connected pair of neurons. In the second, we assumed that the measurements additionally reveal a noisy estimate of strength, which was used as a soft constraint during task optimization.

Consistent with our hypothesis, our results showed that sparsity in the connectome greatly improves the accuracy of neural activity predictions with measurements of connectivity alone (Fig. 5b and Supplementary Fig. 8; median Pearson correlations of 0.85 for 10% connectivity versus 0.38 for 80% connectivity, 100 randomly selected neurons from 25 randomly generated ground-truth networks). However, with the additional availability of connection strength estimates, we find that DMN simulations accurately predict neural activity even in the absence of sparse connectivity (median Pearson correlation of >0.9 across all connectivities).

Our model of the fly visual system lies in an intermediate regime with regards to our knowledge of connection strength. We assumed that connectome measurements provided relative connection strength but not absolute connection strength, as we assumed that the unitary synaptic strength was unknown but the same for connections of the same cell-type pair. Thus, we attribute the success of our visual system model at predicting neural activity to both the sparse structure of connectivity in this circuit and also the estimates of connection strength from the synapse count.

## Discussion

We constructed a neural network with connectivity measured at the microscopic scale. We also required that, at the macroscopic scale, the collective neural activity dynamics across the entire network result in an ethologically relevant computation. This combination of microscopic and macroscopic constraints enabled us to constrain a large-scale computational model spanning many tens of cell types and tens of thousands of neurons. We showed that such large-scale mechanistic models could accurately make detailed predictions of the neural responses of individual neurons to dynamic visual stimuli, revealing the mechanisms by which computations are performed. Knowledge

of the connectome played a critical role in this success, in part by leading to a massive reduction in the number of free model parameters.

We have taken a reductionist modelling approach, simplifying the modelling of individual neurons and synapses, to focus on the role played by the connectivity of a neural network. We found that for the motion pathways of the fruit fly visual system, this model correctly predicts many aspects of visual selectivity. We considered only the role of this circuit in detecting motion, which is but one of many computations performed by the visual system[24]. Our reductionist model cannot, for example, account for the role played in this circuit by electrical synapses[46], nonlinear chemical synapses[47] and neuromodulation[48]. However, richer models of neurons, synapses, plasticity and extrasynaptic modulation, along with a broader range of ethologically relevant tasks, can enable accurate modelling of these and other effects in the fly visual system and beyond.

Task-optimized artificial neural network models, for instance of mammalian visual pathways[22], have previously demonstrated only a coarse correspondence of the population neural activity between model layers and brain regions. By contrast, every neuron and synapse in our connectome-constrained model[49–53] has a direct correspondence to neurons and synapses in the brain. This correspondence enables highly detailed experimentally testable predictions at the single-neuron resolution. Thus, our study more directly links artificial neural network models to the biological neural network.

Our modelling approach provides a discovery tool, aimed at using connectome measurements to generate detailed, experimentally testable hypotheses for the computational role of individual neurons. Measurements of neural activity are necessarily sparse and involve difficult trade-offs. Activity can be measured in limited contexts, and either for a limited number of neurons or for a larger number of neurons with poorer temporal resolution. Connectome-constrained DMN models generate meaningful predictions even in the complete absence of neural activity measurements, but can be further constrained by sparse measurements of neural activity as we showed (Fig. 3), or indeed directly fitted to measured neural activity[51] and behaviour[54].

Whole-brain connectome projects have just been completed for the larval and adult fruit fly[7,8,55,56], including two new connectomes of the entire fruit fly optic lobe[6,9], and whole-mouse-brain connectome projects are now being discussed[57]. Large-scale whole-nervous-system models[51,52,58] will be of critical importance for integrating connectomic,

transcriptomic, neural activity and animal behaviour measurements across laboratories, scales and the nervous system[13]. Furthermore, with the recent development of detailed biomechanical body models for the fruit fly[59,60], we can now contemplate constructing whole-animal models spanning brain and body.

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

# Methods

## Construction of spatially invariant connectome from local reconstructions

We built a computational model of the fly visual system that is consistent with available connectome data[1–5], has biophysically plausible neural dynamics, and can be computationally trained to solve an ethologically relevant behavioural task, namely the estimation of optic flow. To achieve this, we developed algorithms to blend annotations from two separate datasets by transforming, sanitizing, combining and pruning the raw datasets into a coherent connectome spanning all neuropils of the optic lobe (Supplementary Note 1 and Supplementary Data files 1–3).

The original data stem from focused ion beam scanning electron microscopy datasets from the FlyEM project at Janelia Research Campus. The FIB-25 dataset volume comprises seven medulla columns and the FIB-19 dataset volume comprises the entire optic lobe and, in particular, detailed connectivity information for inputs to both the T4 and T5 pathways[2–4]. The data available to us consisted of 1,801 neurons, 702 neurons from FIB-25 and 1,099 neurons from FIB-19. For about 830 neurons, the visual column was known from hand annotation. These served as reference positions. Of the 830 reference positions, 722 belong to neuron types selected for simulation. None of the T5 cells, whose directional selectivity we aimed to elucidate, was annotated. We therefore built an automated, probabilistic expectation maximization algorithm that takes synaptic connection statistics, projected synapse centre-of-mass clusters and existing column annotations into account. We verified the quality of our reconstruction as described in Supplementary Note 1. Only the neurons consistently annotated with both 100% and 90% of reference positions used were counted to estimate the number of synapses between cell types and columns, to prune neuron offsets with low confidences.

Synaptic signs for most cell types were predicted on the basis of known expression of neurotransmitter markers (primarily the cell-type-specific transcriptomics data from ref. 30). For a minority of cell types included in the model, no experimental data on transmitter phenotypes were available. For these neurons, we used guesses of plausible transmitter phenotypes. To derive predicted synaptic signs from transmitter phenotypes, we assigned the output of histaminergic, GABAergic and glutamatergic neurons as hyperpolarizing and the output of cholinergic neurons as depolarizing. In a few cases, we further modified these predictions on the basis of distinct known patterns of neurotransmitter receptor expression (see ref. 30 for details). For example, output from R8 photoreceptor neurons, predicted to release both acetylcholine and histamine, was treated as hyperpolarizing or depolarizing, respectively, depending on whether a target cell type is known to express the histamine receptor gene *ort* (which encodes a histamine-gated chloride channel).

## Representing the model as a hexagonal convolutional neural network

Our end-to-end differentiable[61] DMN model of the fly visual system can be interpreted as a continuous-time neural ordinary differential equation[62] with a deep convolutional recurrent neural network[63] architecture that is trained to perform a computer vision task using backpropagation through time[64,65]. Our goal was to optimize a simulation of the fly visual system to perform a complex visual information processing task using optimization methods from deep learning. One hallmark of visual systems that has been widely exploited in such tasks is their convolutional nature[66–69] (that is, the fact that the same computations are applied to each pixel of the visual input). To model the hexagonal arrangement of photoreceptors in the fly retina, we developed a hexagonal convolutional neural network (CNN) in the widely used deep learning framework PyTorch[21] (ignoring neuronal

superposition[70]), which we used for simulation and optimization of the model. We model columnar cell types, including retinal cells, lamina monopolar and wide-field cells, medulla intrinsic cells, transmedullary cells and T-shaped cells, as well as amacrine cells. The model comprises synapses from all neuropils and downstream- and upstream-projecting connections from the retina, lamina and medulla.

## Neuronal dynamics

In detail, we simulated point neurons with voltages $V_i$ of a postsynaptic neuron $i$, belonging to cell type $t_i$ using threshold-linear dynamics, mathematically equivalent to commonly used formulations of firing-rate models[71]

$$\tau_{t_i}\dot{V}_i = -V_i + \sum_j s_{ij} + V_{t_i}^{\text{rest}} + e_i. \tag{1}$$

Neurons of the same cell type share time constants, $\tau_{t_i}$, and resting potentials, $V_{t_i}^{\text{rest}}$. Dynamic visual stimuli were delivered as external input currents $e_i$ to the photoreceptor (R1–R8), for all other cell types, $e_i = 0$. In our model, instantaneous graded synaptic release from presynaptic neuron $j$ to postsynaptic neuron $i$ is described by

$$s_{ij} = w_{ij}f(V_j) = \alpha_{t_it_j}\sigma_{t_it_j}N_{t_it_j\Delta u\Delta v}f(V_j), \tag{2}$$

comprising the anatomical filters in terms of the synapse count from electron microscopy reconstruction, $N_{t_it_j\Delta u\Delta v}$, at the offset location $\Delta u = u_i - u_j$ and $\Delta v = v_i - v_j$ in the hexagonal lattice between two types of cells, $t_i$ and $t_j$, and further characterized by a sign, $\sigma_{t_it_j} \in \{-1, +1\}$, and a non-negative scaling factor, $\alpha_{t_it_j}$.

The synapse model entails a trainable non-negative scaling factor per filter that is initialized as

$$\alpha_{t_i,t_j} = \frac{0.01}{\langle N_{t_i,t_j}\rangle_{\Delta u,\Delta v}},$$

with the denominator describing the average synapse count of the filter. Synapse counts, $N_{t_it_j\Delta u\Delta v}$ from the connectome, and signs, $\sigma_{t_it_j}$ from the neurotransmitter and receptor profiling, were kept fixed. The scaling factor was clamped during training to remain non-negative.

Moreover, at initialization, the resting potentials were sampled from a Gaussian distribution

$$V_{t_i}^{\text{rest}} \sim \mathcal{N}(\mu_{V^{\text{rest}}}, \sigma^2_{V^{\text{rest}}})$$

with mean $\mu_{V^{\text{rest}}} = 0.5$ (a.u.) and variance $\sigma^2_{V^{\text{rest}}} = 0.05$ (a.u.). The time constants were initialized at $\tau_{t_i} = 50$ ms. The 50 task-optimized DMNs were initialized with the same parameter values. During training, in Euler integration of the dynamics, we clamped the time constants as $\tau_i = \max(\tau_i, \Delta t)$, so that they remain above the integration time step $\Delta t$ at all times.

In total, the model comprises 45,669 neurons and 1,513,231 synapses, across two-dimensional (2D) hexagonal arrays 31 columns across. The number of free parameters is independent of the number of columns: 65 resting potentials, 65 membrane time constants, 604 scaling factors; and connectome-determined parameters: 604 signs and 2,355 synapse counts. Thus, the number of free parameters in the visual system model is 734.

In the absence of connectome measurements, the number of parameters to be estimated is much larger. With $T = 65$ cell types (counting CT1 twice for the compartments in the medulla and lobula) and $C = 721$ cells per type for simplicity, the number of cells in our model would be $TC = 46,865$. Assuming a recurrent neural network with completely unconstrained connectivity and simple dynamics $\tau_i\dot{V}_i = -V_i + \sum_j w_{ij}f(V_j) + V_i^{\text{rest}}$, we would have to find $(TC)^2 + 2(TC) = 2,196,421,955$ free parameters. Assuming a convolutional recurrent

neural network with shared filters between cells of the same post-synaptic type, shared time constants and shared resting potentials, the amount of parameters reduces markedly to $T^2C + 2T = 3,046,355$. Further assuming the same convolutional recurrent neural network but additionally that convolutional filters are constrained to $F = 5$ visual columns (that is, the number of presynaptic input columns in the hexagonal lattice is $P = 3F(F+1)+1$), the amount of parameters reduces to $T^2P + 2T = 384,605$. Assuming as in our connectome that only $Q = 604$ connections between cell types exist, this reduces the number of parameters further to $QP + 2T = 55,185$. Instead of parametrizing each individual synapse strength, we assume that synapse strength is proportional to synapse count from the connectome times a scalar for each filter, reducing the number of parameters to $Q + 2T = 734$ while providing enough capacity for the DMNs to yield realistic tuning to solve the task.

**Convolutions using scatter and gather operations.** For training the network, we compiled the convolutional architecture specified by the connectome and the sign constraints to a graph representation containing: a collection of parameter buffers shared across neurons and/or connections; a collection of corresponding index buffers indicating where the parameters relevant to a given neuron or connection can be found in the parameter buffers; and a list of pairs (presynaptic neuron index, postsynaptic neuron index) denoting connectivity. This allowed us to efficiently simulate the network dynamics through Euler integration using a small number of element-wise, scatter and gather operations at each time step. We found that this is more efficient than using a single convolution operation or performing a separate convolution for each cell type as each cell type has its own receptive field—some much larger than others—and the number of cells per type is relatively small.

## Optic flow task

**Model training.** An optic flow field for a video sequence consists of a 2D vector field for each frame. The 2D vector at each pixel represents the magnitude and direction of the apparent local movement of the brightness pattern in an image.

We frame the training objective as a regression task

$$\hat{\mathbf{Y}}[n] = \text{Decoder}(\text{DMN}(\mathbf{X}[0], \ldots, \mathbf{X}[n])),$$

with $\hat{\mathbf{Y}}$ being the optic flow prediction, and $\mathbf{X}$ being the visual stimulus sequence from the Sintel dataset, both sampled to a regular hexagonal lattice of 721 columns. With the objective to minimize the square error loss between predicted optic flow and target optic flow fields, we jointly optimized the parameters of both the decoder and the visual system network model described above.

In detail, for training the network, we added randomly augmented, greyscaled video sequences from the Sintel dataset sampled to a regular hexagonal lattice of 721 columns to the voltage of the 8 photoreceptor cell types (Fig. 1f and equation (1)). We denote a sample from a minibatch of video sequences as $\mathbf{X} \in \mathbb{R}^{N,C}$, with $N$ being the number of time steps, and $C$ being the number of photoreceptor columns. The dynamic range of the input lies between 0 and 1. Input sequences during training entailed 19 consecutive frames drawn randomly from the dataset and resampled to match the integration rate. At the original frame rate of 24 Hz, this corresponds to a simulation of 792 ms. We did not find that an integration time step smaller than 20 ms (that is, a frame rate of 50 Hz after resampling) yielded qualitatively superior task performance or more realistic tuning predictions. We interpolated the target optic flow in time to 50 Hz temporal resolution. To increase the amount of training data for better generalization, we augmented input and target sequences as described further below. At the start of each epoch, we computed an initial state of the network's voltages after 500 ms of grey stimulus presentation to initialize the network at a steady state for each minibatch during that epoch. The network

integration for a given input $\mathbf{X}$ results in simulated sequences of voltages $\mathbf{V} \in \mathbb{R}^{N,T_C}$, with $T_C$ being the total number of cells. The subset of voltages, $\mathbf{V}_{\text{out}} \in \mathbb{R}^{N,D,C}$, of the $D$ cell types in the black rectangle in Fig. 1g was passed to a decoding network. For decoding, the voltage was rectified to avoid the network finding biologically implausible solutions by encoding in negative dynamic ranges. Furthermore, it was mapped to Cartesian coordinates to apply PyTorch's standard spatial convolution layers for decoding and on each time step independently. In the decoding network, one layer implementing spatial convolution, batch normalization, softplus activation and dropout, followed by one layer of spatial convolution, transforms the $D$ feature maps into the 2D representation of the estimated optic flow, $\hat{\mathbf{Y}} \in \mathbb{R}^{N,2,C}$.

Using stochastic gradient descent with adaptive moment estimation ($\beta_1 = 0.9$, $\beta_2 = 0.999$, learning rate decreased from $5 \times 10^{-5}$ to $5 \times 10^{-6}$ in ten steps over iterations, batch size of four) and the automatic gradient calculation of the fully differentiable pipeline, we optimized the biophysical parameters through backpropagation through time such that they minimize the L2-norm between the predicted optic flow, $\hat{\mathbf{Y}}$, and the ground-truth optic flow, $\mathbf{Y}$:

$$L(\mathbf{Y}, \hat{\mathbf{Y}}) = \|\mathbf{Y} - \hat{\mathbf{Y}}\|.$$

We additionally optimized the shared resting potentials for 150,000 iterations, using stochastic gradient descent without momentum, with respect to a regularization function of the time-averaged responses to naturalistic stimuli of the central column cell of each cell type, $t_{\text{central}}$, to encourage configurations of resting potentials that lead to non-zero and non-exploding activity in all neurons in the network. We weighted these terms independently with $\gamma = 1$, encouraging activity greater than $a$, and $\delta = 0.01$, encouraging activity less than $a$. We chose $\lambda_V = 0.1$ and $a = 5$ in arbitrary units. With $B$ being the batch size and $T$ being the number of all cell types, the regularizer is

$$R(V) = \frac{\lambda_V}{BT} \sum_b \sum_{t_{\text{central}}} \begin{cases} \gamma(\overline{V} - a)^2, & \text{if } \overline{V} = \frac{1}{N}\sum_n V_{bt_{\text{central}}}[n] \le a \\ \delta(\overline{V} - a)^2, & \text{if } \overline{V} > a. \end{cases}$$

We regularly checkpointed the error measure $L(\mathbf{Y}, \hat{\mathbf{Y}})$ averaged across a held-out validation set of Sintel video clips. Models generalized on optic flow computation after about 250,000 iterations, yielding functional candidates for our fruit fly visual system models that we analysed with respect to their tuning properties.

**Task-optimization dataset.** We optimized the network on 23 sequences from the publicly available computer-animated film Sintel[12]. The sequences have 20–50 frames, at a frame rate of 24 frames per second and a pixel resolution of 1,024 × 436. The dataset provides optical flow in pixel space for each frame after the first of each sequence. As the integration time steps we use are faster than the actual sampling rate of the sequences, we resample input frames accordingly over time and interpolate the optic flow.

**Fly-eye rendering.** We first transformed the RGB pixel values of the visual stimulus to normalized greyscale between 0 and 1. We translated Cartesian frames into receptor activations by placing simulated photoreceptors in a 2D hexagonal array in pixel space, 31 columns across resulting in 721 columns in total, spaced 13 pixels apart. The transduced luminance at each photoreceptor is the greyscale mean value in the 13 × 13-pixel region surrounding it.

**Augmentation.** We used: random flips of input and target across one of the three principal axes of the hexagonal lattice; random rotation of input and target around its six-fold rotation axis; adding element-wise Gaussian noise with mean zero and variance $\sigma_n = 0.08$ to the input $X$ (then clamped at 0); random adjustments of contrasts,

$\log c \sim \mathcal{N}(0, \sigma_c^2 = 0.04)$ and brightness, $b \sim \mathcal{N}(0, \sigma_b^2 = 0.01)$, of the input with $X' = c(X - 0.5) + 0.5 + cb$.

In addition, we 'strided' the fly-eye rendering across the rectangular raw frames in width, subsampling multiple scenes from each. We ensured that such subsamples from the same scene were not distributed across training and validation sets. Input sequences in chunks of 19 consecutive frames were drawn randomly in time from the full sequences.

**Black-box decoding network.** The decoding network is feedforward, convolutional and has no temporal structure. Aspects of the architecture are explained in the section entitled Model training. The spatial convolutions have a filter size of $5 \times 5$. The first layer transforms the $D = 34$ feature maps to an eight-channel intermediate representation, which is further translated by an additional convolutional layer to a three-channel intermediate representation of optic flow. The third channel is used as shared normalization of each coordinate of the remaining 2D flow prediction. The decoder uses PyTorch-native implementations for 2D convolutions, batch normalization, softplus activation and dropout. We initialized its filter weights homogeneously at 0.001.

## Model characterization

**Task error.** To rank models on the basis of their task performance, we computed the standard optic flow metric of average end-to-end point error (EPE)[72], which calculates the average over all time steps and pixels (that is, here columns) of the error

$$\text{EPE}(\mathbf{Y}, \hat{\mathbf{Y}}) = \frac{1}{NC} \sum_n \sum_c \sqrt{(y_{1c}[n] - \hat{y}_{1c}[n])^2 + (y_{2c}[n] - \hat{y}_{2c}[n])^2}$$

between predicted optic flow and ground-truth optic flow and averaged across the held-out validation set of Sintel sequences.

**Importance of task optimization and connectome constraints.** We generated DMNs with different constraints to assess their relative importance for predicting tuning properties. First, we studied the importance of task optimization of DMN parameters. We created 50 DMNs with random Gaussian-distributed parameters, and task-optimized only their decoding network, to obtain baseline values for both the task error and the accuracy of predicting tuning curves without task optimization of the DMN.

In the full DMN, we constrained single synapses by connectome cell-type connectivity, cell connectivity, synapse counts and synapse signs (equation (2)) and task-optimized the non-negative type-to-type unitary synapse scaling factor $\boldsymbol{\alpha}_{t_i, t_j}$. Next, we trained five additional task-optimized DMNs with different connectome constraints (Fig. 2d and Extended Data Fig. 2a–d).

In these five additional types of DMN, we additionally task-optimized the terms in bold, rather than using connectome measurements, related to synaptic currents from a presynaptic cell $j$ to a postsynaptic cell $i$: known single-cell connectivity, unknown synapse count: $w_{ij} = \sigma_{t_i, t_j} \mathbf{m}_{t_i, t_j, \Delta u, \Delta v}$, in which $\mathbf{m}_{t_i, t_j, \Delta u, \Delta v}$ is non-negative; known cell-type connectivity, unknown single-cell connectivity and synapse counts: $w_{i,j} = \sigma_{t_i, t_j} \mathbf{m}_{t_i, t_j, -3 < \Delta u, \Delta v, \Delta u + \Delta v < 3}$ (that is, for all connected cell types, a connection weight was learned for all cells up to a distance of three columns in hexagonal coordinates, with known signs); known single-cell connectivity and synapse counts, but unknown synapse signs: $w_{i,j} = \boldsymbol{\alpha}_{t_i, t_j} \boldsymbol{\sigma}_{t_j} N_{t_i, t_j, \Delta u, \Delta v}$ (that is, connection weights were fixed by measurements, but signs optimized); known single-cell connectivity, but unknown synapse signs and synapse counts: $w_{i,j} = \mathbf{w}_{t_i, t_j, \Delta u, \Delta v}$ (that is, all non-zero connection weights were optimized, including their signs); or known cell-type connectivity, unknown single-cell connectivity, synapse counts and synapse signs: $w_{i,j} = \mathbf{w}_{t_i, t_j, -3 < \Delta u, \Delta v, \Delta u + \Delta v < 3}$ (that is, for all connected cell types, a connection weight and sign was learned for all cells up to distance of three columns). We trained 50 models per

DMN type. The task-optimized parameters in each case are highlighted using bold symbols. We randomly initialized the models with $\mathbf{m}_{t_i, t_j}, \mathbf{w}_{t_i, t_j} \sim \mathcal{N}\left(0, \frac{2}{n_{\text{in}}}\right)$, in which $n_{\text{in}}$ is the number of cell connections and $\mathbf{m}$ is non-negative, and $\boldsymbol{\sigma}_{t_j} \in \{-1, 1\}$ with equal probability.

**Unconstrained CNN.** We trained unconstrained, fully convolutional neural networks on the same dataset and task to provide an estimate of a lower bound for the task error of the DMN. Optic flow was predicted by the CNN from two consecutive frames

$$\hat{Y}[n] = \text{CNN}(X[n], X[n-1]).$$

with the original frame rate of the Sintel film. We chose 5 layers for the CNN with 32, 92, 136, 8 and 2 channels, respectively, and kernel size 5 for all but the first layer, for which the kernel size is 1. Each layer performs a convolution, batch normalization and exponential linear unit activation, except the last layer, which performs only a convolution. We optimized an ensemble of 5 unconstrained CNNs with 414,666 free parameters each using the same loss function, $L(Y, \hat{Y})$, as for the DMN. We used the same dataset (that is, hexagonal sequences and augmentations from Sintel) for training and validating the CNN as that used for training and validating the DMN, enabled by two custom modules mapping from the hexagonal lattice to a Cartesian map and back.

**Circular flash stimuli.** To evaluate the contrast selectivity of cell types in task-optimized models, we simulated responses of each DMN to circular flashes. The networks were initialized at an approximate steady state after 1 s of grey-screen stimulation. Afterwards the flashes were presented for 1 s. The flashes with a radius of 6 columns were ON (intensity $I = 1$) or OFF ($I = 0$) on a grey ($I = 0.5$) background. We integrated the network dynamics with an integration time step of 5 ms. We recorded the responses of the modelled cells in the central columns to compute the FRI.

**FRI.** To derive the contrast selectivity of a cell type, $t_i$, we computed the FRI as

$$\text{FRI}_{t_i} = \frac{r_{t_{\text{central}}}^{\text{peak}}(I=1) - r_{t_{\text{central}}}^{\text{peak}}(I=0)}{r_{t_{\text{central}}}^{\text{peak}}(I=1) + r_{t_{\text{central}}}^{\text{peak}}(I=0)}$$

from the non-negative activity

$$r_{t_{\text{central}}}^{\text{peak}}(I) = \max_n V_{t_{\text{central}}}[n](I) + |\min_{n,I} V_{t_{\text{central}}}[n](I)|,$$

from voltage responses $V_{t_{\text{central}}}[n](I)$ to circular flash stimuli of intensities $I \in \{0, 1\}$ lasting for 1 s after 1 s of grey stimulus. We note that our index quantifies whether the cell depolarizes to ON- or to OFF-stimuli. However, cells such as R1–R8, L1 and L2 can be unrectified (that is, sensitive to both light increments and light decrements), which is not captured by our index.

For the $P$ values reported in the results, we carried out a binomial test with probability of correct prediction 0.5 (H0) or greater (H1) to test whether both the median FRI from the DMN ensemble and the task-optimal model can predict the contrast preferences. Additionally, we found for each individual cell type across 50 DMNs that predictions for 29 out of 31 cell types are significant ($P < 0.05$, binomial).

**Moving-edge stimuli.** To predict the motion sensitivity of each cell type in task-constrained DMNs, we simulated the response of each network, initialized at an approximate steady state after 1 s of grey-screen stimulation, to custom generated edges moving to 12 different directions, $\theta \in [0°, 30°, 60°, 90°, 120°, 150°, 180°, 210°, 240°, 270°, 300°, 330°]$. We integrated the network dynamics with an integration time step of 5 ms. ON-edges ($I = 1$) or OFF-edges ($I = 0$) moved on a grey ($I = 0.5$) background. Their movement ranged from $-13.5°$ to $13.5°$

visual angle and we moved them at six different speeds, ranging from $13.92°\,s^{-1}$ to $145°\,s^{-1}$ ($S \in [13.92°\,s^{-1}, 27.84°\,s^{-1}, 56.26°\,s^{-1}, 75.4°\,s^{-1}, 110.2°\,s^{-1}, 145.0°\,s^{-1}]$). In Fig. 2d, we report the correlation between predicted motion-tuning curves to the single experimentally measured tuning curve. We take the maximum correlation across six investigated speeds to make the correlation measure robust to potential variations in preferred speeds.

**DSI.** We computed a DSI[73] of a particular type $t_i$ as

$$\mathrm{DSI}_{t_i}(I) = \frac{1}{|\mathbb{S}|} \sum_{S \in \mathbb{S}} \frac{|\sum_{\theta \in \Theta} r^{\mathrm{peak}}_{t_{\mathrm{central}}}(I, S, \theta) \exp(i\theta)|}{\max\limits_{I \in \mathbb{I}} |\sum_{\theta} r^{\mathrm{peak}}_{t_{\mathrm{central}}}(I, S, \theta)|}$$

from rectified peak voltages

$$r^{\mathrm{peak}}_{t_{\mathrm{central}}}(I, S, \theta) = \max_n V^+_{t_{\mathrm{central}}}[n](I, S, \theta),$$

elicited from moving-edge stimuli. We rectify the voltage to quantify the tuning of the effective output of the cell, and to avoid the denominator becoming zero. We parameterized movement angle $\theta \in \Theta$, intensities $I \in \mathbb{I}$, and speeds $S \in \mathbb{S}$ of the moving edges. To take the response magnitudes into account for comparing the DSI for ON- and for OFF-edges, we normalized by the maximum over both intensities in the denominator. To take different speeds into account, we averaged over $\mathbb{S}$.

**Normalization of model neural activity for averaging across models in a cluster.** Threshold-linear networks have arbitrary units for the voltages and currents. Therefore, we normalized the neural activity before averaging the neural activity predictions from different models. For a single cell or cell type $t$, we first divided responses (voltages or rectified voltages) by the root mean square across the cell's responses to the naturalistic stimuli:

$$r^{\|\cdot\|\sim 1}_t[n] = \frac{r_t[n]}{\left\|\frac{1}{SN}\mathbf{R}^{\mathrm{nat\cdot}}_t\right\|_2},$$

in which $\mathbf{R}^{\mathrm{nat\cdot}}_t \in \mathbb{R}^{S,N}$ is the cell's response vector to $S$ sequences from the Sintel dataset with $N$ time steps and $r_t[n]$ is the cell's response to any stimuli. This normalization makes averages (Fig. 4a,b,d–e and Extended Data Figs. 4, 7, 8, and 9a,b) independent to variation in the scale of neural activity from model to model. We normalized input currents equivalently (Fig. 4b and Extended Data Figs. 4, 7, and 8) by the same normalization factor. We exclude solutions for which the denominator becomes zero.

**Determining whether a cell type with asymmetric inputs counts as direction selective.** We counted a cell type as direction selective if the DSIs from its synthetic measurements were larger than 99% of DSIs from non-motion selective cell types (that is, those with symmetric filters). We note, however, that estimates of the spatial asymmetry of connectivity from existing connectome reconstructions can be noisy.

For deriving the 99% threshold, we first defined a distribution $p(d^*|t_{\mathrm{sym}})$ over the DSI for non-direction-selective cells, from peak responses to moving edges of cell types with symmetric inputs, $t_{\mathrm{sym}}$. We computed that distribution numerically by sampling

$$d^* = \frac{|\sum_{\theta^*} r^{\mathrm{peak}}_{t_{\mathrm{central}}}(I, S, \theta^*) \exp i\theta|}{|\sum_{\theta} r^{\mathrm{peak}}_{t_{\mathrm{central}}}(I, S, \theta)|}$$

for 100 independent permutations of the angle $\theta^*$. We independently computed $d^*$ for all stimulus conditions, models and cell types with

symmetric inputs. From $p(d^*|t_{\mathrm{sym}})$, we derived the threshold $d_{\mathrm{thresh}} = 0.357$ as the 99% quantile of the random variable $d^*$, meaning that the probability that a realization of $d^* > d_{\mathrm{thresh}}$ is less than 1% for cell types with symmetric inputs. To determine whether an asymmetric cell type counts as direction selective, we tested whether synthetically measuring direction selectivity larger than $d_{\mathrm{thresh}}$ in that cell type is binomial with probability 0.1 (H0) or greater (H1). We identified 12 cell types with asymmetric inputs (T4a, T4b, T4c, T4d, T5a, T5b, T5c, T5d, TmY3, TmY4, TmY5a and TmY18) as direction selective ($P < 0.05$) from our models, and 7 cell types with asymmetric inputs to not count as direction selective (T2, T2a, T3, Tm3, Tm4, TmY14 and TmY15; see Extended Data Fig. 5 as reference for cell types with symmetric and asymmetric inputs).

**Uniform manifold approximation and projection and clustering.** We first simulated central column responses to naturalistic scenes (24 Hz Sintel video clips from the full augmented dataset) with an integration time step of 10 ms. We clustered models in feature space of concatenated central column responses and sample dimension. Next, we computed a nonlinear dimensionality reduction to two dimensions using the UMAP (uniform manifold approximation and projection) algorithm, and fitted Gaussian mixtures of 2 to 5 components, with the number of components that minimize the Bayesian information criterion, using the Python libraries umap-learn and scikit-learn[38,74].

**Single-ommatidium flashes.** To derive spatio-temporal receptive fields of cells, we simulated the response of each network to single-ommatidium flashes. Flashes were ON ($I = 1$) or OFF ($I = 0$) on a grey ($I = 0.5$) background and presented for [5, 20, 50, 100, 200, 300] ms after 2 s of grey-screen stimulation and followed by 5 s of grey-screen stimulation.

**Spatio-temporal, spatial and temporal receptive fields.** We derived the spatio-temporal receptive field (STRF) of a cell type $t_i$ as the baseline-subtracted responses of the central column cell to single-ommatidium flashes $J(u, v)$ at ommatidium locations $(u, v)$:

$$\mathrm{STRF}_{t_{\mathrm{central}}}[n](u, v) = V_{t_{\mathrm{central}}}[n](J(u, v)) - V_{t_{\mathrm{central}}}[n = 0](J(u, v)).$$

We derived spatial receptive fields (SRFs) from the responses to flashes (20 ms in Fig. 4d) $J(u, v)$ at the point in time at which the response to the central ommatidium impulse is at its extremum:

$$\mathrm{SRF}(u, v) = \mathrm{STRF}(n = \mathrm{argmax}_n |\mathrm{STRF}[n](0, 0)|, u, v).$$

We derive temporal receptive fields (TRFs) from the response to a flash $J(0, 0)$ at the central ommatidium: $\mathrm{TRF}[n] = \mathrm{STRF}[n](0, 0)$. For averaging receptive fields across multiple models, we first normalize the voltages as described above.

**Maximally excitatory naturalistic and artificial stimuli.** First, we found the naturalistic maximally excitatory stimulus, $X^*$, by identifying the Sintel video clip, **X**, from the full dataset with geometric augmentations that elicited the highest possible response in the central column cell of a particular cell type in our models.

$$X^* = \mathrm{argmax}_{X \in \mathrm{Sintel}} V_{t_{\mathrm{central}}}(X).$$

Next, we regularized the naturalistic maximally excitatory stimulus, to yield $X'$, capturing only the stimulus information within the receptive field of the cell, with the objective to minimize

$$L(X') = \sum_n \|V_{t_{\mathrm{central}}}(X^*)[n] - V_{t_{\mathrm{central}}}(X')[n]\|^2 + \frac{1}{C}\sum_c \|X'[n, c] - 0.5\|^2.$$

The first summand preserves the central response to X*, and the second regularizes the irrelevant portions of the stimulus outside the receptive field to grey ($I = 0.5$).

In addition, we computed artificial maximally excitatory stimuli[75].

**Model selection.** To describe the most data-consistent motion tuning mechanisms predicted by the ensemble at the level of single-cell currents, for Extended Data Figs. 4, 7 and 8, we automatically selected those models from the ensemble with tuning matching to empirical data. Specifically, we selected models with correct contrast tuning in the respective target cells and their inputs (Fig. 4c and Extended Data Fig. 3d), with the DSI larger than the threshold $d^*$ derived above, and with a correctly predicted preferred direction (45° acceptance angle, assuming 225° for TmY3).

### Training synthetic connectomes
**Training feedforward synthetic ground-truth connectome networks.** Sparsified feedforward neural networks with six hidden layers (linear transformations sandwiched between rectifications) with equal number of neurons in each hidden layer functioned as ground-truth connectome networks. The main results describe networks with 128 neurons per hidden layer. We interpret the individual units as neurons with voltage

$$V_i = \sum_j s_{ij} + V_i^{\text{rest}} = \sum_j \sigma_j c_{ij} m_{ij} f(V_j) + V_i^{\text{rest}},$$

with presynaptic inputs $s_{ij}$ and resting potentials $V_i^{\text{rest}}$. The connectome-constrained synapse strength, $w_{ij}$, is characterized by the adjacency matrix $c_{ij}$, the signs $\sigma_j$, and the non-negative weight magnitudes $m_{ij}$. $c_{ij} = 1$ if the connection exists, else $c_{ij} = 0$. To respect Dale's law, the signs were tied to the presynaptic identity, $j$.

We identified the parameters $\sigma_j$, $m_{ij}$ and $V_i^{\text{rest}}$ by task optimization on handwritten digit classification (Modified National Institute of Standards and Technology (MNIST) database)[76]. We determined adjacency matrices, $c_{ij}$, for a given connectivity percentage using an iterative local pruning technique, the lottery ticket hypothesis algorithm[77]. The algorithm decreases the connectivity percentage of the ground-truth connectome networks while maintaining high task accuracy.

We optimized the ground-truth connectome networks and all simulated networks described below in PyTorch with stochastic gradient descent with adaptive moment estimation (ADAM with AMSGrad), learning rate 0.001, batch size 500, and an exponentially decaying learning rate decay factor of 0.5 per epoch. To constrain the weight magnitudes to stay non-negative, we clamped the values at zero after each optimization step (projected gradient descent). The parameters after convergence minimize the cross-entropy loss between the predicted and the ground-truth classes of the handwritten digits. More implementation detail is available in Supplementary Note 5.

**Simulated networks with known connectivity and unknown strength.** Simulated networks inherited connectivity, $c_{ij}$, and synapse signs, $\sigma_j$, from their respective ground-truth connectome networks. In simulated networks, signs and connectivity were held fixed. Weight magnitudes, $m_{ij}$, and resting potentials, $V_i^{\text{rest}}$, were initialized randomly and task-optimized. Just like ground-truth connectome networks, simulated networks were trained on the MNIST handwritten digit classification task until convergence.

**Simulated networks with known connectivity and known strength.** Alternatively, we imitate measurements of synaptic counts from the ground-truth weight magnitudes:

$$\widetilde{m}_{ij} = m_{ij} \epsilon_{ij} \text{ with } \epsilon_{ij} \sim \mathcal{U}(1 - \sigma, 1 + \sigma),$$

with multiplicative noise to imitate spurious measurements. We used $\sigma = 0.5$ for the main results. Weight magnitudes were initialized at the measurement, $\widetilde{m}_{ij}$, and task-optimized on MNIST with the additional objective to minimize the squared distance between optimized and measured weight magnitudes, $\widetilde{m}_{ij}$ (L2 constraint, Gaussian weight magnitude prior centred around the simulated network's initialization). We weighted the L2 constraint ten times higher than the cross-entropy objective to keep weight magnitudes of the simulated networks close to the noisy connectome measurements. Resting potentials, $V_i^{\text{rest}}$, were again initialized randomly and task-optimized.

**Measuring ground-truth-simulated network similarity.** Ground-truth-simulated network similarity was measured by calculating the median Pearson's correlation of tuning responses (rectified voltages) of corresponding neurons in the ground-truth-simulated network pair. In each of the 6 hidden layers, $N = 100$ randomly sampled neurons were used for comparison. Response tuning was measured over input stimuli from the MNIST test set ($N = 10,000$ images). Results are medians over all hidden layers and over 25 ground-truth-simulation network pairs.

### Reporting summary
Further information on research design is available in the Nature Portfolio Reporting Summary linked to this article.

## Data availability
Data, trained models and interactive notebooks are available at https://www.github.com/TuragaLab/flyvis.

## Code availability
Code is available at https://www.github.com/TuragaLab/flyvis.

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

**Acknowledgements** We thank L. Scheffer and L. Umayam for assistance with accessing connectome reconstructions; A. Borst, J. Fitzgerald, N. Klapoetke, G. Rubin, M. Reiser, K. Svoboda and members of the laboratories of S.C.T. and J.H.M. for discussions; J. Fitzgerald, D. Stern, N. Klapoetke, A. Lee, R. Gao, J. Voigts, B. Mensh, A. Schulz, P. Ramesh, M. Deistler, Z. Stefanidi and J. Joyce for feedback on the manuscript; and T. Herman for creating and sharing the colourization of the optic lobe figure[25] (Fig. 1b). The graphics of the fruit fly and the electron microscope in Fig. 1a were created with BioRender.com. This article is subject to HHMI's Open Access to Publications policy. HHMI laboratory heads have previously granted a nonexclusive CC BY 4.0 license to the public and a sublicensable licence to HHMI in their research articles. Pursuant to those licences, the author-accepted manuscript of this article can be made freely available under a CC BY 4.0 licence immediately on publication. This project was supported by the HHMI. J.K.L. and J.H.M. were supported by the German Research Foundation (DFG) through Germany's Excellence Strategy (EXC-Number 2064/1, Project number 390727645), the German Federal Ministry of Education and Research (BMBF; Tübingen AI Center, FKZ: 01IS18039A) and the European Union (ERC, DeepCoMechTome, 101089288). Views and opinions expressed are however those of the author(s) alone and do not necessarily reflect those of the European Union or the European Research Council. Neither the European Union nor the granting authority can be held responsible for them. J.H.M. is the principal investigator of the DFG-financed SFB 1233. J.K.L. is a member of the International Max Planck Research School for Intelligent Systems.

**Author contributions** Conceptualization, methodology: J.K.L., F.D.T., M.M., J.H.M. and S.C.T. Data curation: J.K.L., F.D.T., A.N., K.S. and S.-y.T. Software and investigation: J.K.L., M.M., S.P. and F.D.T. Analysis: J.K.L., E.G., A.N., S.P. and S.C.T. Writing: J.K.L., J.H.M. and S.C.T. Writing (review and editing): E.G., A.N., K.S., M.M., S.P. and F.D.T. Supervision and funding: J.H.M. and S.C.T.

**Competing interests** The authors declare no competing interests.

**Additional information**
**Correspondence and requests for materials** should be addressed to Srinivas C. Turaga.

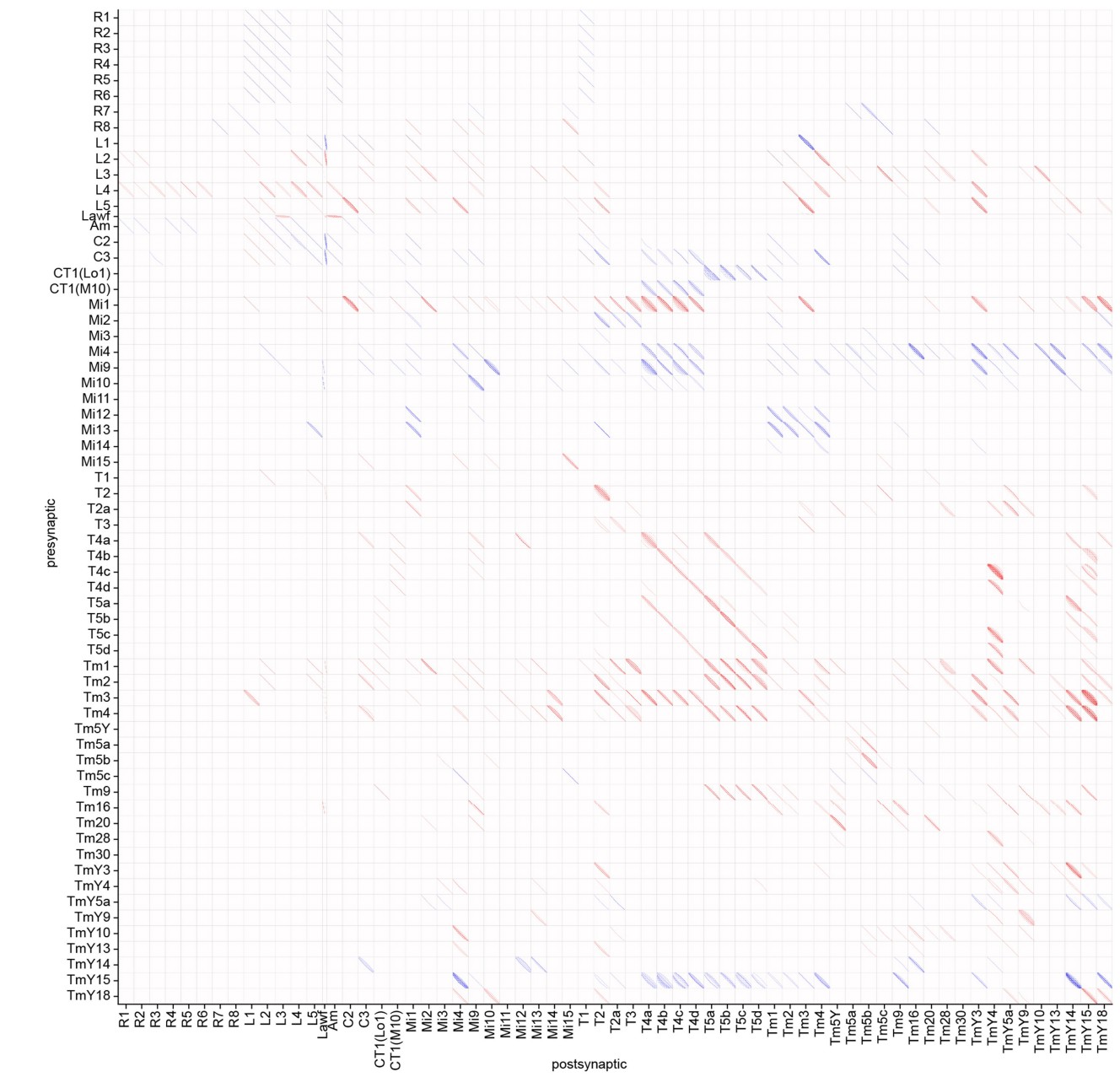

**Extended Data Fig. 1 | Cell connectivity.** The matrix shows how cells of the 64 cell types within the inner 91 columns (of 721) of the recurrent convolutional DMN connect (Supplementary Data file 1), either by excitatory connections (red) or inhibitory connections (blue).

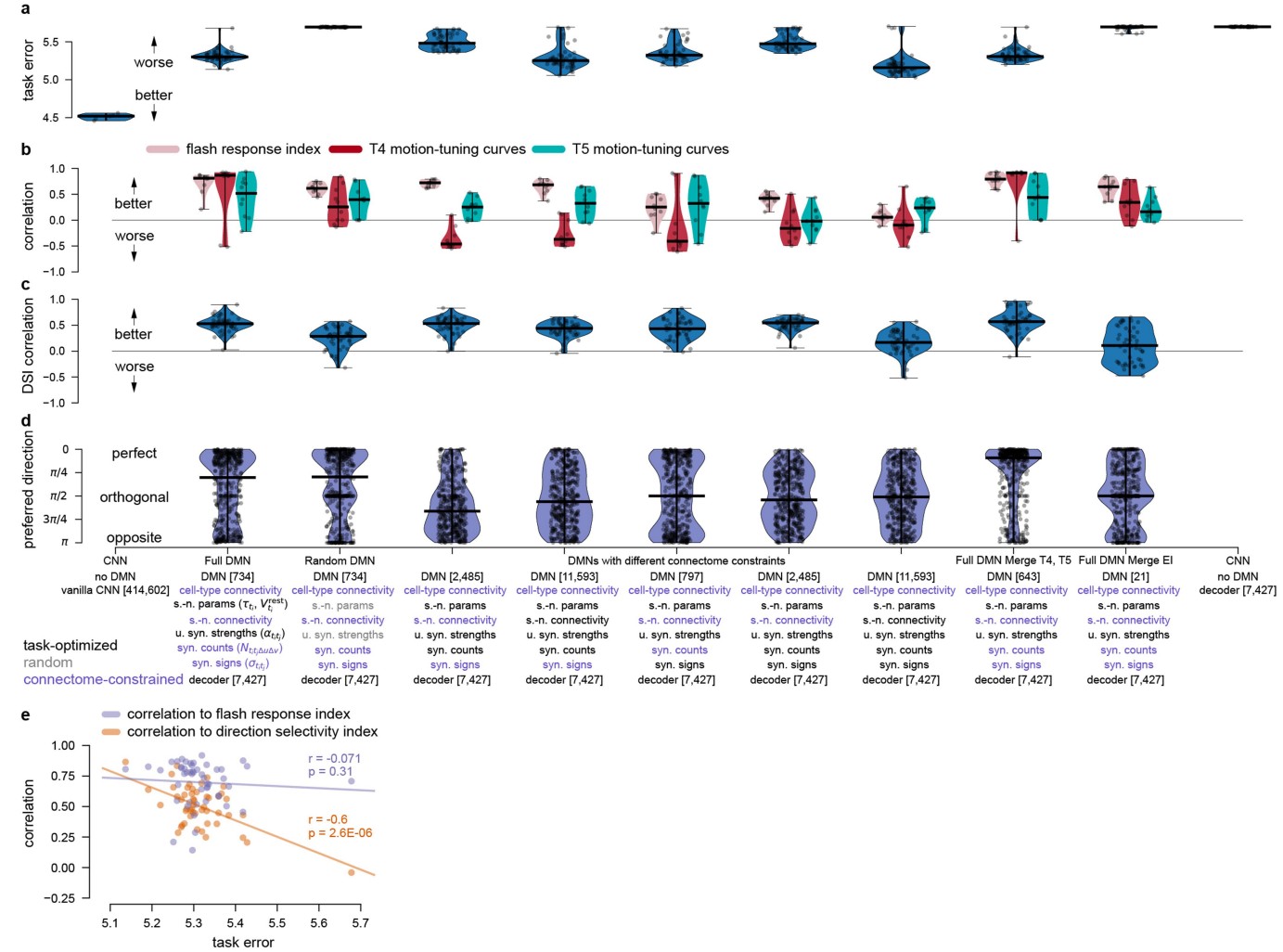

**Extended Data Fig. 2 | DMN benchmark of connectomic constraints.**
(**a-d**) **How would incomplete knowledge of connectome affect the tuning predictions?** We artificially varied DMNs with random parameters, connectome-constrained or task-optimized parameters. Five experiments: Four 'Synapse-optimized models', one 'Fully optimized'. Details in Methods. **How would incomplete knowledge of cell types affect the tuning predictions?** We artificially assumed some cell types to be indistinguishable, with shared physiological parameters (resting potentials, time constants, and unitary synapse strengths). Two experiments: (1) 'Full DMN Merge T4, T5' assumes that T4 and T5 subtypes were indistinguishable, reducing the number of cell types to 58. (2) 'Full DMN Merge E/I' assumes that we had three cell types, excitatory (37 cell types), inhibitory (22 cell types) or both (4 cell types), based on our knowledge of synapse signs. Tuning predictions are shown in comparison to the Full DMN and the DMN with random parameters.

(**a**) Task error. (**b**) Predicted correlations to flash response indices, T4, and T5 motion-tuning curves (10 best models). (**c**) Predicted correlations to known direction selectivity indices. (**d**) Distances between known preferred directions and predicted preferred directions for T4 and T5 neurons. (**e**) **Better task-performing models predict motion-tuning neurons better**. We correlate predicted tuning metrics from each model to the known tuning properties to understand when better performing models give us better tuning predictions. (**orange**) When correlating the direction selectivity index of each model to the binary known properties for T4 and T5 and their input cell types, we find that this correlation is higher for better performing models (Pearson correlation, $r = -0.60$, p = $2.6 \times 10^{-6}$, $t = r\sqrt{\frac{df}{1-r^2}}$, 95% CI = [−1, −0.42], df = 48). (**magenta**) While the models predicted the known contrast preferences generally well, the correlation of flash response index to the binary known contrast preferences of 31 cell types did not significantly increase with better performing models.

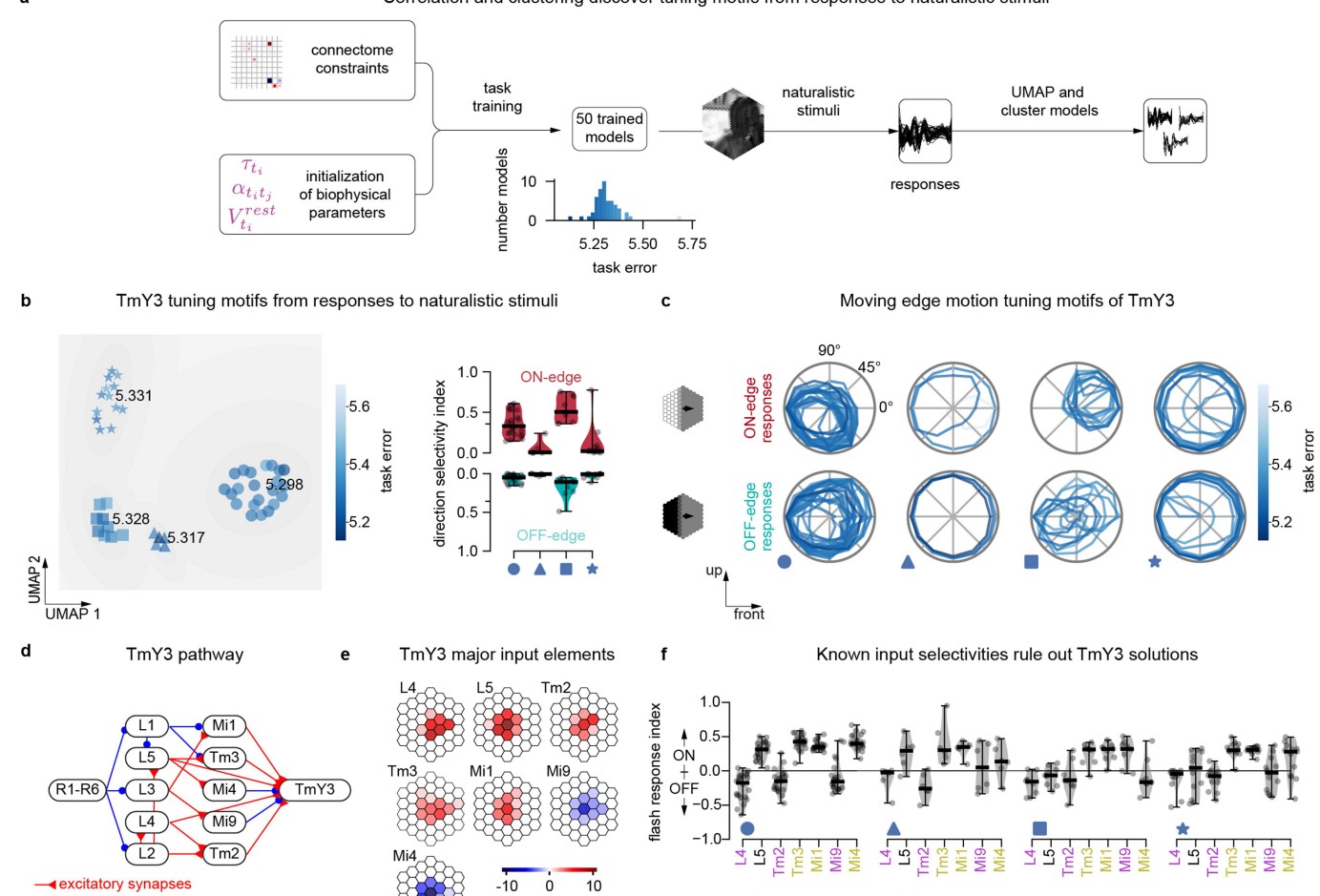

**Extended Data Fig. 3 | DMNs suggest that TmY3 neurons compute motion independently of T4 and T5 neurons. (a)** We clustered 50 DMNs after performing nonlinear dimensionality reduction of their responses to naturalistic scenes for each cell type, and aimed to identify whether clusters correspond to qualitatively different tuning mechanisms. **(b)** Dimensionality reduction on TmY3 responses to naturalistic stimuli reveals 4 clusters of DMNs with average task errors 5.298 (circle), 5.317 (triangle), 5.328 (square) and 5.331 (star). Across clusters, TmY3 shows different strengths of direction selectivity (evaluated with moving edge stimuli). ON-edge direction selectivity is strong in the first and the third cluster. **(c)** Normalized peak responses of TmY3 to moving edge stimuli in the DMNs of each cluster. **(d)** Major cell types and

synaptic connections in the pathway that projects onto TmY3 (simplified). **(e)** The input elements of TmY3 with the highest amount of synapses are L4, L5, Tm2, Tm3, Mi1, Mi9, and Mi4. The asymmetries of their projective fields could allow TmY3 to become motion selective. **(f)** Dependencies between TmY3 tuning and the contrast preference of its input cells. For clusters in which TmY3 is motion selective, cluster 1 (TmY3 tuning to downwards/front-to-back motion, circular marker) indicates ON-selectivity for Tm3, Mi1, and Mi4 cells, and OFF-selectivity for L4, Tm2, and Mi9 cells, in agreement with known selectivities. In contrast, cluster 3 (TmY3 tuning to upwards/back-to-front motion, square marker) indicates ON-selectivity for Mi9 in contradiction to the known selectivities and hence ruling out the third TmY3 tuning solution.

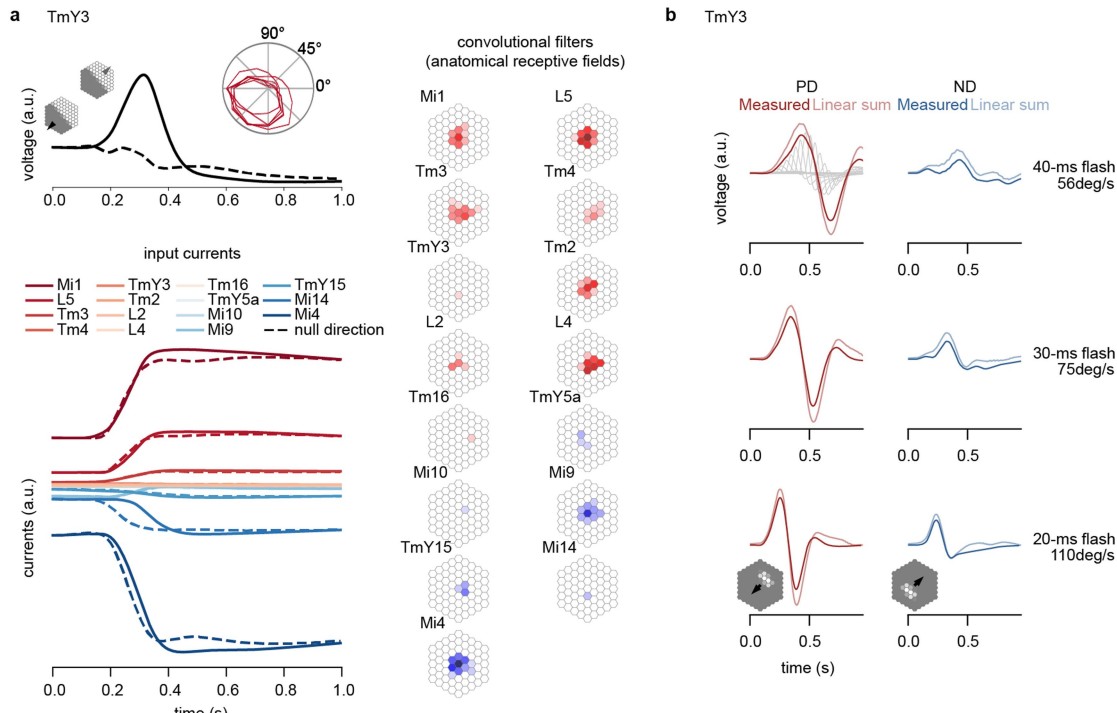

**Extended Data Fig. 4 | TmY3 motion detection mechanisms hypothesized by the model. (a)** Responses to PD and ND ON-edge motion and contributions from input elements. (**b**) PD enhancement and ND suppression in TmY3 in the model.

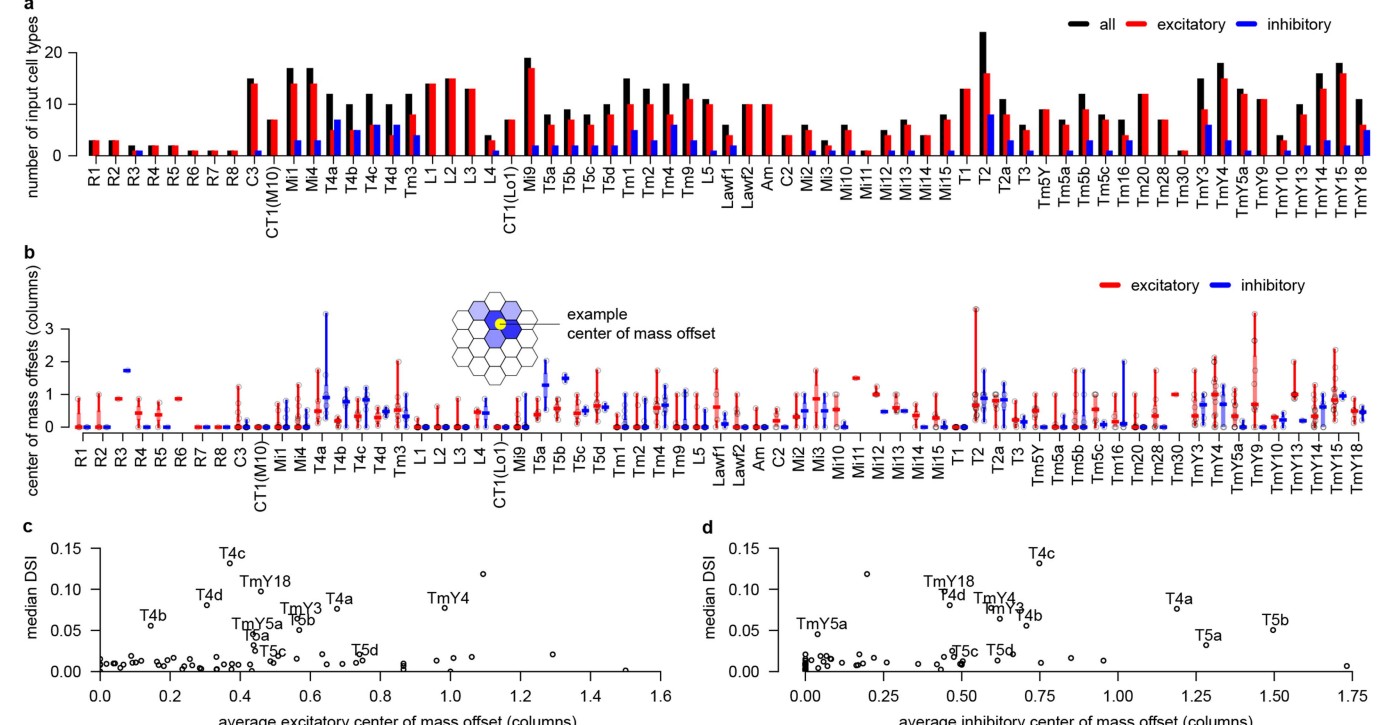

**Extended Data Fig. 5 | Statistics of inhibitory and excitatory synapse inputs.** (**a**) Number of input cell types per cell type. (**b**) Center of mass offsets of synaptic input. (**c**) Average excitatory and (**d**) inhibitory center of mass offset of synaptic inputs against median predicted direction selectivity index for all cell types. Datapoints for cell types that were predicted as significantly motion selective are labeled.

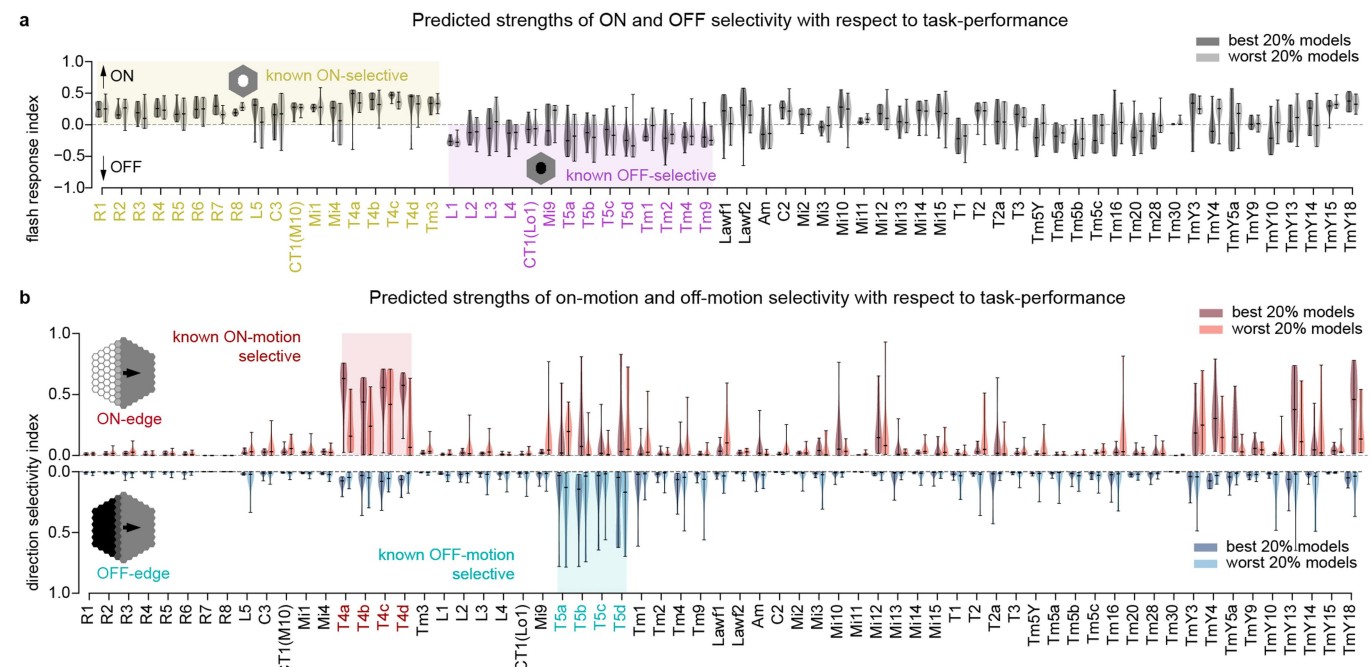

**Extended Data Fig. 6 | Predicted tuning with respect to task-performance.**
(**a**) Flash response index computed as the max-abs-scaled peak response to an off flash subtracted from the max-abs-scaled peak response to an on flash – both of approximately 35° radius and presented for 1 s after 2 s of grey input. Values above 0 indicate on-polarity, values below zero indicate off-polarity.

Known on-polar and off-polar cell types are colored in yellow and magenta.
(**b**) Single-cell type direction selectivity of best 20% task-performing models versus worst 20% task-performing models of an ensemble of 50 models as a result of peak voltage responses in central columns to on-edges and off-edges moving towards all possible directions on grey background.

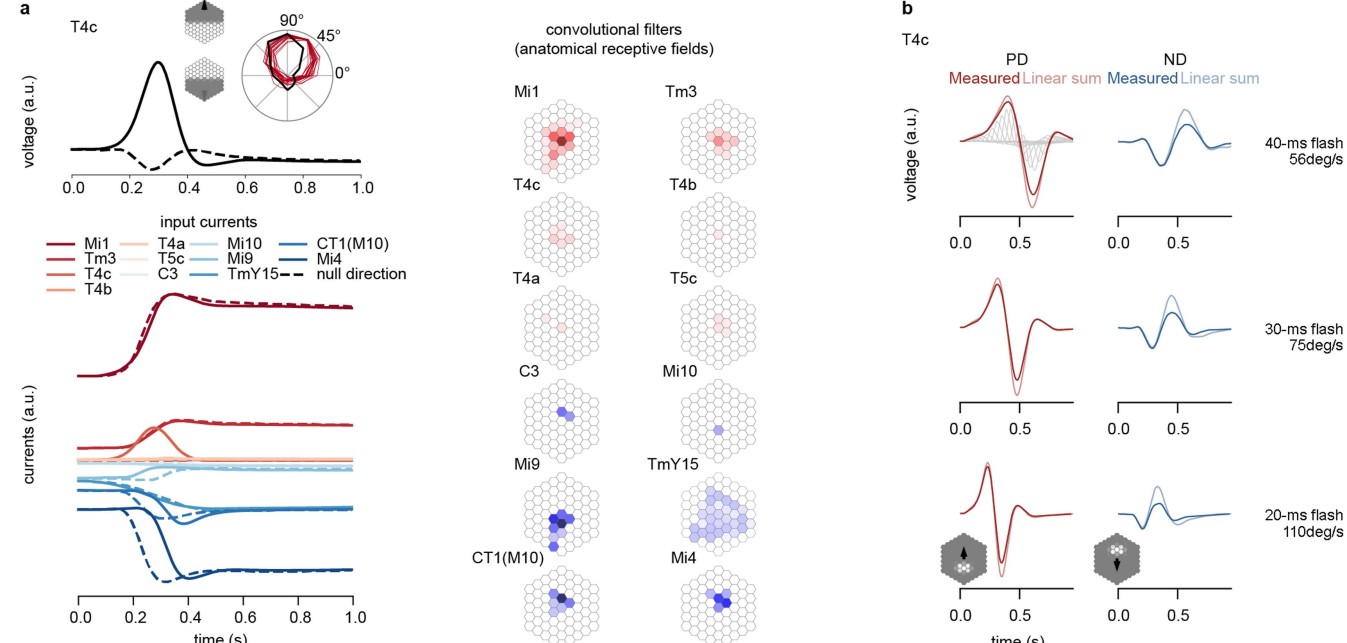

**Extended Data Fig. 7 | T4 motion detection mechanisms hypothesized by the model.** (**a**) Across all T4 cell types (here T4c, Supplementary Fig. 5 for other T4 types), our model predicts that T4 depolarization in response to PD ON-motion (black, solid) is driven by excitatory Mi1 current inputs (darkest red, solid) from roughly a two-column radius of Mi1 cells. Excitatory inputs from neighboring T4 cells of the same type increase the T4 PD-motion response (third darkest red, solid). Tm3 and Mi1 cells excite T4 agnostic to PD vs. ND motion. For ND-motion, Mi4 cells cancel excitatory currents from Mi1 with matching inhibition from the trailing side of the receptive field (darkest blue, dashed). The inhibition from Mi4 cells is delayed for PD-motion (darkest blue, solid), allowing strong depolarization of T4. CT1 shadows the Mi4 mechanism with similar but weaker inhibition from the same location of the receptive field

(second darkest, blue). Our model suggests mechanisms involving Mi9 cells and TmY15 cells: both contribute to T4 motion detection by different inhibitory mechanisms for PD-motion with respect to ND-motion. (**b**) 'Measured': Predicted T4c responses to bars moving in PD (left column) and in ND (right column) at varied speeds (saturated red and blue). 'Linear sum': linear sum of responses to individually flashed frames that constitute the moving bar video (faint red and blue). Faint grey traces in background of first panel show individual flash responses before linear summation. Flash duration in each location matched to length of stay at the location in the moving bar video. Bars were approximately 9° wide and 20.25° high and moved across 45° with respect to receptive field in the center. This figure should be compared to Gruntman et al.[27], Fig. 4f.

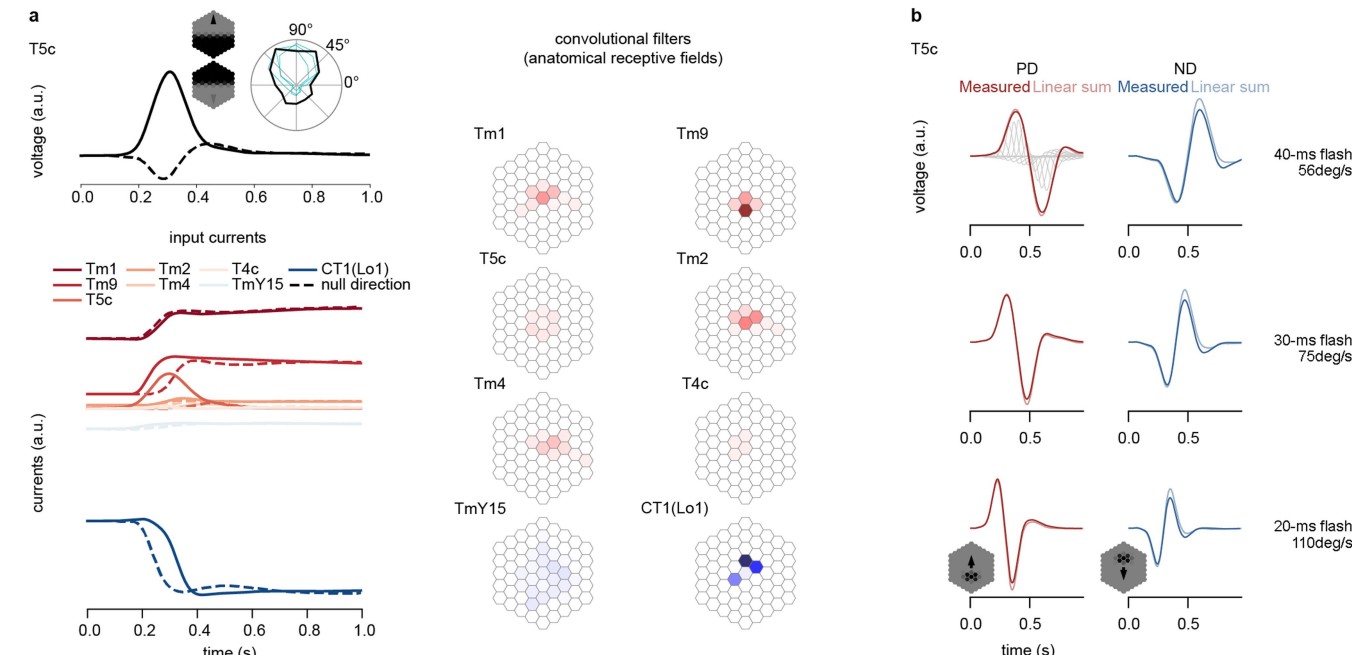

**Extended Data Fig. 8 | T5 motion detection mechanisms hypothesized by the model.** (**a**) Across all T5 cell types (here T5c, Supplementary Fig. 6 for other T5 types), our model predicts that T5 depolarization in response to PD OFF-motion (black, solid) is driven by excitatory Tm1 and Tm9 input currents (darkest and second darkest red). Tm1 currents come from a centered, two-column radius of Tm1 cells. Tm9 inputs come from cells offset by one column towards the leading side of the receptive field. We observe delayed excitation from Tm9 cells for ND-motion. The PD-motion response is increased through excitatory inputs from the neighboring T5 cells of the same type (as for T4 cells), not providing excitation for ND-motion. CT1(Lo1) cells cancel excitatory currents with strong inhibitory currents from the trailing side of the receptive

field leading to the weak ND response. For PD-motion, inhibition from CT1(Lo1) cells is delayed allowing strong T5 depolarization. (**b**) 'Measured': Predicted T5c responses to bars moving in PD (left column) and in ND (right column) at varied speeds (saturated red and blue). 'Linear sum': linear sum of responses to individually flashed frames that constitute the moving bar video (faint red and blue). Faint grey traces in background of first panel show individual flash responses before linear summation. Flash duration in each location matched to length of stay at the location in the moving bar video. Bars were approximately 9° wide and 20.25° high and moved across 45° with respect to receptive field in the center.

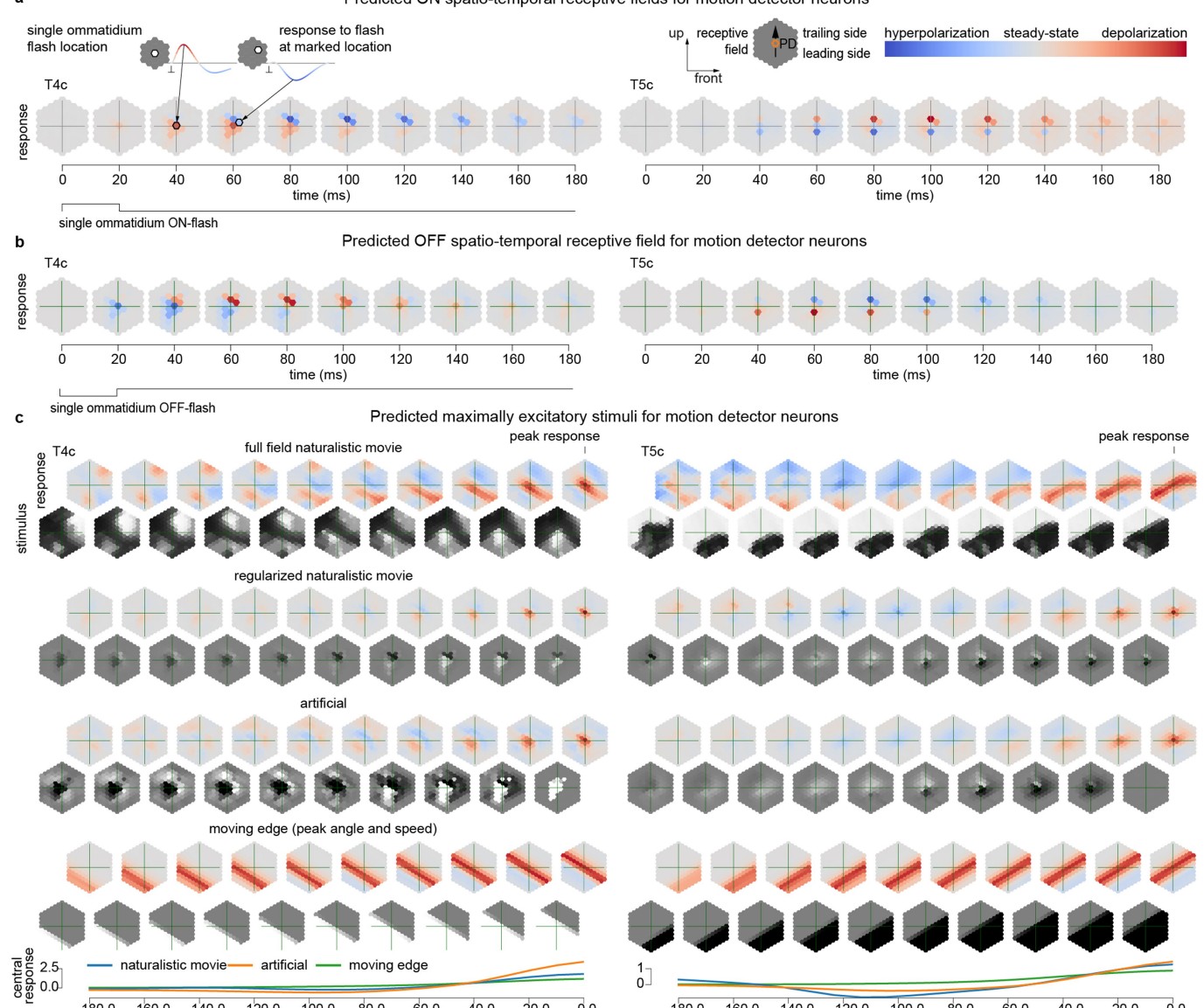

**Extended Data Fig. 9 | Spatio-temporal receptive fields mapped with ON-and OFF-impulses and maximally excitatory stimuli.** (**a**) Spatiotemporal receptive fields for motion detector neurons agree with experimental measurements (Gruntman et al.[27]). (**b**) Spatio-temporal receptive field mapping with single ommatidium OFF-impulses. (**c**) Maximally excitatory stimuli and baseline-subtracted responses. Including full-field naturalistic, regularized naturalistic, artificial, and moving edge stimuli and responses. Moving edge angle and speed maximize the central cell peak response. Artificial stimuli are optimized from initial noise to maximize the central cell activity using gradient ascent plus full-field regularization towards grey. The last row shows the baseline-subtracted central cell responses. Peak central cell responses at time point zero.

# Reporting Summary

## Statistics

For all statistical analyses, confirm that the following items are present in the figure legend, table legend, main text, or Methods section.

| n/a | Confirmed | |
|---|---|---|
| ☐ | ☒ | The exact sample size (*n*) for each experimental group/condition, given as a discrete number and unit of measurement |
| ☐ | ☒ | A statement on whether measurements were taken from distinct samples or whether the same sample was measured repeatedly |
| ☐ | ☒ | The statistical test(s) used AND whether they are one- or two-sided <br> *Only common tests should be described solely by name; describe more complex techniques in the Methods section.* |
| ☒ | ☐ | A description of all covariates tested |
| ☐ | ☒ | A description of any assumptions or corrections, such as tests of normality and adjustment for multiple comparisons |
| ☐ | ☒ | A full description of the statistical parameters including central tendency (e.g. means) or other basic estimates (e.g. regression coefficient) AND variation (e.g. standard deviation) or associated estimates of uncertainty (e.g. confidence intervals) |
| ☐ | ☒ | For null hypothesis testing, the test statistic (e.g. *F*, *t*, *r*) with confidence intervals, effect sizes, degrees of freedom and *P* value noted <br> *Give P values as exact values whenever suitable.* |
| ☒ | ☐ | For Bayesian analysis, information on the choice of priors and Markov chain Monte Carlo settings |
| ☒ | ☐ | For hierarchical and complex designs, identification of the appropriate level for tests and full reporting of outcomes |
| ☐ | ☒ | Estimates of effect sizes (e.g. Cohen's *d*, Pearson's *r*), indicating how they were calculated |

*Our web collection on statistics for biologists contains articles on many of the points above.*

## Software and code

Policy information about availability of computer code

| Data collection | Code, data, trained models, and interactive notebooks for analysis are available under https://www.github.com/flyvis/flyvis. |
|---|---|
| Data analysis | Code, data, trained models, and interactive notebooks for analysis are available under https://www.github.com/flyvis/flyvis. |

For manuscripts utilizing custom algorithms or software that are central to the research but not yet described in published literature, software must be made available to editors and reviewers. We strongly encourage code deposition in a community repository (e.g. GitHub). See the Nature Portfolio guidelines for submitting code & software for further information.

## Data

Policy information about availability of data

All manuscripts must include a data availability statement. This statement should provide the following information, where applicable:
- Accession codes, unique identifiers, or web links for publicly available datasets
- A description of any restrictions on data availability
- For clinical datasets or third party data, please ensure that the statement adheres to our policy

Code, data, trained models, and interactive notebooks for analysis are available under https://www.github.com/flyvis/flyvis.

# Field-specific reporting

Please select the one below that is the best fit for your research. If you are not sure, read the appropriate sections before making your selection.

☒ Life sciences ☐ Behavioural & social sciences ☐ Ecological, evolutionary & environmental sciences

For a reference copy of the document with all sections, see nature.com/documents/nr-reporting-summary-flat.pdf

# Life sciences study design

All studies must disclose on these points even when the disclosure is negative.

| | |
|---|---|
| Sample size | No experimental data was collected. For simulations, number of simulations was constrained depending on our computational resources. |
| Data exclusions | No data was excluded. |
| Replication | The manuscript is based on 50 models, to estimate variability across the model. Additional runs were within this range of variability. |
| Randomization | For our computational experiments, datasets were randomly split into training and test sets. Training data was presented in a random order, and data augmentation was applied randomly. Model parameters were initialized randomly. |
| Blinding | Blinding was not relevant to our study because it was fully computational. |

# Reporting for specific materials, systems and methods

We require information from authors about some types of materials, experimental systems and methods used in many studies. Here, indicate whether each material, system or method listed is relevant to your study. If you are not sure if a list item applies to your research, read the appropriate section before selecting a response.

### Materials & experimental systems

| n/a | Involved in the study |
|---|---|
| ☒ | ☐ Antibodies |
| ☒ | ☐ Eukaryotic cell lines |
| ☒ | ☐ Palaeontology and archaeology |
| ☒ | ☐ Animals and other organisms |
| ☒ | ☐ Human research participants |
| ☒ | ☐ Clinical data |
| ☒ | ☐ Dual use research of concern |

### Methods

| n/a | Involved in the study |
|---|---|
| ☒ | ☐ ChIP-seq |
| ☒ | ☐ Flow cytometry |
| ☒ | ☐ MRI-based neuroimaging |

