## [Peer Review File · Nature]

Manuscript Title: Connectome constrained networks predict neural activity across the fly visual system

Reviewer Comments & Author Rebuttals

Reviewer Reports on the Initial Version:

Referees' comments:

Referee #1 (Remarks to the Author):

This manuscript addresses the fundamental question whether connectomes (mutual connectivity between many neurons) can predict functions of individual neurons in a network when the computation of the network is known. The authors address this question in the visual system of *Drosophila*. They use recent connectomics datasets to create simulations of the visual neuropils with biologically realistic neuron types, spatial arrangement, and connectivity. Parameters describing individual neurons and synapses were adjusted by training networks to detect optic flow. Training improved model performance compared to untrained models with realistic connectivity. Some networks, but not others, reproduced known functional properties of individual neurons such as directional motion sensitivity of T4 and T5 cells. Model neurons showed more biologically realistic properties when connectivity was sparse.

The authors conclude that network models reproducing biological mechanisms of computation can be found by including connectomes as constraints in networks trained end-to-end on a computational task ("DMNs"). They emphasize that it is not necessary to include additional information from neuronal recordings or biophysical measurements. They further propose connectome-constrained network modeling as a tool to discover computational functions and meaningful experiments. These conclusions are based on the observation that some models reproduced functional properties of single neurons such as the motion-sensitive T4 and T5 cells and therefore reproduce the biological mechanism of motion detection. This is indeed a remarkable result. But other models also detect optical flow and fail to reproduce single-neuron properties. Additional knowledge from single-neuron recording experiments (or other sources) is therefore still required to select biologically realistic models. It is also not clear how the approach may or may not generalize to other systems given that the *Drosophila* visual system is a highly specialized, repetitive, extensively studied network of graded potential neurons. In summary, I believe that the results of this study are remarkable and demonstrate that including knowledge about synaptic connectivity into network simulations can take us a huge step forward towards mechanistic modeling of biological neural networks. This is clearly an important result but it remains open whether the approach can be generalized with similar success to other brain circuits.

The main result of the study is the general notion that mechanistic modeling of biological neural networks can be achieved by including connectomes as constraints. So far, the study did not produce major new insights into the function or structure of visual processing in *Drosophila*. A prediction from this study is that neuron TmY3 may be a previously unrecognized motion-sensitive neuron, but this prediction remains to be tested experimentally.

Specific points:

1. The main message of this manuscript is not a deeper understanding of visual processing in *Drosophila* but the general notion that connectomes help (massively) to create biologically realistic network models. Indeed, the authors emphasize that their "...modeling approach provides a discovery tool...". It is thus important to get a good idea how the DMN approach generalizes to other brain circuits. The DMN approach is likely to be facilitated by features of the *Drosophila* visual system such as graded synaptic transmission and a highly repetitive architecture. More insight into the potential for generalization of this approach would be useful (see also below).

2. The graded potential neurons used in this study are biologically realistic for the *Drosophila* visual system but not for most other brain circuits. It may be more difficult to get DMNs to reproduce biological mechanisms of computation with spiking neurons. This may be a limitation of the approach that should be discussed (or explored, if possible).

3. The repetitive layout of the *Drosophila* visual system facilitates network modeling, and so does the extensive knowledge of cell types in this system. How would the DMN approach be affected if knowledge of the connectome were less complete, as is often the case in other brain circuits? How would it be affected if cells were divided into fewer distinct types?

4. An interesting observation is that the same neurons in different networks do not always show the same functional properties and form discrete clusters in functional space. For T4c cells, for example, 3 clusters were found but only one represents biologically realistic neurons with correct motion sensitivity. This observation is interesting because it can, in principle, be a starting point to explore general principles of network design. On the other hand, it means that the DMN approach alone is not sufficient to predict mechanisms of computation, even in this well-established system. Additional knowledge (here: true motion sensitivity of T4 cells) is necessary to distinguish biologically realistic from unrealistic networks. Such knowledge may be hard to come by in other systems. This is a (potentially serious) limitation of the DMN approach that needs to be discussed more. What type of additional information would be most useful to resolve "cluster ambiguities"?

5. The authors emphasize that DMNs can reproduce biological mechanisms of computation, but they do not go very far in analyzing computational mechanisms beyond current knowledge. So far, they mainly asked whether known mechanisms are reproduced in the DMN. For example, motion sensitivity of T4 cells involves direction-dependent temporal shifts between excitatory and inhibitory input currents, which is reproduced by the model. However, the authors could go further and manipulate specific connections to verify that motion sensitivity is indeed generated by the expected implementation of a computational strategy (combination of Hassenstein-Reichardt and Barlow-Levick models) in neural circuitry. Such an analysis should have potential to uncover novel, unknown functions. Similarly, they could use specific manipulations of connectivity to analyze the mechanisms of motion sensitivity in TmY3, following up on the speculations put forward in the text. Generally, the ability to manipulate connections in a biologically realistic simulation has interesting potential because this is often not possible experimentally.

6. The authors suggest that TmY3 is a novel motion-sensitive neuron that has not been recognized previously and computes motion independently from T4 and T5 cells. It is also predicted that other

neurons should be motion sensitive (TmY4, TmY18), probably because they receive input from T4 and T5. So far, these predictions have not been tested experimentally. Doing so could substantially enhance the impact of this study.

7. Abstract: "...we show that with only measurements of the connectivity of a biological neural network, we can predict the neural activity underlying neural computation". This statement is not correct. DMNs also use knowledge of the computation (input-output) for training, and additional knowledge is required for model selection (for example activity/tuning of T4/T5 cells).

8. How good is the optic flow detection achieved by DMNs? The quantification by the error measure is not very intuitive. It shows that training improves performance, but unconstrained CNNs still achieve much better performance than trained DMNs. It would be good to provide more information to get a better intuition how well a DMN is performing in comparison to a real fly.

Minor comments:

9. Ln 481: Fig 1g

Referee #2 (Remarks to the Author):

The authors use connectome data from the fly visual system combined with optic flow training to produce a task-performing mechanistic model with interpretable parameters. Comparing to previous data, the model captures many of the tuning properties of fly visual neurons.

There has been some previous work that uses connection data to define model architectures for task-training. The authors may want to cite some of this work from *C. elegans* (e.g. <https://www.ncbi.nlm.nih.gov/pmc/articles/PMC8253844/> and <https://arxiv.org/pdf/2201.05242.pdf>). The present work is novel in my opinion in the extent to which it compares the neural parameters of the trained model to data.

It would probably be good to provide a contextualization of the DMN task performance and parameter recovery by pulling in some more of the supplementary results (at least more quantified descriptions of them in the text). For example, the main text does not convey how relatively minor the enhancement in performance on the task is for the DMN vs random model, in the context of how well the unconstrained CNN can perform. Also, it seems relevant to note that the random models are still positively correlated with cell tuning and the flash response results can be captured by even the poorly performing DMNs.

Are the T5 off-motion selective neurons in 2c supposed to be tuned to on-motion as well? Or is this a way in which the model does not fit the data? This should be spoken to in the paper.

I have several concerns/questions regarding the synthetic connectome experiments in the MNIST-trained networks. First, I do not understand the motivation behind the version with noisy weight estimates where getting the correct weights is baked into the objective function. What do we learn

from seeing that a network initialized with roughly the correct weights can explicitly learn to recover those weights (regardless of sparsity)?

For the connectome-only version, this still does not seem to necessarily support what the authors seem to be claiming about it. Specifically in the absence of any information that leads to unique cell IDs, comparison of tuning across networks is meaningless. With a sparse network (including the fly connectome) the pattern of connections a cell makes can be a unique identifier for it, and therefore these cells can be labeled as the same and their tuning can be compared across networks. Such unique identities are not possible in densely connected networks. Therefore, the tuning comparisons done here are essentially as if two random neurons were picked across models and expected to have the same tuning. The fact that a random pairing of neurons does not display the same tuning does not mean these networks are not learning the same mechanisms. In fact, ED Fig 9a shows that Dale's law helps with the correlation, which is likely because having a weight constraint offers some kind of (weakly) unique identifier. Also, how were the signs decided for units in these networks?

Clarifications:

Can the authors better explain the differentiation between synapse count and the scalar? It seems the scalar and the count are held constant for all pairs of cells with the same pre- and post-synaptic cell type, so what extra information does having the count as part of the equation provide?

How do the 50 ensemble models differ? Just different draws from the same distribution of resting potentials?

Referee #4 (Remarks to the Author):

Lappalainen et al. build an optic lobe connectome-constrained neural network called a task-optimized deep mechanistic network (DMN), and optimize for a computation performed by that biological circuit (motion detection). They show that constraining both the connectivity and computational task reproduces some of the experimentally determined tuning of specific neurons, and makes predictions about the tuning properties of other neurons in the network that have yet to be experimentally measured. Finally, the paper argues that, for sparse networks (such as some biological neural networks), knowledge of the connectivity, signs of connections, and an estimate of connection strength may be sufficient to predict the mechanism by which the circuit performs a known computational task.

I am in general enthusiastic about the study - it is a useful simulation of a portion of real, complex neural network (64 cell types and 721 columns, plus 1 inhibitory cell type that extends across columns, but missing all of the feedback and neuromodulatory connections) and shows how connectivity shapes many of the known properties of a neural network - the optic lobe is an ideal test case as its cell types have been studied extensively over the past 60 years. Figures 3 and 4 in particular are quite nice - 1) comparisons between the best performing model's T4/T5 cells and the

response properties of their inputs to known tuning curves and responses (although I have concerns about some of the details - see below) and 2) comparisons between different models to understand what properties of particular cell types define the best performing models. This is a nice study that will form the basis for simulations of larger biological networks, for fly, and other species. However, I have several concerns about the modeling, model predictions, and interpretations of the results that should be addressed.

The authors build a hybrid optic lobe connectome from several different datasets - the choices they made in how to combine these datasets must be made transparent in the paper. Ideally, they would present a Supplemental Figure devoted to how this was done and how they handled any discrepancies or differences between the datasets. If there were no discrepancies or differences this should also be explained. It is difficult to interpret the findings from the model without a thorough understanding of how the connectome-constrained model was built. Related to this point, I assume the signs of connections were taken from the literature? Can the authors provide citations for these (and can they compare against the same cell type in the open FlyWire/FAFB whole-brain connectome dataset, for which neurotransmitter predictions (from Eckstein et al.) are available in the optic lobe)?

Related to the point above, the authors perform a sort of normalization step with the data so that they can model every column identically (making sure synapse numbers are the same in each column) - this ignores heterogeneity across columns (that might be important for motion detection). Can the authors provide more detail on how this simplification deviates from the actual connectivity (how much heterogeneity is there across columns?) and show how this choice affects modeling results (if they incorporate some of the heterogeneity into the model, how do the results change)? The manuscript claims that the actual OL connectivity was critical. Extended Data Figure 3 shows that a task-optimized DMN outperforms a DMN with random parameters. But what is missing is a demonstration that the specific connectivity of the fly visual system is what enables optimal performance, rather than a generic neural network with the same level of sparseness and gross connectivity statistics of the biological network. How would a task-optimized artificially generated network (constrained by biological connectivity statistics, rather than the exact connectivity of the fly visual system) perform compared to the task-optimized DMN and the random DMN?

Neurons are modeled with leaky linear non-spiking voltage dynamics and as point-neurons with a single electrical compartment. The authors show abundant tuning/response data for each model cell type, but they should compare the detailed temporal dynamics and delays (critical for motion detection) of model responses to real recordings of these same neurons (if this is present somewhere in the supplement, apologies if I missed it). Many optic lobe neurons have been recorded via Ca⁺⁺ imaging, voltage imaging, or electrophysiology (e.g., compare with published responses in Behnia et al. Nature 2014 or Yang et al. Cell 2016).

The L1 and L2 neurons are categorized as "known OFF selective" (Fig. 2b), but this deviates from my reading of the literature - shouldn't they show similar responses to both on and off flashes? Also, the responses of L1 and L2 in Fig. 3e are shown as monophasic and producing an off response - shouldn't they be biphasic (in contrast with the responses for L3 and L4, which do look biphasic)? These potential mismatches between the literature and model results have me concerned that the model is not producing the expected responses for cell types that have been extensively studied (like L1 and L2).

Given the constraints provided by the connectome, there are only 734 free parameters in the model

(resting membrane potential for each cell type (65) and unitary synapse strength (604)) - but the authors could have also included the synapse NL as a free parameter (varying across synapses) - how would this have affected modeling results? Is there a reason they need to limit to ~700 free parameters? The question has to do with the choice of which parameters to fix across the model and which to vary - how much do these choices affect results?

The authors optimize the model network to perform a particular motion vision task - this makes a lot of sense given that the major function of the optic lobe is to detect motion, but it is not clear how the results depend on this specific task - this needs to be addressed. What differences might they observe or expect if the network was optimized to perform a different task that the fly optic lobe mediates (for example, color or shape detection) or a specific fly behavioral task (like the optomotor response) - how would optimizing for a different task change the responses properties of neurons in the network? Addressing this question is critical for understanding the constraints of the connectome.

Model responses to moving edges in Fig. 2c: Why do many of the models show that T5 is responsive to ON moving edges (which it is not)? Could this be expanded upon in Figure 4 (that compares different models)?

I'm concerned about claims that the mechanism of motion detection matches experimental findings: The mechanism of how direction selectivity in T4 and T5 cells emerges is still debated. Haag et al. suggest there is both PD enhancement and ND suppression; Gruntman et al. argue for ND suppression only; Wienecke et al. argue for neither. It seems likely that the mechanism is different for the ON and OFF pathways. The manuscript shows that the tuning of T4 and T5 cells and their inputs (in some models) qualitatively matches the experimentally measured tuning, but can the authors clarify which mechanism of direction selectivity is matched/favored by their models? The authors have missed an opportunity to test their model through silencing experiments - inputs to T4 and T5 cells have been silenced experimentally (Strother et al. 2017; Serbe et al., 2016) - does silencing these neurons in the trained models recapitulate experimental findings/impair motion detection by the decoder network?

Of the 19 cell types with asymmetric inputs, only 12 are predicted to be motion selective - can the authors comment on differences between the inputs of these 12 and the other 7?

One of the most exciting findings is the prediction that TmY3 could possess direction selectivity independent of the T4/T5 pathway. This is a bold claim - that the model can be used to identify new motion sensors in the fly visual system that has been studied for more than 60 years. It seems reasonable (and feasible given that the authors' local collaborators) to ask the authors to validate this prediction with experimental data - whereas elsewhere the authors rely on published experimental data for comparisons, this prediction would require new experiments. Further, the authors could discuss if TmY3 is expected to function like an HR detector or BL detector, or is direction selectivity predicted to arise via a different mechanism?

I would like to see some of the statistics of the decoder network. The manuscript argues that it cannot compute motion itself, but is this true? Does the decoder only depend on the output of direction-tuned neurons? Does it perform equally well when the weights for non-direction tuned neurons are forced to zero? What is the minimal set of T and Tm cells required to detect optic flow?

Minor comments

Figure 1 depicts a male fly, though much of the connectome data used, I believe, comes from female

flies.

Winding et al. 2023 larval connectome paper should also be cited at line 7.

The meaning of lines 99-101 is not clear. The text argues that the DMN was optimized for a computational task performed by the real biological network (i.e. fly visual system), but citations 44 & 76 refer to mammalian cortex.

Line 119: typo in "backpropagation"

Line 105 says that motion detection is a "challenging" computation. "Challenging" is too subjective a term and can be removed. Arguably, theoretical models for motion detection proposed in the 1960's by Hassenstein & Reichert and Barlow & Levick are relatively simple.

Lines 283-285 claim that networks with different sparsity must use different computations. Why must this be true?

Lines 333-336 claim that "DMN models generate meaningful predictions in absence of neural activity measurements... (Fig 4)" feels too strong, since knowing actual tuning of Mi9 was critical in determining "correct" tuning of T4.

The limitations of the approach (e.g. simplistic modeling of neural dynamics, and lack of electrical synapses, neuromodulation and glia) are introduced at the outset. It would be nice if these limitations were further elaborated/explored in the discussion.

Line 1038: Missing reference to extended data figure.

Author Rebuttals to Initial Comments:

We thank Editor and the reviewers for the constructive and detailed comments, and for appreciating the importance of the study. We are excited about the assessment of our results as *remarkable* (R1) and *a huge step towards mechanistic modeling of biological neural networks* (R1), as *novel* (R2), and that our *study will form the basis for simulations of larger biological networks, for fly, and other species* (R4). In our study, we provide an approach for turning a connectome into detailed hypotheses of how the neural circuit works, and which neurons are involved in which computations. We show that our approach makes concrete predictions at the level of single-cell responses, which are surprisingly accurate, as we showed by testing its predictions against measurements made across 26 studies (and our study makes a large number of additional predictions).

Remarkably, connectome-constrained neural networks seem to learn solutions which are surprisingly similar to the ones implemented in the fly, at the level of yielding predictions for individual cell tuning and even circuit mechanisms. This finding goes vastly beyond previous ‘NeuroAI’ approaches e.g. in the mammalian cortex, in which the correspondence between artificial and biological networks was weaker, largely at the level of brain regions, and which provided limited mechanistic circuit-level insights.

Importantly, our methodology provides a new approach to generate meaningful hypotheses before making any activity measurements. The timeliness and importance of our approach and findings is underscored by recent developments: High-profile releases of connectomes have provided an unprecedented abundance of neural connectivity measurements, including many neurons from which activity measurements are unavailable (and likely will not be available for some time). This data gap highlights the dire need for frameworks to extract an understanding of how neural systems perform computations, and hence the importance of our approach.

Indeed, in a comment in Nature last month (“How AI could lead to a better understanding of the brain”, 07/11/23), Viren Jain writes that “*Guided by connectomic and other data to optimize thousands or even billions of parameters, machine-learning models could be trained to produce neural-network behaviour that is consistent with the behaviour of real neural networks — measured using cellular-resolution functional recordings.*” and “*Researchers could evaluate such models, for instance, by comparing their predictions about the neural activity of a system with recordings from the actual biological system.*” This is, precisely, what our study is successfully doing. We believe that the reviewers appreciated both the importance of the study, and the central messages.

At the same time, their comments also revealed opportunities for strengthening our manuscript. In addition to multiple clearer explanations, the updated manuscript now includes substantial additional analyses (details below):

1. To show that our models can be used to study circuit mechanisms underlying specific computations, we analyzed circuit mechanisms of direction selectivity in T4, T5 and TmY3, both by inspecting the current distributions from each cell type and by simulating inactivation experiments which we compared to published experimental measurements, finding yet more agreement. For instance, while research into the mechanism of T4 motion selectivity has largely focused on the role of feedforward inputs, our new analysis shows a novel prediction suggesting an important role for the significant lateral connectivity between T4 neurons enhancing responses to coherent motion across the visual field.
2. We conducted a large set of additional numerical experiments to analyze how specific constraints [cell connectivity, synapse counts, synapse signs] contribute to the prediction of tuning properties. These results provide a clear account that all of these constraints are necessary for obtaining a close match between model activity and empirically measured tuning.
3. We now also provide a detailed analysis of the temporal properties of major inputs to T4 and T5 cells, providing an example of how our modeling approach can be used as a hypothesis generated for detailed dissection of neural circuits.

Referee #1 (Remarks to the Author):

*This manuscript addresses the fundamental question whether connectomes (mutual connectivity between*
*many neurons) can predict functions of individual neurons in a network when the computation of the*
*network is known. The authors address this question in the visual system of Drosophila. They use recent*
*connectomics datasets to create simulations of the visual neuropils with biologically realistic neuron*
*types, spatial arrangement, and connectivity. Parameters describing individual neurons and synapses*
*were adjusted by training networks to detect optic flow. Training improved model performance compared*
*to untrained models with realistic connectivity. Some networks, but not others, reproduced known*
*functional properties of individual neurons such as directional motion sensitivity of T4 and T5 cells.*
*Model neurons showed more biologically realistic properties when connectivity was sparse.*

*The authors conclude that network models reproducing biological mechanisms of computation can be*
*found by including connectomes as constraints in networks trained end-to-end on a computational task*
*("DMNs"). They emphasize that it is not necessary to include additional information from neuronal*
*recordings or biophysical measurements. They further propose connectome-constrained network modeling*
*as a tool to discover computational functions and meaningful experiments. These conclusions are based*
*on the observation that some models reproduced functional properties of single neurons such as the*
*motion-sensitive T4 and T5 cells and therefore reproduce the biological mechanism of motion detection.*
*This is indeed a remarkable result. But other models also detect optical flow and fail to reproduce*
*single-neuron properties. Additional knowledge from single-neuron recording experiments (or other*
*sources) is therefore still required to select biologically realistic models.*

*It is also not clear how the approach may or may not generalize to other systems given that the*
*Drosophila visual system is a highly specialized, repetitive, extensively studied network of graded*
*potential neurons. In summary, I believe that the results of this study are remarkable and demonstrate that*
*including knowledge about synaptic connectivity into network simulations can take us a huge step forward*
*towards mechanistic modeling of biological neural networks. This is clearly an important result but it*
*remains open whether the approach can be generalized with similar success to other brain circuits.*

*The main result of the study is the general notion that mechanistic modeling of biological neural networks*
*can be achieved by including connectomes as constraints. So far, the study did not produce major new*
*insights into the function or structure of visual processing in Drosophila. A prediction from this study is*
*that neuron TmY3 may be a previously unrecognized motion-sensitive neuron, but this prediction remains*
*to be tested experimentally.*

Thank you for your comments, and for appreciating the importance of the central question of our
study and its results-- indeed, we also find it remarkable that a large number of single-cell
properties can be predicted from connectome- and task-constraints alone. Our large circuit
model can be mapped onto individual cells and performs motion computation while capturing an
impressive amount of biological realism.

We should clarify that our goal with this study was to evaluate the utility of a connectome by
demonstrating how far we can get while only relying on measurements of
connectivity---essentially a show-case of how powerful connectomic measurements can be.
While we do still find the degree of accuracy and specificity of the predictions remarkable, we
also emphasize that it would be unrealistic to expect such a model to be correct in all its details.

We did not intend to claim that experimental measurements of neural activity are not needed.
We wholeheartedly agree that measurements and perturbations of neural activity and behavior
will be essential for going beyond the current results. And in our paper, we show an example of
how such measurements can be incorporated to refine our hypotheses [l. 197-222, Fig. 3].

We will respond to your additional questions and concerns (how important are single-neuron
recordings for selecting realistic models, how well does this generalize to other circuits,
predictions about TmY3) below.

*Specific points:*

*1. The main message of this manuscript is not a deeper understanding of visual processing in Drosophila*
*but the general notion that connectomes help (massively) to create biologically realistic network models.*
*Indeed, the authors emphasize that their "...modeling approach provides a discovery tool...". It is thus*
*important to get a good idea how the DMN approach generalizes to other brain circuits. The DMN*
*approach is likely to be facilitated by features of the Drosophila visual system such as graded synaptic*
*transmission and a highly repetitive architecture. More insight into the potential for generalization of this*
*approach would be useful (see also below).*

Thank you for appreciating our central results. Indeed we have demonstrated our method in the
fruit fly optic lobe, where until recently our understanding of circuit connectivity was most
comprehensive, with 64 cell types and their connectivity mapped. The study of the optic lobe is
supplemented by numerical experiments with MNIST. In principle, we cannot rule out the
possibility that our method will only work in the fruit fly visual system (note, however, that even
then the recent full-brain connectomes would still yield a deluge of data to apply our method
on...). But we can clarify why we believe the fly visual system will not be unique:

- 1. repetitive architecture: in our modeling, the highly repetitive architecture was exploited to
extrapolate a connectome for the whole eye. With a full connectome, the number of
unknown parameters in our model would still remain identical, and the model fitting
procedure would remain largely the same (very minor difference: convolutions would be
replaced by sparse matrix multiplications). The parameters in our model only depend on
the number of cell types, and not on their spatial organization. With full connectomes for
the fly visual system now becoming available, it will become possible to directly compare
the predictions of convolutional models to non-convolutional ones. [However, we note
that most published activity measurements also aggregate across columns, so
fine-grained validation of non-convolutional models will be challenging.] Finally, we note
that the mammalian retina is another model system for which our architecture will be
directly applicable, and for which one will also have to use repetitive structure to
extrapolate incomplete connectome reconstructions.
- 2. graded synapses: Yes, our model is based on graded synapses. However, we note that
the network equations resulting from our modeling choices result in overall dynamics
which are given by threshold linear network dynamics. Threshold linear dynamics have
been used extensively to approximate the firing rates in spiking neurons with non-graded
(quantal) synapses on a wide range of circuits, and our formalism will likewise be useful
for building connectome-constrained models of such circuits.

3. More generally, the approach of constructing a connectome constrained deep
mechanistic network is not confined to the specific single neuron and synapse model
used in this study. Indeed we believe a more expansive approach where we also search
over the space of single neuron and synapse models would be productive and enable
the inference of the best model class for each system. The main idea behind our work is
to use machine learning to constrain a model simultaneously with the connectome and a
computational task.

We agree that it is important to understand the generality of the DMN technique. We eagerly
await the mapping of more connectomes and future work modeling other circuits and other
model organisms. In the revised paper, we have provided additional clarifications and
explanations of these points in the Discussion [l. 360-371].

2. *The graded potential neurons used in this study are biologically realistic for the Drosophila visual*
*system but not for most other brain circuits. It may be more difficult to get DMNs to reproduce biological*
*mechanisms of computation with spiking neurons. This may be a limitation of the approach that should be*
*discussed (or explored, if possible).*

We agree that standard backpropagation through time is not readily applicable as a gradient
estimator for spiking neural networks, so that optimization of spiking neurons might pose
additional computational challenges. However, there has been much recent progress in the
training of spiking neural networks with surrogate gradient methods, (e.g. Zenke et al 2018,
Neftci et al 2019, Wang et al 2020) Further, reinforcement learning algorithms which have been
used to optimize robotics/physics simulations without the need for gradients are also readily
applicable to train spiking networks. We will note that, in Mi et al ICLR 2022, we already
demonstrated the training of a more complex (albeit still non-spiking) biophysical synapse
model. Therefore, while this is definitely an area where more research will be important for future
studies modeling spiking circuits, we do not think that this poses a limitation to the applicability of
the DMN approach. We have clarified this in the Discussion.

3. *The repetitive layout of the Drosophila visual system facilitates network modeling, and so does the*
*extensive knowledge of cell types in this system. How would the DMN approach be affected if knowledge*
*of the connectome were less complete, as is often the case in other brain circuits?*

The hexagonally convolutional structure of the optic lobe, and the extensive knowledge of
cell-types in it, was primarily useful for two reasons: First, it allowed us to construct a network
model even from incomplete connectomic measurements, as we were able to use the
assumption of columnarity to 'fill in' missing measurements. Second, it allowed us to validate the
model by comparing predictions of the model with the extensive literature on estimates of
single-cell selectivity. Third, the availability of cell-types reduces the number of free parameters
(if one assumes parameters to be shared across cell-types).

However, the approach could equally well be applied in a setting one has no spatial structure in
the connectome whatsoever, and possibly even not (agreed-upon) cell-types: Indeed, in Figure

6, we demonstrate (on the MNIST dataset) that single-tuning in a neural network can be
recovered from dense connectomic measurements-- this example neither has convolutional (i.e.,
columnar) structure, nor does it rely on any notion of cell types. Therefore, a repetitive structure
is not critically needed for the DMN approach. In particular, the approach can be directly applied,
e.g., to any other brain area in the fruit fly brain/VNC, or dense reconstructions of cortical tissue
in the mouse or zebrafish, or the mammalian retina (which even also follows a stereotyped
spatial arrangement).

Obviously, *some* way to link model-predictions with experimental measurements is required to
validate DMNs--- this could be a notion of cell-types (to compare model predictions to either
single-cell tuning, or aggregate statistics across many types of a cell), or a means to directly
perturb single-cells. However, this is true for any computational network model in neuroscience.

To more concretely illustrate the importance of connectomic constraints and cell-types for
scaling-up brain models with single-cell fidelity, it is useful to consider the case of a
whole-Drosophila DMN: This would have about 130,000 neurons (Lin et al 2023). Assuming an
unconstrained RNN with passive point neuron voltage dynamics and instantaneous graded
release synapses, we would have to find 16,952,040,000 connection strengths and 260,400
neuron parameters in this case (time constant and resting potential per cell). With the
approximately 4,200 cell types (Schlegel et al 2023) reported from the connectome and
approximately 30,435 cell-type to cell-type connections that are conserved across the
hemispheres, our DMN approach would reduce the number of free parameters in a
whole-Drosophila DMN to approximately 38,835 (time constant and resting potential for each
cell type and 30,435 synapse count scaling factors). I.e. only 0.00023% of the number of
parameters that need to be estimated in the naive setting without the connectome-- we posit the
DMNs will make it possible to build whole-brain models of behavior from connectomes.

We tested these ideas explicitly and now address this question in Fig 2d and in the Results [I.
171-191].

*How would it be affected if cells were divided into fewer distinct types?*

The DMN approach is independent of the number of distinct cell types. Cell types facilitate the
understanding of neural circuits in general, and our approach leverages cell types to reduce the
number of parameters in our model, by assuming that cells of the same type share the same
parameters. If there are fewer cell types, we would thus have fewer (free) parameters --
conversely, if there are more cell types (or cells which can not be assigned to any type, and
therefore need to have their own parameters) we would have more parameters. Of course, the
degree to which this assumption (cells of the same type share parameters) is an empirical
question which might have different answers for different model systems. Note that, in Figure 6,
we demonstrate a setting in which we do not have any cell types at all and still derive accurate
neural tuning predictions due to the sparse nature of synaptic connectivity.

In the future, it is likely that a deeper understanding of the function of specific neurotransmitters,
receptors, and synaptic morphology, combined with the ability to infer such synaptic functional
parameters from electron microscopy (Eckstein 2020) will enable us to share synaptic
parameters based on these, rather than cell types.

*4. An interesting observation is that the same neurons in different networks do not always show the same*
*functional properties and form discrete clusters in functional space. For T4c cells, for example, 3 clusters*
*were found but only one represents biologically realistic neurons with correct motion sensitivity. This*
*observation is interesting because it can, in principle, be a starting point to explore general principles of*
*network design. On the other hand, it means that the DMN approach alone is not sufficient to predict*
*mechanisms of computation, even in this well-established system. Additional knowledge (here: true motion*
*sensitivity of T4 cells) is necessary to distinguish biologically realistic from unrealistic networks. Such*
*knowledge may be hard to come by in other systems. This is a (potentially serious) limitation of the DMN*
*approach that needs to be discussed more. What type of additional information would be most useful to*
*resolve “cluster ambiguities”?*

We agree that connectome+task-constraints, by themselves, are unlikely to be able to identify a
*unique* model in general. We should clarify, we deliberately did not utilize the “well-established”
nature of this circuit in constructing the model, since we did not use the extensive neural activity
measurements in the model construction. In that sense, our work attempts to show how far we
can get with as little experimentation in the living animal as possible. Indeed, it is very easy to
construct DMN models to be additionally constrained with neural activity and perturbation
experiments, as well as measurements of behavior. We propose that connectome constrained
DMN models should serve as an integral part of the hypothesis-experiment loop. As we show in
Fig 3, DMN ensemble hypotheses can be used to suggest targeted experiments distinguishing
between model classes. And these new experiments would then be used to refine hypotheses
via new DMN models constrained with all available measurements of connectivity, neural
activity, and behavior.

We clarify that many predictions, for instance the contrast preferences of most neurons, are
correctly predicted without the need for further experimental measurements. In particular, all
analyses in Figure 2 [prediction of on/off tuning on flash responses, direction selectivity] and
Figure 4 [DMNs largely recapitulate known mechanisms of motion computation], are across the
entire ensemble of networks, and are not conditioned on the true motion sensitivity of T4 cells.

We do use true contrast preferences to show (in Figure 35) how to deal with cases in which the
ensemble is not uniquely constrained, and show (arguably remarkably) how these constraints
do identify a small number of highly specific and experimentally testable hypotheses. The goal
of our analysis was to highlight how our general approach which generates multiple hypotheses
can be combined with further experimental measurements. The DMNs demonstrate that these
constraints lead to a handful of highly specific and experimentally testable predictions for neural
tuning: One single tuning measurement (in this case T4c) is sufficient to identify the correct
cluster, and thereby to also constrain the selectivity of the other cells in the circuits. [Even for
T4c, the hypothesis cluster with the ‘correct’ tuning actually has the best average

task-performance, which is used in Figure 4]. We use a similar approach in Figure 5, to narrow
down predictions for the tuning of TmY3-- while, across the ensemble, there is a variety of
predictions for TmY3, selecting only ensemble-members with the correct T4c tuning leads to
clear predictions for TmY3.

We do not agree that the inability to perform experimental measurements of neural activity in a
given circuit is a "(potentially serious) limitation of the DMN approach". With the connectome and
our DMN approach, at least we can make a small number of hypotheses for such a circuit [as
demonstrated in Figures 3 and 5]. In contrast, without the connectome+DMN we would have no
hypotheses at all. Obviously, the issue of not being able to make physiological measurements is
unrelated to DMNs---in the end measurements will always be useful or even critical, but we do
expect that DMNs will be a very powerful way to dramatically reduce the number of
neurophysiological measurements needed to characterize complex neural systems.

We have clarified this point in the Discussion [l. 381-388].

*5. The authors emphasize that DMNs can reproduce biological mechanisms of computation, but they do*
*not go very far in analyzing computational mechanisms beyond current knowledge. So far, they mainly*
*asked whether known mechanisms are reproduced in the DMN. For example, motion sensitivity of T4 cells*
*involves direction-dependent temporal shifts between excitatory and inhibitory input currents, which is*
*reproduced by the model. However, the authors could go further and manipulate specific connections to*
*verify that motion sensitivity is indeed generated by the expected implementation of a computational*
*strategy (combination of Hassenstein-Reichardt and Barlow-Levick models) in neural circuitry. Such an*
*analysis should have potential to uncover novel, unknown functions. Similarly, they could use specific*
*manipulations of connectivity to analyze the mechanisms of motion sensitivity in TmY3, following up on*
*the speculations put forward in the text. Generally, the ability to manipulate connections in a biologically*
*realistic simulation has interesting potential because this is often not possible experimentally.*

We are grateful for this suggestion. Based on your suggestion, we have now added extensive
new analyses and a completely new section that explores these questions: We have further
studied the circuit mechanism of direction selectivity in T4, T5, and TmY3 neurons through a
combination of techniques and indeed find a combination of preferred direction enhancement
(Hassenstein-Reichardt) and null direction suppression (Barlow-Levick) mechanisms for T4, T5,
and TmY3.

In our mechanistic model, we can inspect the input current contributions from each cell type and
study the differences in these contributions for motion stimuli in the preferred and null directions.
This allows us to directly inspect the contribution of neurons to a computation without needing to
perturb the circuit through inactivation [however, we also report predictions for inactivation
experiments below, see below]. For T4 neurons, we observe null direction suppression mediated
by inhibition from Mi4, and also significant enhancement of coherent visual motion mediated by
excitatory T4 to T4 connectivity, attributing a role for this lateral connectivity. For T5 neurons, we
observe null direction suppression mediated by CT1 inhibition and consistent excitatory input

from neighboring T5 to T5 neurons and by excitatory input from Tm9. For TmY3, we see null
 direction suppression predominantly via Mi14 and Mi4 inhibition and excitation via Mi1 and L5.

We summarize these findings now in the main manuscript by adding text in the Results [I.
 237-249], Fig 4b showing input current contributions of T4c. And Extended Data Figs 4, 5, 6
 showing detailed input current analysis for preferred and null direction stimuli for all T4 and T5
 subtypes, and for TmY3. Extended Data Fig 4 showing input current analysis for T4 subtypes is
 copied below.

In addition, we now simulated the requested silencing of T4 inputs, and compared model
 predictions to experimental results reported in Strother et al, Neuron, 2017, finding good
 agreement for most effects, as shown below. For these results, we averaged model-predictions
 over all models in the model-cluster with correct T4c tuning (which is also the best
 task-performing cluster). The two columns in panel a below are direct copies of Strother et al
 2017 Figures 3b [left column] and 3c [right column], and panel b shows, for comparison, the
 output of our analyses. We emphasize that, for these analyses, only a qualitative comparison of
 the effect is meaningful-- for example, Strother et al report Delta F/F from calcium imaging,
 whereas we show standardized voltage responses. In addition, our 'silencing' analyses are
 based on simply clamping the respective T4c inputs to 0, which is likely a crude approximation
 of the effect of blocking synaptic transmission with shibire(ts1) and temperature increases.
 Nevertheless, the models correctly capture that: i) Removal of Mi1 excitation removes the T4c
 response ii) Tm3: Removal of Tm3 excitation decreases, but does not remove, the T4c
 response. iii) Mi4: Removal of Mi4 does not remove the T4c response [it does, however, lead to
 a prolonged response in the network model which is not observed experimentally] iv). Mi9: no
 effect on T4c.

6. The authors suggest that TmY3 is a novel motion-sensitive neuron that has not been recognized
previously and computes motion independently from T4 and T5 cells. It is also predicted that other
neurons should be motion sensitive (TmY4, TmY18), probably because they receive input from T4 and T5.
So far, these predictions have not been tested experimentally. Doing so could substantially enhance the
impact of this study.

This study has been conducted by computational labs enabled by the availability of this rich and
large dataset. We agree that experimental validation of this prediction would be very exciting--
but as computational labs, we do not have direct access to experimental resources, and are
therefore eagerly awaiting whether this prediction will bear out. Nonetheless, we have been
informed by our colleague Michael Reiser, that unpublished work in progress in his lab does
indeed hint at motion selectivity for TmY3. He says the whole picture for this cell type is more
interesting and complex and they plan to report the results of their study in the near future.

In the meantime, we do wish to emphasize that no neural activity measurements were used in
constructing our model, therefore we have already tested the predictions of our model against
experimental measurements of neural activity across 26 studies. Furthermore, we provide a
large supplement with hundreds of pages worth of experimentally testable predictions, of which
this is but one prediction. We do hope that these predictions, and the fact that we are willing to
publish them and ask the community to verify or refute them, already strongly speak to the utility
and realism of our model approach.

7. Abstract: "...we show that with only measurements of the connectivity of a biological neural network,
we can predict the neural activity underlying neural computation". This statement is not correct. DMNs
also use knowledge of the computation (input-output) for training, and additional knowledge is required
for model selection (for example activity/tuning of T4/T5 cells).

We apologize for the confusion. We meant to highlight the fact that our model was constructed
using experimental measurements of only connectivity, and not also neural activity, etc. Indeed,
knowledge of the computation was also necessary. We have now updated the sentence to read
"We show that with **experimental measurements of only the connectivity** of a biological
neural network, we can predict the neural activity underlying a **specified** neural computation."

8. How good is the optic flow detection achieved by DMNs? The quantification by the error measure is not
very intuitive. It shows that training improves performance, but unconstrained CNNs still achieve much
better performance than trained DMNs. It would be good to provide more information to get a better
intuition how well a DMN is performing in comparison to a real fly.

We agree that comparisons of optic flow estimation between the model and the real fly would be
interesting. One challenge, however, is that our model is focused on computation of local motion
(in particular, the model does not include LPTCs which spatially integrate local motion signals),
whereas experimental characterization (e.g. optomotor responses) has focused on *global*
computation of motion by the real fly on relatively simple stimuli (moving gratings). Indeed, the

qualitative tuning of behavioral responses depends strongly on the actual behavioral task (see
e.g. Creamer et al 2018).

Please note that the goal of task optimization is simply to constrain the parameters of the
connectome network enough to make good predictions of neural activity. The accuracy of neural
activity predictions is our main goal and the absolute accuracy of the optic flow estimation is not
as relevant. Indeed, because we use a blackbox motion decoder network to predict optic flow,
the nature of this decoder could lead to better or worse detection of optic flow compared to the
real fly, even if the resulting connectome network predicted neural activity with perfect accuracy.

*Minor comments:*

*9. Ln 481: Fig 1g*

Thank you, fixed.

**Referee #2 (Remarks to the Author):**

*The authors use connectome data from the fly visual system combined with optic flow training to produce*
*a task-performing mechanistic model with interpretable parameters. Comparing to previous data, the*
*model captures many of the tuning properties of fly visual neurons. There has been some previous work*
*that uses connection data to define model architectures for task-training. The authors may want to cite*
*some of this work from C. elegans (e.g. <https://www.ncbi.nlm.nih.gov/pmc/articles/PMC8253844/> and*
*<https://arxiv.org/pdf/2201.05242.pdf>). The present work is novel in my opinion in the extent to which it*
*compares the neural parameters of the trained model to data.*

Thank you for your comments and appreciating the novel contributions of our work. We agree
that the two studies the reviewer mentioned are relevant work, and have now cited them in the
revised manuscript [1. 102]. At the same time, we do want to emphasize that our work goes
substantially beyond these two studies: Sakamoto et al trains a model of 69 motor cells and 95
muscle cells to reproduce realistic locomotion patterns in C elegans, and shows that a
connectome-constrained network can be successfully trained to solve this task. Bhattasali et al
train a neural network architecture inspired by C elegans locomotion circuits, and analyzes the
properties of the resulting networks (e.g. in terms of inductive bias).

However, *neither* of these two studies perform any comparison of the neural activity predicted by
the model with experimental measurements. Thus, the primary contribution of our paper--
namely, that task-trained connectome-constrained models can predict neural activity remarkably
well-- is a clear advance over both of these studies. We achieved these results in a vastly more
complex model system (fruit fly visual system vs. C elegans), which also required us to address
substantial engineering challenges (e.g., implementing differentiable simulations of convolutional
recurrent networks defined on hexagonal grids).

*It would probably be good to provide a contextualization of the DMN task performance and parameter*
*recovery by pulling in some more of the supplementary results (at least more quantified descriptions of*
*them in the text). For example, the main text does not convey how relatively minor the enhancement in*
*performance on the task is for the DMN vs random model, in the context of how well the unconstrained*
*CNN can perform. Also, it seems relevant to note that the random models are still positively correlated*
*with cell tuning and the flash response results can be captured by even the poorly performing DMNs.*

Thank you for this excellent suggestion which has helped us to considerably expand our study--
You are raising an important point about how (quantitatively) good different models are at
predicting different tuning properties, and how specific constraints [cell connectivity, synapse
counts, synapse signs] contribute to the prediction of tuning properties.

Your comment (and a related comment by reviewer 4, see below) inspired us to perform a large
set of additional experiments to tackle this question more systematically. Briefly, we constructed
a set of 9 different model ensembles for which we systematically varied which parameters were
constrained by connectomic measurements, or which were set by task-optimization. We find
that:

1. The preferred contrast as measured by flash response index (FRI) is generally well
predicted across model ensembles provided with connectome derived connectivity at
the cell-type resolution and synapse signs. Thus, cell-type connectivity and synapse
signs are crucial, but all other parameters --- including synapse counts and single cell
parameters --- can be randomized or task optimized, and are *not* critical for predicting
FRIs.

2. DSI: Accurate predictions of the direction selectivity index (DSI) requires
cell-connectivity. However, it does *not require synapse counts or synapse signs*.

3. Preferred directions: To achieve the correct cardinal direction tuning, we need *all*
connectome constraints [i.e., all of cell-connectivity, synapse counts, synapse signs].

Thus, these results are still entirely consistent with our overall findings, but provide a detailed
picture of the relative importance of different constraints. We are very grateful for your
suggestion which we believe to have considerably strengthened the paper. A summary of these
additional results has now been added to Figure 2 of the main paper (new Figure 2d, see
below), a brief description of these results is now described in the corresponding section in
Results [l. 171-191], and a new Extended Data figure 9 (see below) includes a full summary of
results.

Your second question as to why unconstrained CNNs outperform the DMNs-- we do believe that
this is simply because of the flexibility of the unconstrained network, as they have 414,602 free
parameters to transform the naturalistic movie sequences to pixel-wise motion (instead of only
734 free parameters plus 7,427 decoder parameters for the DMNs). This is consistent with
general findings in deep learning that, in most settings, task-training models with more
parameters typically leads DMN to better performance, given enough training data.

*Are the T5 off-motion selective neurons in 2c supposed to be tuned to on-motion as well? Or is this a way*
*in which the model does not fit the data? This should be spoken to in the paper.*

We have clarified this in the updated version of the manuscript in the Results [l. 231-236] and
added Extended Data Fig 13. Comparing the relative strength of direction selectivity to edges of
ON vs OFF contrast, we find that all four T4 subtypes are correctly predicted to respond more
strongly to ON edges than OFF, and three out of four T5 subtypes are correctly predicted to
respond more strongly to OFF than ON edges (Extended Data Fig. 13).

Across our model ensemble, we find that T5 neurons are predicted to be off-motion selective in
more models than on-motion selective (p=0.0009). However, we do find substantial variability in
the prediction for T5 on-motion selectivity. When we also filter the ensemble to only select
models which exhibit the correct contrast-tuning [to flashes] *and* the correct preferred directions,
then these models only predict very weak on-motion tuning for T5 cells. For this selection of
models, we show below the model responses to motion in the preferred contrast vs
non-preferred/null contrast.

I have several concerns/questions regarding the synthetic connectome experiments in the MNIST-trained
networks. First, I do not understand the motivation behind the version with noisy weight estimates where
getting the correct weights is baked into the objective function. What do we learn from seeing that a
network initialized with roughly the correct weights can explicitly learn to recover those weights
(regardless of sparsity)?

*For the connectome-only version, this still does not seem to necessarily support what the authors seem to*
*be claiming about it. Specifically in the absence of any information that leads to unique cell IDs,*
*comparison of tuning across networks is meaningless. With a sparse network (including the fly*
*connectome) the pattern of connections a cell makes can be a unique identifier for it, and therefore these*
*cells can be labeled as the same and their tuning can be compared across networks. Such unique identities*
*are not possible in densely connected networks. Therefore, the tuning comparisons done here are*
*essentially as if two random neurons were picked across models and expected to have the same tuning.*
*The fact that a random pairing of neurons does not display the same tuning does not mean these networks*
*are not learning the same mechanisms. In fact, ED Fig 9a shows that Dale's law helps with the*
*correlation, which is likely because having a weight constraint offers some kind of (weakly) unique*
*identifier. Also, how were the signs decided for units in these networks?*

Thank you for pointing out that this section was difficult to follow. We have rewritten this whole
section [l. 313-350] to clarify the motivation and the results. We also clarify below:

When might connectome constrained and task-optimized DMN models accurately predict neural
activity? Sparse connectivity is a hallmark of biological neural circuits, and in this section, we
ask whether sparse connectivity enables DMN models to make accurate predictions of neural
activity. For sparsely connected circuits---assuming the connectome is known---there are fewer
synapse parameters left to estimate using task-optimization. We hypothesized that such
networks might support fewer possible mechanisms by which to perform a given task, compared
to more densely connected circuits, and so a task-optimized DMN model is more likely to find
the true mechanism and accurately predict neural activity.

We addressed this hypothesis in simulation, by constructing feedforward artificial neural
networks solving the classic MNIST handwritten digit classification tasks (Fig 6a). These
networks had varying degrees of sparse connectivity, and random assignment of neurons as
excitatory and inhibitory respecting Dale's law (25 groundtruth networks for each sparsity level,
Methods). We simulated the process of making connectomic measurements from these
groundtruth networks, and used those measurements to build connectome-constrained
task-optimized DMN simulations of each groundtruth network. Since there is still uncertainty
about the degree to which connection strength can be inferred from noisy connectomic
measurements of synapse count, we simulated two settings. First, that connectomic
measurements reveal connectivity but not connection strength. Or second, that connectomic
measurements reveal connectivity and additionally a *noisy* estimate of strength. We then asked
how well each task-optimized DMN simulation predicted the neural activity of its corresponding
groundtruth network, as a function of the sparseness of the connectivity.

When connectivity is assumed to be known but not connection strength, DMN simulations were
connectome-constrained and task-optimized to estimate both the resting membrane potential of
each neuron, as well as the connection strength of each pairs of neurons (provided they were
connected in the in the corresponding groundtruth network, all other connections were kept at
zero). When connectomic measurements can be assumed to also provide noisy estimates of
strengths, task-optimization was used to **denoise the noisy estimates**: We used the noisy

estimates as a prior on the strength of each connection, to regularize the task-optimized
connection strength **towards the noisy measurement**.

Consistent with our hypothesis, we found that when connection strengths cannot be inferred
from connectomic measurements, sparsity in the connectome greatly improves the correlation of
neural activity between a groundtruth network and its DMN simulation at the single neuron level
(Fig 6b, median Pearson correlations of 0.85 for 10% connectivity vs 0.38 for 80% connectivity,
100 randomly selected neurons from 25 randomly generated groundtruth networks). Conversely,
when noisy estimates of connection strengths can additionally be inferred from the connectome,
we find that DMN predictions of neural activity correlate well independent of connection sparsity
(median Pearson correlation >0.9 for all connectivities).

In our fly visual system model, we assumed an intermediate regime: Pairs of connected
neurons of the same pre- and post-synaptic cell type likely express the same neurotransmitter
and receptor combination leading to a common unitary synapse strength for all such
connections. So our model assumes the synapse count reveals the relative magnitude of
connection strength between all such connections. However, our model assumes that absolute
connection strength is unknown from the connectome, since the strength of a unitary synapse
might vary across cell types expressing different neurotransmitters and receptors. In other
words, 5 synapses between Mi1 and T4 neurons could be stronger or weaker than 5 synapses
between Mi9 and T4 neurons, but 5 synapses between Mi1 and T4 neurons is assumed to be
exactly half as strong as 10 synapses between Mi1 and T4 neurons.

Finally, we agree with your point about the correspondence of neurons between groundtruth
networks and their DMN simulations. In our modeling, we assumed that neurons can be
uniquely identified and put in correspondence. Experimentally, morphology and gene expression
genes are frequently used in addition to connectivity, in order to identify neurons uniquely.
However, if we restrict ourselves to identifying and corresponding neurons solely based on
connectivity, this is still possible, except in the extreme case of dense all-to-all 100%
connectivity, since all neurons will have the same connectivity profiles. For simplicity, we now
drop this last data point in our figure, which is not needed to show the overall trend that
sparseness improves the accuracy of DMN predictions of neural activity.

*Clarifications:*

*Can the authors better explain the differentiation between synapse count and the scalar? It seems the*
*scalar and the count are held constant for all pairs of cells with the same pre- and post-synaptic cell type,*
*so what extra information does having the count as part of the equation provide?*

For each pair of cells i and j , the corresponding filter weight w_{ij} is determined by three factors:
 The synapse sign σ_{t_i, t_j} and the scaling coefficient α_{t_i, t_j} which are indeed the same
 for each pair of pre- and post-synaptic cells. The third contribution comes from the synapse
 counts-- however, and importantly, the synapse counts do not only depend on the cell-types, but
 also on the relative spatial offsets of the two specific cells. This is denoted by the coordinate
 subscripts $\Delta u = u_i - u_j$, $\Delta v = v_i - v_j$ which are the relative columnar offsets of a
 postsynaptic target cell i of type t_i and a presynaptic source cell j of type t_j in the
 two-dimensional hexagonal coordinate system.

Thus, the synapse counts determine the *shape* of the filter [as visualized in Figure 1e,
 copy-pasted above for easy reference], whereas the scalar scales the overall strength of the
 filter. In total, the model is based on 2355 (non-zero) synapse counts, and 604 (non-zero)
 connection-filters, and hence there are 604 scalars that are trained. We have updated the
 description of the model in Results [l. 81-101] to clarify this.

*How do the 50 ensemble models differ? Just different draws from the same distribution of resting*
 *potentials?*

The 50 models are initialized at a random location in the parameter space and also differ in the
 stochastic optimization. This includes the order by which samples are drawn from the Sintel
 dataset and their randomized augmentation (random flips, rotations, pixel-wise gaussian noise,
 random contrast and brightness). We have updated the description of the model in Methods [l.
 491-493 and 580-588] to clarify this.

**Referee #4 (Remarks to the Author):**

*Lappalainen et al. build an optic lobe connectome-constrained neural network called a task-optimized*
*deep mechanistic network (DMN), and optimize for a computation performed by that biological circuit*
*(motion detection). They show that constraining both the connectivity and computational task reproduces*
*some of the experimentally determined tuning of specific neurons, and makes predictions about the tuning*
*properties of other neurons in the network that have yet to be experimentally measured. Finally, the paper*
*argues that, for sparse networks (such as some biological neural networks), knowledge of the connectivity,*
*signs of connections, and an estimate of connection strength may be sufficient to predict the mechanism by*
*which the circuit performs a known computational task.*

*I am in general enthusiastic about the study - it is a useful simulation of a portion of real, complex neural*
*network (64 cell types and 721 columns, plus 1 inhibitory cell type that extends across columns, but*
*missing all of the feedback and neuromodulatory connections) and shows how connectivity shapes many*
*of the known properties of a neural network - the optic lobe is an ideal test case as its cell types have been*
*studied extensively over the past 60 years. Figures 3 and 4 in particular are quite nice - 1) comparisons*
*between the best performing model's T4/T5 cells and the response properties of their inputs to known*
*tuning curves and responses (although I have concerns about some of the details - see below) and 2)*
*comparisons between different models to understand what properties of particular cell types define the*
*best performing models. This is a nice study that will form the basis for simulations of larger biological*
*networks, for fly, and other species. However, I have several concerns about the modeling, model*
*predictions, and interpretations of the results that should be addressed.*

Thank you for enthusiastic support of our work, and for your constructive suggestions which
have helped us to substantially strengthen our work. Based on your suggestions, we have now
exhaustively detailed how we constructed the DMNs from connectomic data with new tables and
descriptions [Supplementary Information, Supplementary Data: connectome_constructions.csv,
connectome_construction_merge_fib19_fib25.json]. We have also provided extensive new
analyses to characterize which aspects of the connectomic data (e.g., cell-type connectivity,
synapse counts, synapse signs) are important for achieving a close match between model
predictions and neural activity measurements [new main paper Fig. 2d, Extended Data Fig. 9],
described in detail in the response to R2. In addition, we performed additional analyses of the
L1-5 cell types. Additionally, we characterized the input asymmetries and their relationship to the
predicted direction selectivity indices [Extended Data Fig. 2].

*The authors build a hybrid optic lobe connectome from several different datasets - the choices they made*
*in how to combine these datasets must be made transparent in the paper. Ideally, they would present a*
*Supplemental Figure devoted to how this was done and how they handled any discrepancies or differences*
*between the datasets. If there were no discrepancies or differences this should also be explained. It is*
*difficult to interpret the findings from the model without a thorough understanding of how the*
*connectome-constrained model was built.*

*Related to this point, I assume the signs of connections were taken from the literature? Can the authors*
*provide citations for these (and can they compare against the same cell type in the open FlyWire/FAFB*

whole-brain connectome dataset, for which neurotransmitter predictions (from Eckstein et al.) are
available in the optic lobe)?

Thank you for pointing out this omission. We have now included extensive tables documenting
the source of the connectivity, signs, and neural activity measurements for each cell type in the
Supplement. A description of the algorithm used to generate the filters (spatial connectivity
between cell types) is also given in the Supplement.

Regarding the comparison to the neurotransmitter predictions from Eckstein et al.: the signs of
our connectivity are not computational predictions but actually based on direct experimental
measurements of the neurotransmitter and receptors expressed by each cell type, as given by
Davis et al. 2020. We have now clarified this in the Results [l. 89-90] and Methods [l. 455-464].

*Related to the point above, the authors perform a sort of normalization step with the data so that they can
model every column identically (making sure synapse numbers are the same in each column) - this ignores
heterogeneity across columns (that might be important for motion detection). Can the authors provide
more detail on how this simplification deviates from the actual connectivity (how much heterogeneity is
there across columns?) and show how this choice affects modeling results (if they incorporate some of the
heterogeneity into the model, how do the results change)?*

At present, the best investigation of synapse count variability across columns in the fly optic lobe
comes from only a local analysis of 7 columns in Takemura 2015, showing a variability of
synapse count of roughly 10% between neighboring columns. It is also known that the color and
polarization pathways introduce larger-scale deviations in the presence or absence of neurons in
a perfect hexagonal lattice beginning with the random organization of the pale and yellow
photoreceptors and the dorsal rim specialization of polarization sensitive photoreceptors (Kind et
al 2021). More comprehensive studies of variability across the entire optic lobe are a substantial
undertaking and await the full completion and analysis of Flywire and Janelia optic lobe
connectomes.

We agree that with these new datasets, it will be possible to explicitly study the importance of
spatial/retinotopic deviations from the simplifying assumption of perfect homogeneity of
connectivity across the connectome which was essential for the present study. It is worth
pointing out that not only our model, but also the *experimental literature* characterizing the
neural activity patterns of each cell type, which we use as validation, invariably averages across
neurons. Thus both our model and experiment make the same assumption of spatial
homogeneity.

*The manuscript claims that the actual OL connectivity was critical. Extended Data Figure 3 shows that a
task-optimized DMN outperforms a DMN with random parameters. But what is missing is a
demonstration that the specific connectivity of the fly visual system is what enables optimal performance,
rather than a generic neural network with the same level of sparseness and gross connectivity statistics of
the biological network. How would a task-optimized artificially generated network (constrained by*

*biological connectivity statistics, rather than the exact connectivity of the fly visual system) perform*
*compared to the task-optimized DMN and the random DMN?*

Thank you for this suggestion. In response to your comment and a similar issue raised by
Reviewer 2, we performed a large set of experiments to characterize the importance of different
aspects of the connectivity [cellular resolution connectivity, synapse counts, synapse signs] and
how they relate to both task performance and accuracy of predicting neural activity, (for details
also see our reply to referee 2 above) and added Fig 2d and Extended Data Fig 9 and to the text
in Results [l. 171-191]. This analysis based on your comment substantially clarifies our
understanding of the importance of connectomic constraints as well as task optimization.

In summary, we found that both task optimization and detailed connectomic measurements at
the single neuron resolution were critical to prediction of preferred contrast of the 32
characterized cell types, and preferred direction of motion for the T4 and T5 subtypes, at the
single neuron resolution (Fig 2d, Extended Data Fig 9). A DMN model ensemble with full
connectomic constraints but no task optimization (randomized unknown single cell and synapse
parameters) led to accurate predictions of preferred contrast, but poor predictions of direction
selectivity and preferred direction. Task-optimized DMN ensembles which assumed the
absence of full connectomic measurements of single cell resolution connectivity, synapse
counts, or synapse signs, and therefore used task optimization to infer any of these parameters
struggled to correctly predict the preferred direction of motion. However, accurate predictions of
the direction selectivity --- but not preferred direction --- could be achieved with measurements
of cell-connectivity without synapse counts (Extended Data Fig 9c). This demonstrates the
importance of both detailed connectomic measurements and task optimization to achieve the
best predictions of neural activity.

Regarding the high-level task performance of differently constrained models: consistent with
general findings in deep learning, we find that task-training models with more parameters
typically leads to better asymptotic performance.

In addition to the task-performance of the unconstrained CNN with 414,602 free parameters, we
now show task-performances for the other DMN variations we describe in Figure 2d and in
Extended Data Figure 9, showing e.g. that an unconstrained DMN (only constrained by cell-type
connectivity) with 11,593 free parameters (+ 7427 decoder parameters) performs only marginally
better than the DMN constrained by all known connectomic-constraints with only 734 free
parameters (+ 7427 decoder parameters).

While gradient-based methods on unconstrained networks find better solutions in terms of
task-performance, our comparison suggests that the full connectomic constraints of the optic
lobe provide a good structural initialization for the high-level task and for predicting
single-neuron tuning properties.

*Neurons are modeled with leaky linear non-spiking voltage dynamics and as point-neurons with a single*
*electrical compartment. The authors show abundant tuning/response data for each model cell type, but*
*they should compare the detailed temporal dynamics and delays (critical for motion detection) of model*
*responses to real recordings of these same neurons (if this is present somewhere in the supplement,*
*apologies if I missed it).*

*Many optic lobe neurons have been recorded via Ca⁺⁺ imaging, voltage imaging, or electrophysiology*
*(e.g., compare with published responses in Behnia et al. Nature 2014 or Yang et al. Cell 2016).*

*The L1 and L2 neurons are categorized as “known OFF selective” (Fig. 2b), but this deviates from my*
*reading of the literature - shouldn't they show similar responses to both on and off flashes?*

*Also, the responses of L1 and L2 in Fig. 3e are shown as monophasic and producing an off response -*
*shouldn't they be biphasic (in contrast with the responses for L3 and L4, which do look biphasic)? These*
*potential mismatches between the literature and model results have me concerned that the model is not*
*producing the expected responses for cell types that have been extensively studied (like L1 and L2).*

Thank you for pointing out these details. We indeed agree that it is important to validate our
model predictions for the function of the critical L1-L5 visual neurons.

First, we apologize for being unclear in our terminology. With “known OFF selective”, we meant
to say that cells of these cell types depolarize in response to a light-decrement (and
hyperpolarize in response to a light-increment) which is in agreement with the literature [Reiff
2010, Clark 2011, Freifeld 2013, Strother 2014, Fisher 2015, Yang 2016, Drews 2020, Matulis
2020, Kettkar 2022]. We clarified our terminology in the manuscript [l. 142-143].

Regarding your second question: thank you very much for pointing out this fine-grained
mismatch between the model presented in Fig. 4e (former Fig. 3e) and the literature. We found
that other models in the ensemble correctly capture L1 and L2's biphasic tuning. You helped us
realize the limitation of grounding our current Figure 4 only on the single model with the best
performance on the optic flow task. Instead, now we updated Figure 4 to show average tunings
from the best-task-performing cluster of models from the ensemble of 50 models for each of
these cell types. We found that the average response of models from this task optimal cluster
consistently provide better predictions.

We have included a new version of Figure 4e with additional L1-L5 tuning predictions from our
model: L1 and L2 are indeed correctly predicted as biphasic on average and across a range of
single-ommatidium flash durations of both ON and OFF contrast (see below).

We believe that this substantially strengthens the validity of our model predictions and our
claims. We emphasize once more that our models are not constrained on any neural activity
measurements (but only on the connectome and the optic flow task) -- thus while we do not
expect that they capture all detailed tuning properties, they nevertheless yield remarkably
accurate predictions for single-cell tuning.

e Predicted flash responses of motion detector input and lamina neurons

*Given the constraints provided by the connectome, there are only 734 free parameters in the model*
 *(resting membrane potential for each cell type (65) and unitary synapse strength (604)) - but the authors*
 *could have also included the synapse NL as a free parameter (varying across synapses) - how would this*
 *have affected modeling results?*

*Is there a reason they need to limit to ~700 free parameters? The question has to do with the choice of*
 *which parameters to fix across the model and which to vary - how much do these choices affect results?*

In this work, we focused on building the simplest possible model consistent with measured
 connectomes, and evaluating to what extent it is sufficient to account for neural tuning. We
 agree that our framework would be the perfect testing ground to explore a larger class of
 mechanistic single neuron and synapse models and this would be exciting future work.

Our deliberate choice of the threshold-nonlinearity (ReLU) reduces the number of possible
 parameters because it is scale invariant. Rescaling the nonlinearity in this model is equivalent to
 rescaling the unitary synapse strength parameter α , and shifting the nonlinearity
 corresponds to changing the resting membrane potential. For this reason we did not add
 additional parameters to the nonlinearity. But for other nonlinear functions, additional parameters
 could indeed be explored.

Relevant to this question, we also characterized the number of parameters in the models we
 described above, and in Fig 2d and Extended Data Fig. 9 to characterize the relationship
 between connectome-constraints and task-optimization — which includes models with different
 parametrization of synapses and their numbers of free parameters. We characterize in detail
 how much the choices of these constraints affect the results above. We do emphasize that our
 approach is not computationally limited to ~700 free parameters (and this is also shown in these
 additional analyses in which we optimize over much bigger parameter sets), but a modeling
 choice.

*The authors optimize the model network to perform a particular motion vision task - this makes a lot of*
*sense given that the major function of the optic lobe is to detect motion, but it is not clear how the results*
*depend on this specific task - this needs to be addressed. What differences might they observe or expect if*
*the network was optimized to perform a different task that the fly optic lobe mediates (for example, color*
*or shape detection) or a specific fly behavioral task (like the optomotor response) - how would optimizing*
*for a different task change the responses properties of neurons in the network? Addressing this question is*
*critical for understanding the constraints of the connectome.*

We focused on a local motion detection task since the connectome available to us largely
focused on reconstructing the circuitry of local motion processing the optic lobe. We agree that
an exciting future direction would be to investigate and reverse engineer the panel of tasks for
which the circuitry of the optic lobe is optimally evolved over evolutionary time-scales. As you
suggest, task optimization across a large variety of behaviorally relevant tasks will likely be
critical for constructing accurate future models of the entire optic lobe, which must support all
visually guided behavior.

*Model responses to moving edges in Fig. 2c: Why do many of the models show that T5 is responsive to ON*
*moving edges (which it is not)? Could this be expanded upon in Figure 4 (that compares different*
*models)?*

In our model ensemble, we find that T5 neurons are more often and more strongly predicted to
be off-motion ($p=0.0009$). However, it is the case that across the entire ensemble, we do find
substantial variability in the prediction for T5 on-motion selectivity. When we also filter the
ensemble to only select models which exhibit the correct contrast-tuning [to flashes] *and* the
correct preferred directions, then these models only predict very weak on-motion tuning for T5
cells (see details in the Figure below, which is replicated here again). Thus, while the ensemble
as a whole is not inconsistent with the data, there is considerable variability. We have clarified
this in the Results [l. 231-236].

I'm concerned about claims that the mechanism of motion detection matches experimental findings: The
 mechanism of how direction selectivity in T4 and T5 cells emerges is still debated. Haag et al. suggest
 there is both PD enhancement and ND suppression; Gruntman et al. argue for ND suppression only;
 Wienecke et al. argue for neither. It seems likely that the mechanism is different for the ON and OFF
 pathways. The manuscript shows that the tuning of T4 and T5 cells and their inputs (in some models)
 qualitatively matches the experimentally measured tuning, but can the authors clarify which mechanism of
 direction selectivity is matched/favored by their models?

Thank you for this suggestion. We have further studied the circuit mechanism of direction
selectivity in T4, T5, and TmY3 neurons through a combination of techniques and indeed
predominantly find null direction suppression (Barlow-Levick) mechanisms for T4, T5, and TmY3
but also enhancement of coherent motion across the visual field through lateral interactions
between T4 and T5 neurons of the same subtype.

In our mechanistic model, we can inspect the input current contributions from each cell type and
study the differences in these contributions for motion stimuli in the preferred and null directions.
This allows us to directly inspect the contribution of neurons to a computation without needing to
perturb the circuit through inactivation. For T4 neurons, we observe null direction suppression
mediated by inhibition from Mi4, and also surprisingly significant excitatory T4 to T4 input
attributing a role for this connectivity for the first time. For T5 neurons, we observe null direction
suppression mediated by CT1 inhibition and consistent excitatory input from neighboring T5 to
T5 neurons and by excitatory input from Tm9. For TmY3, we see null direction suppression
predominantly via Mi14 and Mi4 inhibition and excitation via Mi1 and L5.

We summarize these findings now in the main manuscript in the Results [l. 237-249] and in Fig
4. In addition, we show full results for all T4 and T5 subtypes and for TmY3 as Extended Data
Figures.

*The authors have missed an opportunity to test their model through silencing experiments - inputs to T4*
*and T5 cells have been silenced experimentally (Strother et al. 2017; Serbe et al., 2016) - does silencing*
*these neurons in the trained models recapitulate experimental findings/impair motion detection by the*
*decoder network?*

Thank you for this suggestion. In addition to the analyses of input current contributions, we now
also simulated silencing experiments, and compared the model predictions with experimental
results reported in Strother et al, Neuron, 2017, finding good agreement for most effects, as
shown below. For these results, we averaged model-predictions over all models in the
model-cluster with correct T4c tuning (which, as explained above, is also the cluster with the
best task-performing model). The two column in panel a below are direct copies of Strother et al
2017 Figures 3b [left column] and 3c [right column], panel b shows, for comparison, the output of
our analyses. We emphasize that, for these analysis, only a qualitative comparison of the effect
is meaningful-- for example, Strother et al report Delta F/F from calcium imaging, whereas we
show standardized voltage responses. In addition, our 'silencing' analyses are based on simply
clamping the respective T4c inputs to 0, which is likely a crude approximation of the effect of
blocking synaptic transmission with shibire(ts1) and temperature increases. Nevertheless, the
models correctly capture that: i) Removal of Mi1 excitation removes the T4c response ii) Tm3:
Removal of Tm3 excitation decreases, but does not remove, the T4c response. iii) Mi4: Removal
of Mi4 does not remove the T4c response [it does, however, lead to a prolonged response in the
network model which is not observed experimentally] iv). Mi9: no effect on T4c.

a Screenshot from Strother et al. (2017) silencing experiments

b Our model

*Of the 19 cell types with asymmetric inputs, only 12 are predicted to be motion selective - can the authors*
 *comment on differences between the inputs of these 12 and the other 7?*

Thanks for this excellent suggestion. In response to your comment, we have now characterized
 the asymmetries of the inputs and in comparison to the different direction selectivities that our
 model predicts for these cells to understand the relationship (Fig. below). We find no trivial
 relationship between the asymmetry of the inputs and the predicted direction selectivity. We
 have included this new analysis as an Extended Data Figure and summarized in the Results.

*One of the most exciting findings is the prediction that TmY3 could possess direction selectivity*
 *independent of the T4/T5 pathway. This is a bold claim - that the model can be used to identify new motion*
 *sensors in the fly visual system that has been studied for more than 60 years. It seems reasonable (and*
 *feasible given that the authors' local collaborators) to ask the authors to validate this prediction with*
 *experimental data - whereas elsewhere the authors rely on published experimental data for comparisons,*
 *this prediction would require new experiments.*

This study has been a substantial undertaking, conducted by computational labs enabled by the
 availability of this rich and large dataset. This is a new era for computational neuroscience, one
 in which an enormous number of detailed, comprehensive, and experimentally testable
 predictions can be made through detailed and comprehensive modeling.

As we have responded earlier, we wish to emphasize that no neural activity measurements were
 used in constructing our model, and we already tested the predictions of our model against
 experimental measurements of neural activity across 26 studies. This is already a highly
 nontrivial validation of our model. Further, our model makes hundreds of pages of experimentally
 testable predictions (see supplement), of which this is but one prediction. We do not believe that
 just one more experimental validation of just one of these hundreds of predictions is necessary
 to substantiate the main claims of our paper.

Nonetheless, we are informed by our colleague Michael Reiser (HHMI Janelia), that unpublished
 work in progress in his lab does indeed hint at motion selectivity for TmY3. He says the whole
 picture for this cell type is more interesting and complex and they plan to report the results of
 their own study in the near future.

Further, the authors could discuss if TmY3 is expected to function like an HR detector or BL detector, or is
direction selectivity predicted to arise via a different mechanism?

As described in detail above, for TmY3, we see null direction suppression predominantly via
Mi14 and Mi4 inhibition and excitation via Mi1 and L5.

I would like to see some of the statistics of the decoder network. The manuscript argues that it cannot
compute motion itself, but is this true? Does the decoder only depend on the output of direction-tuned
neurons? Does it perform equally well when the weights for non-direction tuned neurons are forced to
zero? What is the minimal set of T and Tm cells required to detect optic flow?

Thanks for the suggestion. We now analyzed the relevance of each individual decoded cell type
for the optic flow task. We evaluated the increase in task-error when we replaced responses of
individual cells from a cell type by their spatio-temporal mean. We find that removing T4
subtypes strongly increases the task error. The decoder therefore most strongly relies on T4
cells which are predicted as on-motion detectors, but typically draws information from each
decoded cell type to encode the synthetic flow field. We have included this figure below as an
Extended Data Figure.

*Minor comments*

*Figure 1 depicts a male fly, though much of the connectome data used, I believe, comes from female flies.*

We updated figure 1.

*Winding et al. 2023 larval connectome paper should also be cited at line 7.*

Thank you, we have fixed this, and additionally also cited the newly published Flywire pre-prints.

*The meaning of lines 99-101 is not clear. The text argues that the DMN was optimized for a computational*
*task performed by the real biological network (i.e. fly visual system), but citations 44 & 76 refer to*
*mammalian cortex.*

Thank you, fixed. The citations were meant for the general idea of “task optimization”.

*Line 119: typo in “backpropagation”*

Thank you, fixed.

*Line 105 says that motion detection is a “challenging” computation. “Challenging” is too subjective a*
*term and can be removed. Arguably, theoretical models for motion detection proposed in the 1960’s by*
*Hassenstein & Reichert and Barlow & Levick are relatively simple.*

Thank you, fixed.

*Lines 283-285 claim that networks with different sparsity must use different computations. Why must this*
*be true?*

We have no mathematical proof that this statement is true, so we have removed this claim.

*Lines 333-336 claim that “DMN models generate meaningful predictions in absence of neural activity*
*measurements... (Fig 4)” feels too strong, since knowing actual tuning of Mi9 was critical in determining*
*“correct” tuning of T4.*

The tunings of T4 and Mi9 are correctly predicted by the cluster with the best task performance.
While our DMN ensemble generates multiple hypotheses, task performance can be used to rank
these hypotheses, and as we show in Extended Data Fig. 11, DMN models with better task
performance are also more likely to correctly predict neural activity. Even if this were not the
case, our connectome-constrained and task-optimized DMN framework is able to generate just a
small number of reasonable hypotheses for the neural activity of each cell type, without the use
of any neural activity. In the absence of our method and the connectome, this hypothesis space
would be extremely large since any neuron could potentially take on any visual tuning. We feel
that we have proven that our model ensemble generates “meaningful predictions in the absence
of neural activity”, even if multiple hypotheses are generated.

*The limitations of the approach (e.g. simplistic modeling of neural dynamics, and lack of electrical*
*synapses, neuromodulation and glia) are introduced at the outset. It would be nice if these limitations*
*were further elaborated/explored in the discussion.*

We have now expanded discussion of this issue.

Reviewer Reports on the First Revision:

Referees' comments:

Referee #1 (Remarks to the Author):

The authors have addressed most of my comments but one issue remains open. This concerns the question how incomplete knowledge of the connectome or the catalog of cell types would affect model performance. The importance of cell type information is of particular interest. Cell type information in the *Drosophila* visual system is more detailed and more complete than in almost all other circuits and species. How would model performance be affected if not all cell types were known, or if similar cell types were indistinguishable? This could be simulated by pooling some cell types, or by merging similar cell types. This knowledge would be valuable to assess the requirements to generalize the DMN approach because cell type information in other systems is usually incomplete or inaccurate. I recommend to the authors to address this issue, assuming that it can be addressed by a modest amount of additional work.

Referee #2 (Remarks to the Author):

I thank the authors for their substantial work revising the paper. My concerns have been sufficiently addressed and I think the new analyses determining which elements of the model and training process are crucial for replicating specific response properties is insightful.

Referee #4 (Remarks to the Author):

Many of our comments and concerns have been adequately addressed by the inclusion of new data and analyses - these include:

Additional supplemental data to document how the connectome was constructed from disparate datasets.

Figure 2D and extended data figure 9 are a welcome addition to the manuscript. These show that constraints from the connectome along with task optimization are required to accurately predict direction selectivity, but that connectomic constraints alone are sufficient to predict ON/OFF selectivity of cell types.

Updated analysis in Figure 4e, taking into account the best-performing cluster of models, captures the experimentally measured tuning of lamina cells.

Extended data figure 2 addresses differences in asymmetric input to motion selective and non-motion selective cell types, finding no strong correlation.

Some of our comments and concerns have been addressed, but have raised further questions:

Figure 4 and extended data figures uncovers a new potential mechanism for how direction selectivity arises in T4 and T5 cells, as well as the hypothesized direction selectivity of TmY3. The

DMN suggests that ND suppression is involved in computing motion in all three cell types. In addition, lateral interactions between T4 and T5 neurons of the same subtype is suggested as a novel mechanism for PD enhancement. This result seems to contradict experimental findings showing that PD enhancement contributes significantly to direction selectivity in T4 and T5 cells (e.g. Fisher et al., 2015; Haag et al., 2016 & 2017; Arenz et al., 2017). This discrepancy should be discussed.

In silico silencing of inputs to T4c qualitatively recapitulate experimental findings from Strother et al (2017). The authors say that for these simulations they “clamped the respective T4c inputs to zero”. It is not clear if only the inputs to T4c were clamped to zero, or all outputs of the silenced presynaptic cell type. The latter is a better approximation of the effect of a shibire(ts) experiment, which silences all synaptic transmission, not only the synaptic transmission to the downstream neurons of interest.

Extended data figure 3 is a welcome addition which shows some properties of the decoder network. However, this has raised some further concerns. These data show that the decoder network predominantly uses the output of T4 to compute motion. Incidentally, the DMN most accurately predicts the experimentally measured tuning of T4 (compared to T5). Could it be that the mechanism of direction selectivity in T4 is “simpler” or more directly constrained by the connectome than, e.g., direction selectivity in T5 (or TmY3)? Therefore, the DMN and decoder network learn this first, while direction selectivity in T5 is learned more sloppily (explaining the aberrant ON responsiveness in T5 even in the best performing models)? This seems to be a limitation of the whole approach, as it suggests that there are certain computations that the DMN is particularly good at learning, or certain computations are more constrained by the connectome than others (this also relates to our point below - how much of this is dependent on the specific task chosen?). If T4 is excluded from the decoder, is the network better at learning the experimentally measured tuning of T5? Does it affect the predicted tuning of TmY3?

One major concern is not addressed:

The authors explain that they chose local motion detection to optimize their model due to it being an important and well-studied computation of the optic lobe. However, given the results presented in extended data figure 3, it seems that training on one specific task allows the model to learn the tuning of one cell type to the detriment of others. We still suggest that it would be useful to readers (and within scope) to provide analysis to address how choice of decoder or task optimization affects the learned tuning of neurons in the network. The authors make strong claims that connectivity alone can be used to predict the function of individual neurons, but remains open as to how much this depends on the specific task used.

Minor:

“Whole brain connectome projects have just been completed for the larval and adult fruit fly” - larval connectome is not whole brain exactly (missing the SEZ) and adult whole brain connectome should point to refs 11, 16, and 17.

The authors should reference the new publication

<https://www.biorxiv.org/content/10.1101/2023.10.12.562119v2.full.pdf> - this paper contains connectivity of all neurons of one optic lobe (along with cell typing). The authors work here preceded the completion of the Matsliah et al. wiring diagram, but it would be useful for the authors

to note that going forward full connectivity could be used (rather than constructing an RNN from disparate datasets), and that there are important differences in cell type classifications between Matsliah et al. and this paper (for example, Tm5 and TmY neurons).

Relative to the points above, there are 148 references in this paper! Given that almost 100 references will need to be cut for publication (and choices will need to be made), the authors should do this trimming for any further revision.

Connectome is used rather liberally throughout the manuscript (for example, in this sentence in the Discussion “Knowledge of the connectome played a critical role in this success, in part by leading to a massive reduction in the number of free model parameters.”) - however, the authors have used wiring diagrams of portions of the optic lobe and generated an RNN from these constraints. Since connectome, to my knowledge, implies completeness (one would not use the term genome to refer to the sequencing of a subset of genes...), it would be less confusing for the authors to distinguish their data sources from newer more complete wiring diagrams (suggestions: refer to the data sources as partial connectomes or columnar connectomes or wiring diagrams).

Author Rebuttals to First Revision:

We thank you and the reviewers for their helpful comments. In this revision, we have performed new experiments and analysis to address three main questions:

1. The importance of having detailed cell type annotations. With new experiments, we show the results of coarse-graining our knowledge of the cell type of each neuron.
2. The mechanism of direction selectivity in T4, T5, and TmY3. With new analysis, we shed light on whether we see signatures of preferred direction enhancement (no) or null direction suppression (yes) and relate this to the literature.
3. The role of the computational task in task optimization. With new experiments, we show the importance of choosing ethologically relevant tasks for task optimization.

We believe that these additional experiments and analyses strengthen our main claim, that with (appropriate) task optimization, one can use connectomic measurements to construct remarkably predictive mechanistic simulations of the fruit fly visual system at single cell resolution.

A detailed response to the reviewers is attached.

Referee #1 (Remarks to the Author):

The authors have addressed most of my comments but one issue remains open. This concerns the question how incomplete knowledge of the connectome or the catalog of cell types would affect model performance. The importance of cell type information is of particular interest. Cell type information in the *Drosophila* visual system is more detailed and more complete than in almost all other circuits and species. How would model performance be affected if not all cell types were known, or if similar cell types were indistinguishable? This could be simulated by pooling some cell types, or by merging similar cell types. This knowledge would be valuable to assess the requirements to generalize the DMN approach because cell type information in other systems is usually incomplete or inaccurate. I recommend to the authors to address this issue, assuming that it can be addressed by a modest amount of additional work.

We agree that an incomplete knowledge of the connectome could impact the usefulness of connectomic constraints. We thank you for the suggestion of pooling cell types. Based on this suggestion, we performed two experiments training new models with pooled cell types. We artificially assumed some cell types to be indistinguishable, and thus modeled them with shared single neuron and synapse parameters (resting potentials, time constants, and unitary synapse strengths). The figure below summarizes these results and is now added as Extended Data Figure 15 of the revised manuscript.

1. **Full DMN Merge T4, T5:** We assumed that the four 'T4' subtypes were indistinguishable from each other, and also that the four 'T5' subtypes were indistinguishable, pooling them into one T4 type and one T5 type. This reduced the number of cell types to 58. We found that this model accurately predicts neural tuning, presumably because these highly similar subtypes share the same neurotransmitters, receptors, and biophysical properties.
2. **Full DMN Merge E/I:** We assumed that we had only three cell types, 'excitatory' (merging 37 cell types), 'inhibitory' (merging 22 cell types) or 'mixed' (merging 4 cell types), based on our knowledge of expressed presynaptic neurotransmitters and postsynaptic receptor types. This model performs similarly to the random DMN model, poorly predicting tuning curves and direction selectivity index.

Referee #2 (Remarks to the Author):

I thank the authors for their substantial work revising the paper. My concerns have been sufficiently addressed and I think the new analyses determining which elements of the model and training process are crucial for replicating specific response properties is insightful.

Referee #4 (Remarks to the Author):

Many of our comments and concerns have been adequately addressed by the inclusion of new data and analyses - these include:

Additional supplemental data to document how the connectome was constructed from disparate datasets. Figure 2D and extended data figure 9 are a welcome addition to the manuscript. These show that constraints from the connectome along with task optimization are required to accurately predict direction selectivity, but that connectomic constraints alone are sufficient to predict ON/OFF selectivity of cell types. Updated analysis in Figure 4e, taking into account the best-performing cluster of models, captures the experimentally measured tuning of lamina cells. Extended data figure 2 addresses differences in asymmetric input to motion selective and non-motion selective cell types, finding no strong correlation.

Some of our comments and concerns have been addressed, but have raised further questions:

Figure 4 and extended data figures uncovers a new potential mechanism for how direction selectivity arises in T4 and T5 cells, as well as the hypothesized direction selectivity of TmY3. The DMN suggests that ND suppression is involved in computing motion in all three cell types. In addition, lateral interactions between T4 and T5 neurons of the same subtype is suggested as a novel mechanism for PD enhancement. This result seems to contradict experimental findings showing that PD enhancement contributes significantly to direction selectivity in T4 and T5 cells (e.g. Fisher et al., 2015; Haag et al., 2016 & 2017; Arenz et al., 2017). This discrepancy should be discussed.

Thank you for this comment. We now realize that our wording was potentially misleading due to the loaded meaning of the words “enhancement” and “suppression” in the motion detection subfield. These terms are used to refer to the *nonlinear* enhancement or suppression of the neural response to motion stimuli in the preferred or null direction relative to a “*linear sum*” null model. We have now reworded these sentences in our paper to make clear our original intent, which was to comment on a different effect.

Further, we now directly address the question of preferred direction enhancement (PDE) and null direction suppression (NDS) relative to a “linear sum” model by performing a new analysis which exactly replicates the voltage recordings and analysis of Fig 4 of Gruntman 2018 (replicated below). In our models, for all three cell types T4c, T5c, and TmY3, we only observe NDS and no (nonlinear) PDE. These results for T4 and T5 are in excellent agreement with Gruntman 2018 and Gruntman 2019. We should note that there is considerable experimental disagreement on this topic, largely due to differences in experimental methodology (calcium imaging vs voltage recordings). Therefore we emphasize that the predictions of voltage responses in our model are in excellent agreement with measurements of voltage responses by Gruntman 2018 and Gruntman 2019.

We have added these new results to the revised manuscript as Extended Data Fig 14.

In silico silencing of inputs to T4c qualitatively recapitulate experimental findings from Strother et al (2017). The authors say that for these simulations they “clamped the respective T4c inputs to zero”. It is not clear if only the inputs to T4c were clamped to zero, or all outputs of the silenced presynaptic cell type. The latter is a better approximation of the effect of a shibire(ts) experiment, which silences all synaptic transmission, not only the synaptic transmission to the downstream neurons of interest.

Thank you for requesting this clarification. Indeed we clamped all outputs of the silenced presynaptic cell type to zero, exactly as expected of a shibire(ts) silencing experiment. We have clarified this.

Extended data figure 3 is a welcome addition which shows some properties of the decoder network. However, this has raised some further concerns. These data show that the decoder network predominantly uses the output of T4 to compute motion. Incidentally, the DMN most accurately predicts the experimentally measured tuning of T4 (compared to T5). Could it be that the mechanism of direction selectivity in T4 is “simpler” or more directly constrained by the connectome than, e.g., direction selectivity in T5 (or TmY3)? Therefore, the DMN and decoder network learn this first, while direction selectivity in T5 is learned more sloppily (explaining the aberrant ON responsiveness in T5 even in the best performing models)? This seems to be a limitation of the whole approach, as it suggests that there are certain computations that the DMN is particularly good at learning, or certain computations are more constrained by the connectome than others (this also relates to our point below - how much of this is dependent on the specific task chosen?). If T4 is excluded from the decoder, is the network better at learning the experimentally measured tuning of T5? Does it affect the predicted tuning of TmY3?

One major concern is not addressed:

The authors explain that they chose local motion detection to optimize their model due to it being an important and well-studied computation of the optic lobe. However, given the results presented in extended data figure 3, it seems that training on one specific task allows the model to learn the tuning of one cell type to the detriment of others. We still suggest that it would be useful to readers (and within scope) to provide analysis to address how choice of decoder or task optimization affects the learned tuning of neurons in the network. The authors make strong claims that connectivity alone can be used to predict the function of individual neurons, but remains open as to how much this depends on the specific task used.

We believe that we have convincingly demonstrated our central claim: that task optimization (with an appropriate task) leads to significantly more accurate predictions of neural activity, than without task optimization (random parameters). We are not convinced that an expansive exploration of the choice of task and decoder architectures will further strengthen this claim.

Nevertheless, we have now trained an additional DMN variation with a new autoencoding task (predicting the input from the input) to address this concern. We do not expect this to be an ethologically relevant task, and do not believe that the fly visual system preserves all the information from the visual inputs, but rather computes behaviorally relevant features in a lossy manner. Further, this task does not force the visual system to perform temporal computations on visual inputs in the same way as the optic flow task. Consistent with this, we find that the DMN trained to perform autoencoding does not predict neural tuning more accurately than the DMN with random parameters. This supports our claim that it is important to choose a behaviorally relevant task for task optimization.

Minor:

“Whole brain connectome projects have just been completed for the larval and adult fruit fly” - larval connectome is not whole brain exactly (missing the SEZ) and adult whole brain connectome should point to refs 11, 16, and 17.

The authors should reference the new publication <https://www.biorxiv.org/content/10.1101/2023.10.12.562119v2.full.pdf> - this paper contains connectivity of all neurons of one optic lobe (along with cell typing). The authors work here preceded the completion of the Matsliah et al. wiring diagram, but it would be useful for the authors to note that going forward full connectivity could be used (rather than constructing an RNN from disparate datasets), and that there are important differences in cell type classifications between Matsliah et al. and this paper (for example, Tm5 and TmY neurons).

Relative to the points above, there are 148 references in this paper! Given that almost 100 references will need to be cut for publication (and choices will need to be made), the authors should do this trimming for any further revision.

Connectome is used rather liberally throughout the manuscript (for example, in this sentence in the Discussion “Knowledge of the connectome played a critical role in this success, in part by leading to a massive reduction in the number of free model parameters.”) - however, the authors have used wiring diagrams of portions of the optic lobe and generated an RNN from these constraints. Since connectome, to my knowledge, implies completeness (one would not use the term genome to refer to the sequencing of a subset of genes...), it would be less confusing for the authors to distinguish their data sources from newer more complete wiring diagrams (suggestions: refer to the data sources as partial connectomes or columnar connectomes or wiring diagrams).

Thank you. We have addressed these minor comments. Regarding the larval connectome, we note that Winding et al. 2023 claim to have mapped the “complete CNS”, writing on page 10: “Our EM volume contains the complete CNS (brain, SEZ, and nerve cord), allowing us to assess communication between the brain and the rest of the CNS.”

Reviewer Reports on the Second Revision:

Referees' comments:

Referee #1:

Remarks to the Author:

The authors have addressed my concerns. I have no further points. Congratulations!

Referee #4:

Remarks to the Author:

The majority of minor comments have been addressed and terms “enhancement” and “suppression” have been sufficiently clarified in the text. However, the reference list is still too long.

The addition of Extended Data Fig 14 and comparison to Gruntman et al. 2018 sufficiently addresses our questions about the mechanism of direction selectivity in T4, T5 and TmY3 in the model.

Training the model on an autoencoding task addresses our concern about task optimization, and does demonstrate that task optimization is required to predict known tuning of direction selective cell types.

One comment is not yet addressed: the authors should explain (in the Discussion) why the DMN seems to be better at predicting direction selectivity in T4 over T5. Could this be a limitation of the model in the kinds of computations it can learn?